# Spatial N-glycan rearrangement on α5β1 integrin nucleates galectin-3 oligomers to determine endocytic fate

Massiullah Shafaq-Zadah [1,2,14] ✉, Estelle Dransart[1,2,14], Ilyes Hamitouche[1,2,3], Christian Wunder [1,2], Valérie Chambon[1], Cesar A. Valades-Cruz [1,4,5,13], Ludovic Leconte[4,5], Nirod Kumar Sarangi[6], Jack Robinson[6], Siau-Kun Bai[1], Raju Regmi [7], Aurélie Di Cicco[7], Agnès Hovasse [8,9], Richard Bartels[3], Ulf J. Nilsson [10], Sarah Cianférani-Sanglier [8,9], Hakon Leffler [11], Tia E. Keyes[6], Daniel Lévy [7], Stefan Raunser [12], Daniel Roderer [3] ✉ & Ludger Johannes [1,2] ✉

Membrane glycoproteins frequently adopt different conformations when altering between active and inactive states. Here, we discover a molecular switch that exploits dynamic spatial rearrangements of N-glycans during such conformational transitions to control protein function. For the conformationally switchable cell adhesion glycoprotein α5β1 integrin, we find that only the bent-closed state arranges N-glycans to nucleate the formation of up to tetrameric oligomers of the glycan-binding protein galectin-3. We propose a structural model of how these galectin-3 oligomers are built and how they clamp the bent-closed state to select it for endocytic uptake and subsequent retrograde trafficking to the Golgi for polarized distribution in cells. Our findings reveal the dynamic regulation of the glycan landscape at the cell surface to achieve oligomerization of galectin-3. Galectin-3 oligomers are thereby identified as functional decoders of defined spatial patterns of N-glycans on specifically the bent-closed conformational state of α5β1 integrin and possibly other integrin family members.

Integrins are heterodimeric glycoproteins that are made up of one out of 18 α and one out of 8 β subunits, with key functions in the adhesion of cells to extracellular matrix ligands such as fibronectin[1–3]. Integrins exist in a continuum of conformations between bent-closed non-ligand-bound (also termed inactive) and extended ligand-bound (also termed active) states[4]. N-glycans affect the equilibrium between these two conformers[5], but the underlying mechanisms remain largely unexplored.

Glycosylation is required for integrin functions[5–13]. Membrane distal N-glycosylation sites within the β-propeller domain of α5 and at

[1]Chemical Biology of Cancer Unit, Institut Curie, U1339 INSERM, UMR3666 CNRS, PSL Research University, Paris, France. [2]Chemical Biology of Cancer Unit, SAIRPICO Team, U1339 INSERM, Institut Curie, Inria Center at University of Rennes, Paris, France. [3]Leibniz-Forschungsinstitut für Molekulare Pharmakologie (FMP), Berlin, Germany. [4]SERPICO Project Team, Inria-UMR144 CNRS Institut Curie, PSL Research University, Paris, France. [5]SERPICO Project Team, Inria Centre Rennes-Bretagne Atlantique, Rennes, France. [6]School of Chemical Sciences and INSIGHT Centre for Data Analytics, Dublin City University, Dublin, Ireland. [7]Physique des Cellules et Cancer, Institut Curie, UMR168 CNRS, PSL Research University, Paris, France. [8]BioOrganic Mass Spectrometry Laboratory, IPHC UMR 7178, CNRS, Strasbourg University, Strasbourg, France. [9]National Proteomic Infrastructure ProFI – FR2048, Strasbourg, France. [10]Department of Chemistry, Lund University, Lund, Sweden. [11]Section MIG (Microbiology, Immunology, Glycobiology), Department of Laboratory Medicine, Lund University, Lund, Sweden. [12]Department of Structural Biochemistry, Max Planck Institute of Molecular Physiology, Dortmund, Germany. [13]Present address: Institute of Hydrobiology, Chinese Academy of Sciences, Wuhan, China. [14]These authors contributed equally: Massiullah Shafaq-Zadah, Estelle Dransart. ✉e-mail: massiullah.shafaq-zadah@curie.fr; roderer@fmp-berlin.de; ludger.johannes@curie.fr

the βI domain of $β_1$ integrins are required for heterodimerization and efficient cell surface expression[7,9]. Membrane proximal N-glycosylation sites are needed for the biological activities of integrins, including interactions with other membrane proteins such as the epidermal growth factor receptor[9,14].

Both clathrin-dependent and independent endocytic mechanisms have been documented for integrin internalization[15–23]. Clathrin, clathrin adapters such as Dab2, Numb, Eps8, AP2, and the scission GTPase dynamin-2[16,17,24–26] have all been linked to the dynamic turnover of focal adhesions and in the regulation of active $β_1$ integrin endocytosis near ventral focal adhesions and in the leading edge of migrating cells[17,18,25].

While the clathrin pathway remains the best characterized endocytic route[27–29], mechanisms by which membranes are bent and cargoes selected without the clathrin coat are also under investigation[21,23,30–32]. Glycans on proteins and lipids, as well as glycan-binding proteins of the galectin family, are at the center stage in one of the proposed models, termed glycolipid-lectin (GL-Lect) driven endocytosis[33–37]. Notably, galectin-3 (Gal3) has been analyzed in depth in this context[22,23,38]. For building endocytic pits, Gal3 recognizes N-glycans on glycoproteins (including integrins) and oligomerizes. Only oligomerization competent Gal3 acquires the capacity to induce the glycosphingolipid (GSL) dependent formation of tubular endocytic pits from which clathrin-independent endocytic carriers (CLICs) emerge[23]. Of note, Gal3 has also been described to form lattice assemblies at the plasma membrane that negatively affect endocytosis[39]. The interplay between GL-Lect and lattice processes controls the cell surface dynamics of glycoproteins, including integrins, in an intertwined manner[40–43].

Gal3 has proinflammatory activity, and its expression has been associated with many pathophysiological situations. For example, increased levels of Gal3 are associated with heart failure due to fibroinflammation in myocardium remodeling[44–46] and with chronic kidney disease, including defects in glomerular filtration, proteinuria, and high risk of chronic renal fibrosis[47,48]. Other fibrotic diseases, including Alzheimer's and Huntington's neuroinflammation, and pulmonary infections such as idiopathic pulmonary fibrosis, Hermansky-Pudlak syndrome, and COVID-19, also implicate altered Gal3 expression levels[49–53], identifying the protein as a potential therapeutic target. The mechanisms by which Gal3 regulates the underlying biological functions remain poorly explored.

The oligomerization capacity of Gal3 has been shown to be important for its function in endocytosis[23]. However, the shapes and assembly mechanisms of Gal3 oligomers remained controversial[54,55]. Here, we have found that a conformational state-specific spatial arrangement of N-glycans on $α_5β_1$ integrin sets its capacity to nucleate Gal3 oligomers that range up to tetramers. This determines the integrin's endocytic fate and ensuing intracellular compartmentalization. We have termed this glycan pattern recognition mechanism the conformational glycoswitch, which may also apply to other members of the integrin family. The conformational glycoswitch positions glycosylation as a highly dynamic regulator of integrin function at the cell surface.

## Results

### Retrograde trafficking of inactive bent-closed $α_5β_1$ integrin depends on Gal3

We have previously described that only the inactive bent-closed non-ligand-bound conformational state of $α_5β_1$ integrin (Fig. 1A), and not the active extended ligand-bound state, follows the retrograde trafficking route from the plasma membrane to the Golgi apparatus, from where the protein then undergoes polarized secretion for its dynamic localization to the leading edge of migrating cells[56]. We discovered this dichotomic behavior using well-characterized conformational state-specific antibodies[57–60], i.e., mAb13 against the βI domain of the $β_1$ chain of inactive bent-closed $α_5β_1$ integrin[61], and 9EG7 against the

EGF-repeat leg domain of the $β_1$ chain of active ligand-bound $α_5β_1$ integrin[62]. These antibodies were modified with benzylguanine (BG) to be captured in live cell antibody uptake experiments by a GFP-labeled SNAP-tag fusion protein that was localized to the lumen of the Golgi apparatus, using a HeLa cell line that expresses endogenous Gal3 (Fig. 1B and Supplementary Fig. 1A)[56,63].

Here, we found that retrograde transport of inactive bent-closed $α_5β_1$ integrin (mAb13) from the plasma membrane to the Golgi apparatus was strongly decreased by a membrane impermeable inhibitor of Gal3 (GB0149-03) that we termed I3 in the current study[64,65] (Fig. 1C), and stimulated by exogenous Gal3 (Fig. 1D). In contrast, the depletion of clathrin heavy chain (CHC) did not inhibit but rather increased retrograde trafficking of inactive bent-closed $α_5β_1$ integrin (mAb13) (Fig. 1E and Supplementary Fig. 1B). This was not due to a possible effect of CHC depletion on Golgi exit, since BG-tagged mAb13 that undergoes retrograde trafficking was irreversibly captured in the Golgi apparatus by the GalT-GFP-SNAP-tag fusion protein (Fig. 1A, B). CHC depletion slightly inhibited endocytic uptake of inactive bent-closed $α_5β_1$ integrin (Supplementary Fig. 1C, mAb13)[66] without a major effect on cell surface integrin levels (Supplementary Fig. 1D). As controls, we showed that the endocytic uptake of the clathrin pathway marker transferrin was strongly inhibited under CHC depletion conditions (Supplementary Fig. 1E)[67], while Gal3 inhibition with I3 had no effect on Tf uptake (Supplementary Fig. 1F).

Using the BG/SNAP-tag assay, we demonstrate that exogenously added BG-modified Gal3 also undergoes retrograde trafficking (Supplementary Fig. 1G, H). Its accumulation in the Golgi apparatus was found to be increased upon CHC depletion (Fig. 1F), while its endocytosis was partly inhibited under these conditions (Supplementary Fig. 1I). Taken together, our findings strongly suggest that only the fractions of inactive bent-closed $α_5β_1$ integrin and of Gal3 that enter cells by clathrin-independent endocytosis gain access to the retrograde route.

Internalized inactive bent-closed $α_5β_1$ integrin (mAb13) and Gal3 both strongly colocalized on endosomal membranes with the Vps26 component of the retromer complex (Fig. 1G, H), a retrograde sorting machinery[68]. The colocalization of internalized active ligand-bound $α_5β_1$ integrin (9EG7) with Gal3 and Vps26 was much lower (Fig. 1G, H), as expected from the inability of this conformer to undergo retrograde trafficking[56]. Furthermore, inactive bent-closed endosomal $α_5β_1$ integrin (mAb13) specifically co-immunoprecipitated both the retromer subunit Vps35 and Gal3, while active ligand-bound $α_5β_1$ integrin (9EG7) did not (Fig. 1I and Supplementary Fig. 1J).

These data demonstrate that retrograde trafficking of inactive bent-closed $α_5β_1$ integrin is Gal3-dependent and independent of the clathrin pathway, and further suggest that the inactive $α_5β_1$ integrin conformer and Gal3 traffic together to the Golgi apparatus.

### Specifically the inactive bent-closed $α_5β_1$ integrin is internalized by GL-Lect driven endocytosis

Human retinal pigment epithelial cells (RPE-1) express endogenous Gal3 (Supplementary Fig. 1A). A functional link between $α_5β_1$ integrin and Gal3 was shown previously in these highly polarized and migratory cells[22,23,69–71]. Here, we investigated whether the Gal3 dependency for retrograde trafficking of inactive bent-closed $α_5β_1$ integrin was already set at the plasma membrane. This integrin conformer efficiently interacted with Gal3 in co-immunoprecipitation experiments (Fig. 2A, mAb13), and colocalized with Gal3 by immunofluorescence at the plasma membrane (mAb13 for $β_1$ integrin, Fig. 2B top; mAb16 for $α_5$ integrin, Supplementary Fig. 2A left) and in endocytic structures (Fig. 2C, top). Both at the plasma membrane (Supplementary Fig. 2B) and in endocytic structures (Supplementary Fig. 2C), colocalization was much reduced with a Gal3 mutant whose N-terminal oligomerization domain was deleted (Gal3ΔNter)[72]. Gal3ΔNter is deficient in all functions that have been attributed to Gal3, including GL-Lect driven

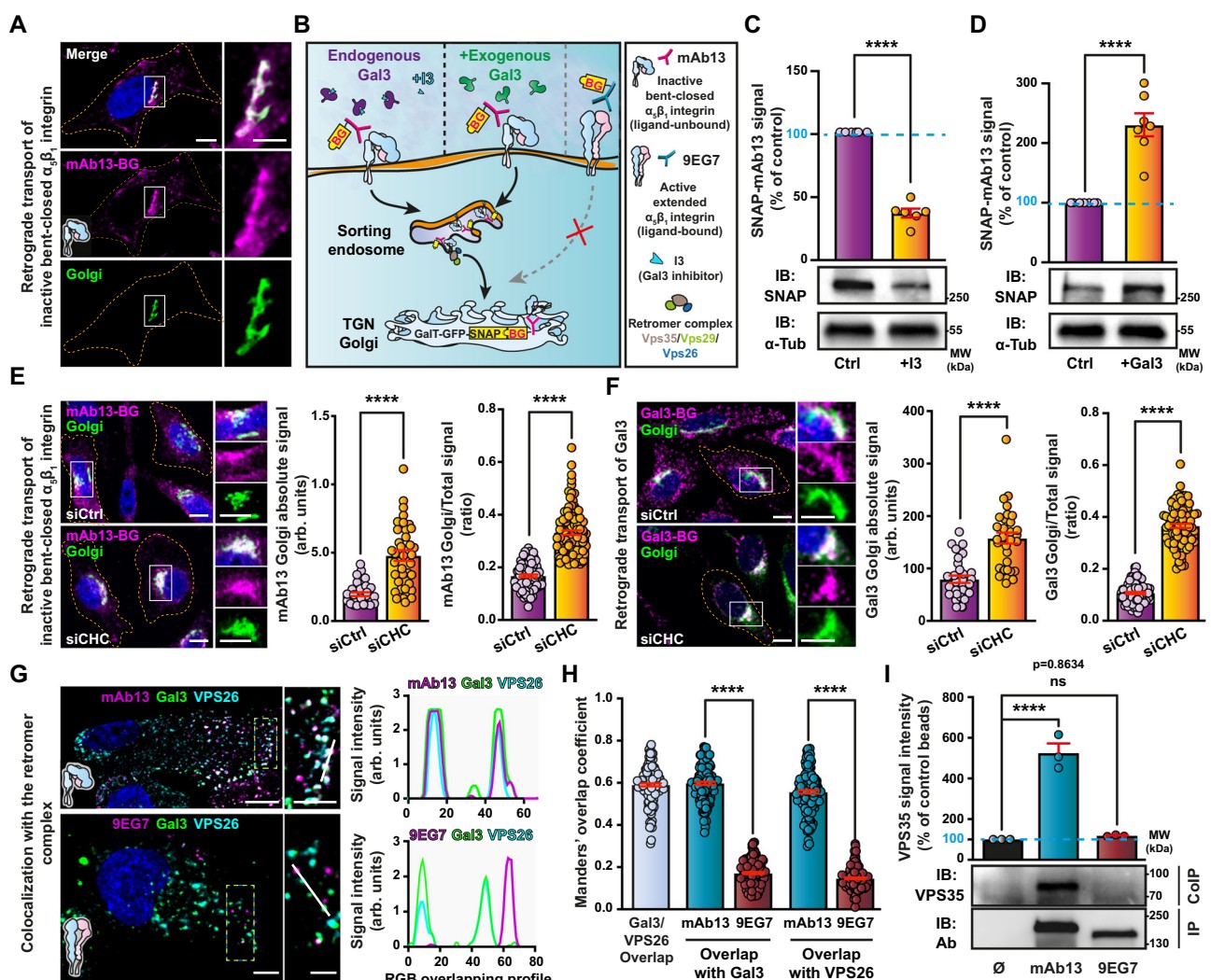

**Fig. 1 | Retrograde trafficking of inactive bent-closed $\alpha_5\beta_1$ integrin.**
**A** Continuous incubation of GalT-GFP-SNAP expressing HeLa cells for 1 h at 37 °C with Cy3 and benzylguanine (BG)-labeled mAb13 (mAb13-BG). At least three independent experiments with similar results were conducted. Scale bars = 10 μm. Nuclei in blue. **B** Schematic of experimental retrograde trafficking setup. Active ligand-bound $\alpha_5\beta_1$ integrin (9EG7) does not undergo retrograde trafficking[56]. GalT-GFP-SNAP expressing HeLa cells that had continuously been incubated for 3 h at 37 °C with mAb13-BG with or without I3 (10 μM) (**C**), or exogenous Gal3 (200 nM) (**D**). mAb13-BG that reached the Golgi was quantified by immunoblotting (IB: SNAP). α-tubulin was used for normalization. $n = 6$ (**C**) and $n = 7$ (**D**) independent experiments. Means ± SEM, two-sided unpaired $t$-test; ****$P < 0.0001$. **E** As in (**A**) in the control condition (siCtrl) or after clathrin depletion (siCHC). Absolute intensities of mAb13 fluorescent signals in the Golgi (left) or ratios of mAb13 fluorescent signals in the Golgi over total internalized mAb13 signals (right) were quantified. Left graph, $n = 31$ (for siCtrl) and $n = 36$ (for siCHC) cells, representative of three independent experiments. Right graph, $n = 94$ (for siCtrl) and $n = 106$ (for siCHC) cells, compiled data from three independent experiments. Means ± SEM, two-sided unpaired $t$-test; ****$P < 0.0001$. Scale bars = 10 μm. Nuclei in blue. **F** Cy3 and BG-

labeled Gal3 (200 nM) was incubated for 45 min at 37 °C with GalT-GFP-SNAP expressing HeLa cells. As in (**E**), absolute intensities or ratios of Gal3 fluorescence signals were quantified. Left graph, $n = 30$ (for siCtrl) and $n = 32$ (for siCHC) cells, representative of three independent experiments. Right graph, $n = 90$ (for siCtrl) and $n = 96$ (for siCHC) cells, compiled data from three independent experiments. Means ± SEM, unpaired two-sided $t$-test; ****$P < 0.0001$. Scale bars = 10 μm. Nuclei in blue. **G** Sequential binding of Gal3 (200 nM) and mAb13 or 9EG7 antibodies to RPE-1 cells at 4 °C, incubation for 15 min at 37 °C, followed by labeling of fixed cells for Vps26. The right panels show intensities along the white lines in the zooms to the left. Scale bars = 10 μm and 5 μm for zoomed insets. Nuclei in blue. **H** Quantification of overlaps from experiments as in (**G**). $n = 150$ cells per condition, compiled from three independent experiments. Means ± SEM, unpaired two-sided $t$-test; ****$P < 0.0001$. **I** mAb13 or 9EG7 were continuously incubated for 20 min at 37 °C with RPE-1 cells and then immunoprecipitated (IP). Cells without antibodies (∅) served as controls. Immunoblotting for Vps35 (IB: VPS35) and antibodies (IB: Ab). $n = 3$ independent experiments, means ± SEM, one-way ANOVA; ns = $P > 0.05$, ****$P < 0.0001$.

endocytosis[23]. This finding indicates that Gal3 oligomerization is required for its efficient co-distribution with the inactive bent-closed $\alpha_5\beta_1$ integrin conformer.

For active ligand-bound $\alpha_5\beta_1$ integrin, interaction (Fig. 2A, 9EG7) and colocalization (9EG7 for the $\beta_1$ chain, Fig. 2B bottom; SNAKA51 for the $\alpha_5$ chain, Supplementary Fig. 2A right) with Gal3 were very low. This dichotomic colocalization behavior with Gal3 between inactive bent-closed (mAb13) and active ligand-bound (9EG7) $\alpha_5\beta_1$ integrin was even more pronounced at the leading edge of migrating cells

(Supplementary Fig. 2D for plasma membrane, and Supplementary Fig. 2E for endosomes). To what extent other components of the membrane domains to which these conformers are localized also contribute to trafficking remains to be studied. Time-resolved imaging by lattice light sheet microscopy furthermore demonstrated the exclusive association of inactive bent-closed $\alpha_5\beta_1$ integrin (mAb13) and Gal3 onto dynamic population 2 (Pop 2) co-tracks, when compared to the active ligand-bound state (9EG7) (Fig. 2D, E and Supplementary Fig. 2F; Supplementary Movies 1 and 2).

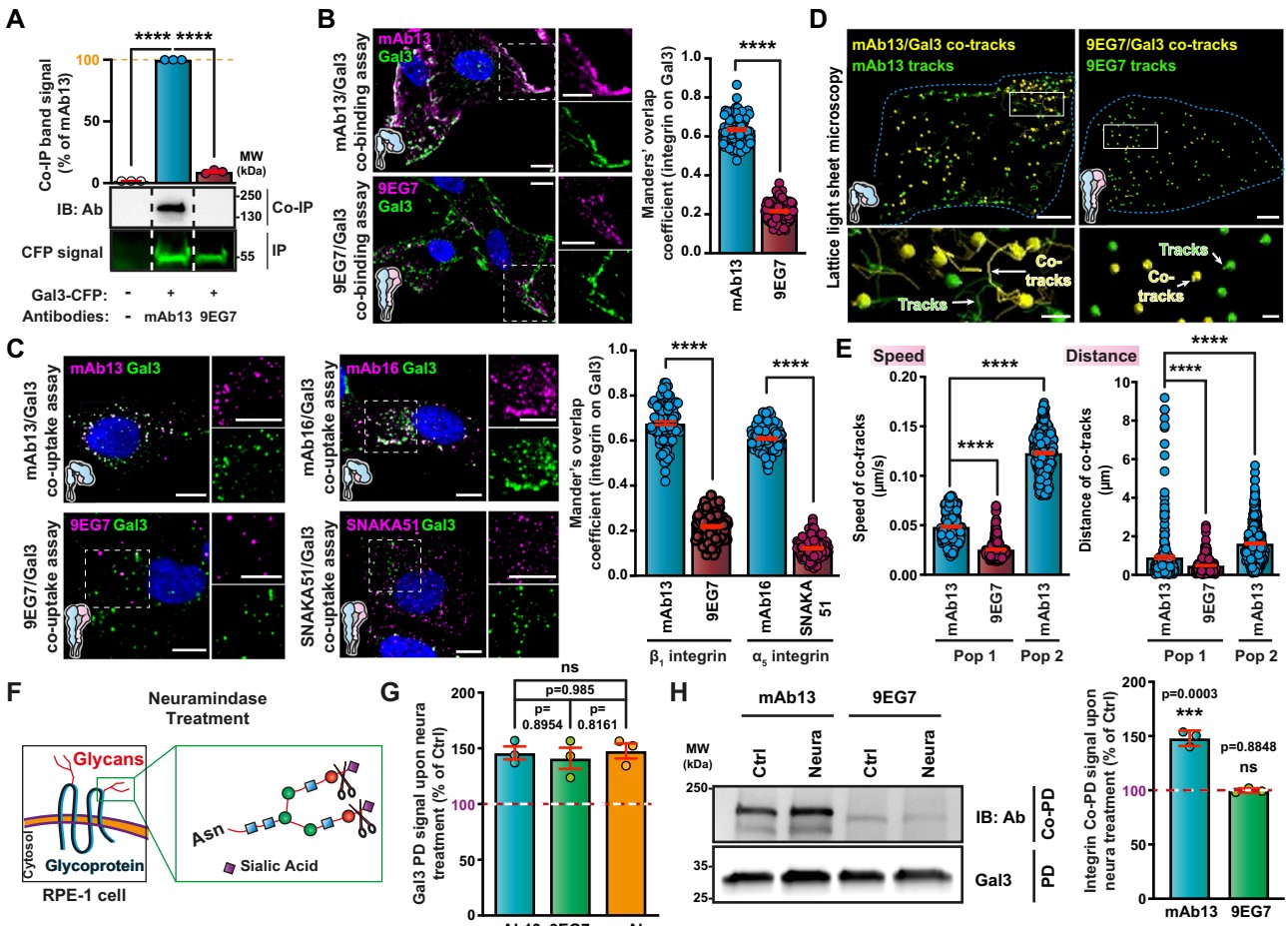

**Fig. 2 | Gal3 specifically interacts with inactive bent-closed $\alpha_5\beta_1$ integrin. A** GFP-Trap pulldown of Gal3-CFP transiently expressed in RPE-1 cells and immunoblotting for co-immunoprecipitated (Co-IP) mAb13 and 9EG7 (IB: Ab) that had been surface-bound (4 °C) to these cells. CFP fluorescent signal was used for normalization. $n = 3$ independent experiments, means ± SEM, one-way ANOVA; ****$P < 0.0001$. **B** Colocalization of exogenous Gal3 (200 nM) with mAb13 or 9EG7 antibodies after incubations at 4 °C with RPE-1 cells. $n = 100$ cells per condition, compiled from three independent experiments. Means ± SEM, unpaired two-sided $t$-test; ****$P < 0.0001$. Scale bars = 10 μm and 5 μm for zoomed insets. Nuclei in blue. **C** RPE-1 cells were sequentially incubated at 4 °C with Gal3 (200 nM) and the indicated conformational state-specific antibodies and then shifted for 10 min to 37 °C. Overlap of fluorescence signals from $n = 110$ (for mAb13/9EG7) and $n = 100$ (for mAb16/SNAKA51) cells, compiled from three independent experiments. Means ± SEM, unpaired two-sided $t$-test; ****$P < 0.0001$. Scale bars = 10 μm and 5 μm for zoomed insets. Nuclei in blue. **D** Co-tracking by lattice light sheet microscopy of Gal3 with mAb13 (7 cells) or 9EG7 (10 cells) for 3 min at 27 °C. Co-tracks in yellow,

Gal3-negative mAb13 or 9EG7 tracks in green. Scale bars = 5 μm. Inset: 3D rendering of cargo (mAb13 or 9EG7 antibodies) detection (spheres) and tracks (lines) from representative cells. At least three independent experiments with similar results were conducted. Scale bars = 1 μm. **E** Left: Quantification of Gal3/mAb13 (blue bars) or Gal3/9EG7 co-track (red bars) velocities: Co-tracks with speeds inferior (Pop 1) or superior to 0.08 μm/s (Pop 2). Right: Co-tracking distances within the two speed-delimited populations. $n = 289$, $n = 426$, and $n = 399$ co-tracks were analyzed in mAb13-Pop1, 9EG7-Pop1 and mAb13-Pop2 conditions, respectively. Means ± SEM, one-way ANOVA; ****$P < 0.0001$. **F** Schematic of neuraminidase (sialidase) activity. **G, H** RPE-1 cells were sequentially incubated at 4 °C with SETA-555 labeled Gal3 and mAb13 or 9EG7 antibodies. **G** Neuraminidase (neura) equally increased the total amount of SETA-555 Gal3 that was bound to and pulled down from cells in all conditions. $n = 3$ independent experiments, means ± SEM, one-way ANOVA, ns = $P > 0.05$. **H** Experiments as in (**G**) in which the pulled down (Co-PD) of $\alpha_5\beta_1$ integrin conformers with SETA-555 Gal3 (PD) was analyzed. $n = 3$ independent experiments, means ± SEM, unpaired two-sided $t$-test; ns = $P > 0.05$, ***$P < 0.0002$.

Terminal modification of N-glycans by $\alpha 2$-6-linked sialic acids has been shown to prevent Gal3 binding[73,74]. Using neuraminidase to remove sialic acids from plasma membrane glycans (Fig. 2F and Supplementary Fig. 2G, SiaFind), we indeed measured a 50% increase in global cell surface Gal3 binding (Supplementary Fig. 2G, Gal3), as described previously[38]. This increase in Gal3 binding was robustly observed and unaffected by the presence of the integrin conformational state-specific antibodies employed in our study (Fig. 2G). However, the neuraminidase treatment only enhanced the pulldown of mAb13-labeled bent-closed $\alpha_5\beta_1$ integrin with Gal3, and not that of 9EG7-labeled extended ligand-bound $\alpha_5\beta_1$ integrin (Fig. 2H). This again underlined the dichotomic relationship between the two conformational states. Of note, the binding capacity of both antibodies to their specific conformers was not or only very little affected upon surface

desialylation (Supplementary Fig. 2H), making it unlikely that these antibodies recognize different $\alpha_5\beta_1$ integrin glycoforms.

In accordance with its strong interaction and colocalization with Gal3, the endocytosis of inactive bent-closed $\alpha_5\beta_1$ integrin (mAb13) turned out to be GL-Lect driven, since uptake was (i) inhibited by the Gal3 inhibitor I3 (Fig. 3A, B, top); (ii) stimulated by exogenously added wild-type Gal3 (Fig. 3A, C, D, top), but not by Gal3ΔNter (Supplementary Fig. 3A); and (iii) inhibited by the depletion of another component of the GL-Lect machinery, GSLs, using the glucosylceramide synthase inhibitor Genz-123346[75] (Fig. 3E, F, top), that also inhibited the endocytosis of Gal3 itself (Supplementary Fig. 3B). In contrast, all these treatments did not affect the uptake of active ligand-bound 9EG7-labeled $\alpha_5\beta_1$ integrin (Fig. 3B–D, F, bottom), nor that of transferrin (Supplementary Figs. 1F and 3C). The dichotomic behavior between

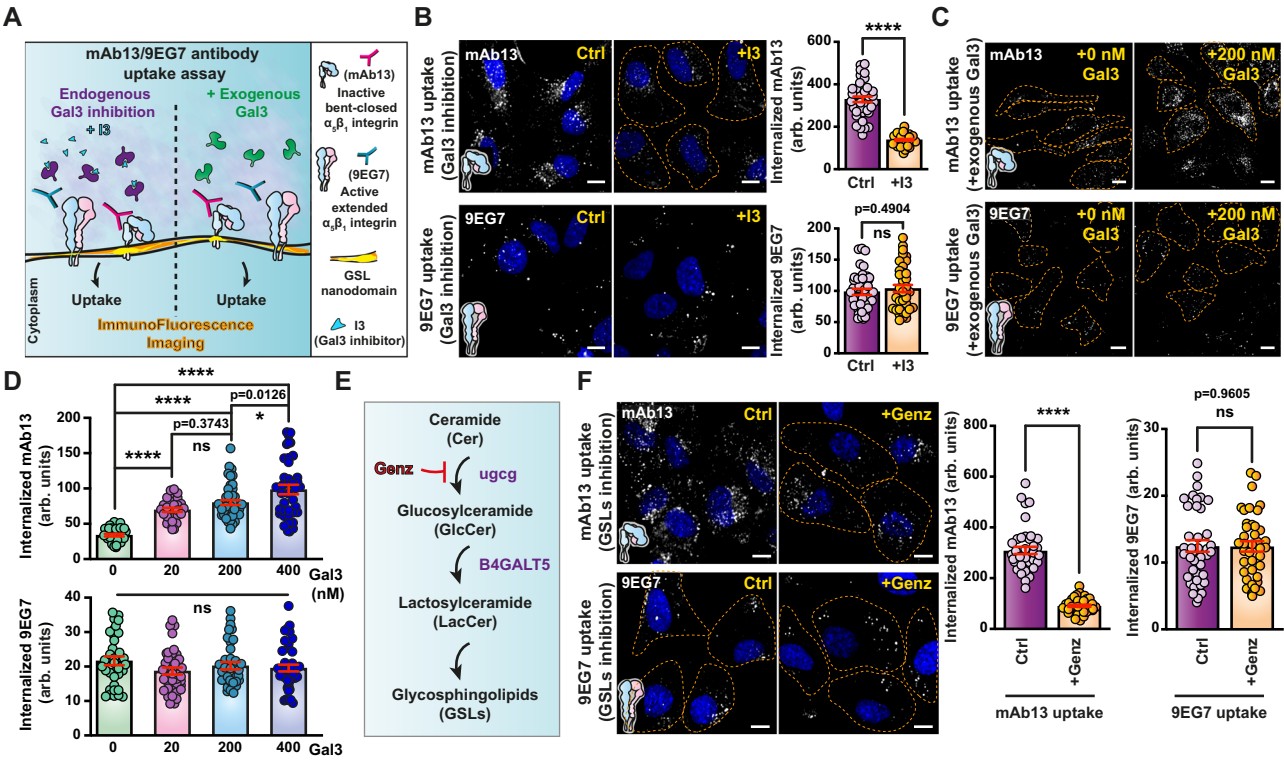

**Fig. 3 | Gal3 specifically drives the endocytosis of inactive bent-closed $\alpha_5\beta_1$ integrin. A** Schematic of experimental procedures for endocytosis studies with $\alpha_5\beta_1$ integrin conformers. **B** Effect of Gal3 inhibitor I3 on $\alpha_5\beta_1$ integrin endocytosis. Quantification by confocal microscopy of internalized mAb13 (3 µg/mL) and 9EG7 (10 µg/mL) after continuous incubation for 5 min at 37 °C with RPE-1 cells (dashed lines) pre-treated with I3 (+I3, 10 µM). $n = 40$ cells per condition, representative of three independent experiments. Means ± SEM, unpaired two-sided $t$-test; ns = $P >$ 0.05, ****$P < 0.0001$. Scale bars = 10 µm. Nuclei in blue. **C** Effect of exogenous Gal3 on $\alpha_5\beta_1$ integrin endocytosis. After mAb13 or 9EG7 binding to RPE-1 cells (dashed lines) at 4 °C, internalization was measured after 10 min incubation at 37 °C in the presence of the indicated concentrations of exogenous Gal3. Scale bars = 10 µm.

**D** Quantification from experiments as in (**C**) of fluorescence signals of internalized mAb13 or 9EG7. Top graph (mAb13), $n = 31$ (for 0 nM), 30 (for 20 nM), 46 (for 200 nM), and 40 (for 400 nM) cells. Bottom graph (9EG7), $n = 35$ (for 0 nM), 38 (for 20 nM), 37 (for 200 nM), and 44 (for 400 nM) cells. Representative of three independent experiments. Means ± SEM, one-way ANOVA; ns = $P > 0.05$, *$P < 0.05$, ****$P < 0.0001$. **E** Pathway of GSL biosynthesis and effect of Genz. **F** Effect of GSL depletion on $\alpha_5\beta_1$ integrin endocytosis. Quantification of internalized mAb13 (3 µg/mL) and 9EG7 (10 µg/mL) antibodies after continuous incubation for 5 min at 37 °C with RPE-1 cells in conditions of GSL depletion (+Genz). $n = 40$ cells per condition, representative of three independent experiments. Means ± SEM, unpaired two-sided $t$-test; ns = $P > 0.05$, ****$P < 0.0001$. Scale bars = 10 µm. Nuclei in blue.

inactive bent-closed and active ligand-bound $\alpha_5\beta_1$ integrin was thereby also observed for GL-Lect driven endocytosis.

One of the hallmarks of GL-Lect driven endocytosis is the formation of tubular, often crescent-shaped CLICs[23]. To analyze the structures via which inactive bent-closed and active ligand-bound $\alpha_5\beta_1$ integrins are internalized, respectively, mAb13 and 9EG7 were coupled to horseradish peroxidase (HRP) and incubated with cells at 37 °C for a time course between 6 and 9 min. Endocytic carriers were then analyzed by electron microscopy (EM) for the presence of these markers. Inactive bent-closed $\alpha_5\beta_1$ integrin (mAb13) was found predominantly in short tubular CLICs, and much less in vesicles (Fig. 4A), which corroborates the conclusion that a substantial fraction of this conformer enters cells by GL-Lect driven endocytosis. We also found that the dynein inhibitor ciliobrevin (CBD), which interferes with friction-driven scission that specifically operates during clathrin-independent endocytosis[76,77], strongly reduced the endocytosis of inactive bent-closed $\alpha_5\beta_1$ integrin (mAb13) (Fig. 4B). This effect was further potentiated when exogenous Gal3 was added to cells (Fig. 4B). CBD also strongly inhibited the uptake of Gal3 (Supplementary Fig. 3D) but only weakly that of transferrin (Supplementary Fig. 3E).

The presence of some inactive bent-closed $\alpha_5\beta_1$ integrin (mAb13) in vesicles (Fig. 4A) is consistent with a previous report on the contribution of clathrin-dependent endocytosis to the cellular uptake of this conformer close to focal adhesions[66], and with our finding that depletion of CHC and expression of a dominant negative mutant of the

clathrin accessory protein Eps15 (EPS15DN)[78] had both measurable effects on the cellular uptake of inactive bent-closed $\alpha_5\beta_1$ integrin (mAb13, Supplementary Figs. 1C and 3F, top) and Gal3 (Supplementary Figs. 1I and 3G, top).

Active ligand-bound $\alpha_5\beta_1$ integrin (9EG7) was almost equally distributed between CLICs and vesicles (Fig. 4A). The vesicular fraction likely reflects clathrin-dependent uptake, based on: (i) previous findings[17,18,20]; (ii) vesicle size (120 nm ±29 nm, $n = 95$)[79]; (iii) the observation of a substantial inhibition of uptake of this conformer (9EG7; Supplementary Fig. 3F, bottom) and of transferrin (Supplementary Fig. 3G, bottom) in EPS15DN expressing cells; (iv) co-immunoprecipitation with CHC (Supplementary Fig. 3H); and (v) the pronounced colocalization of endocytic tracks of active ligand-bound $\alpha_5\beta_1$ integrin (9EG7) with the clathrin pathway component adapter protein 2 (AP2), as monitored under unperturbed conditions[80,81] using lattice light sheet microscopy (Supplementary Fig. 3I–L and Supplementary Movies 3–6). The presence of a fraction of active ligand-bound $\alpha_5\beta_1$ integrin in CLICs is consistent with a previous report[21] and the observation of a slight inhibition of uptake by ciliobrevin (Fig. 4C), suggesting that Gal3-dependent and independent mechanisms exist for the generation of CLICs.

The cell-based experiments from above on interaction, co-distribution at the plasma membrane and in endosomes, molecular mechanisms of endocytic uptake and retrograde trafficking to the Golgi apparatus led to the discovery of a dichotomic imprint of Gal3

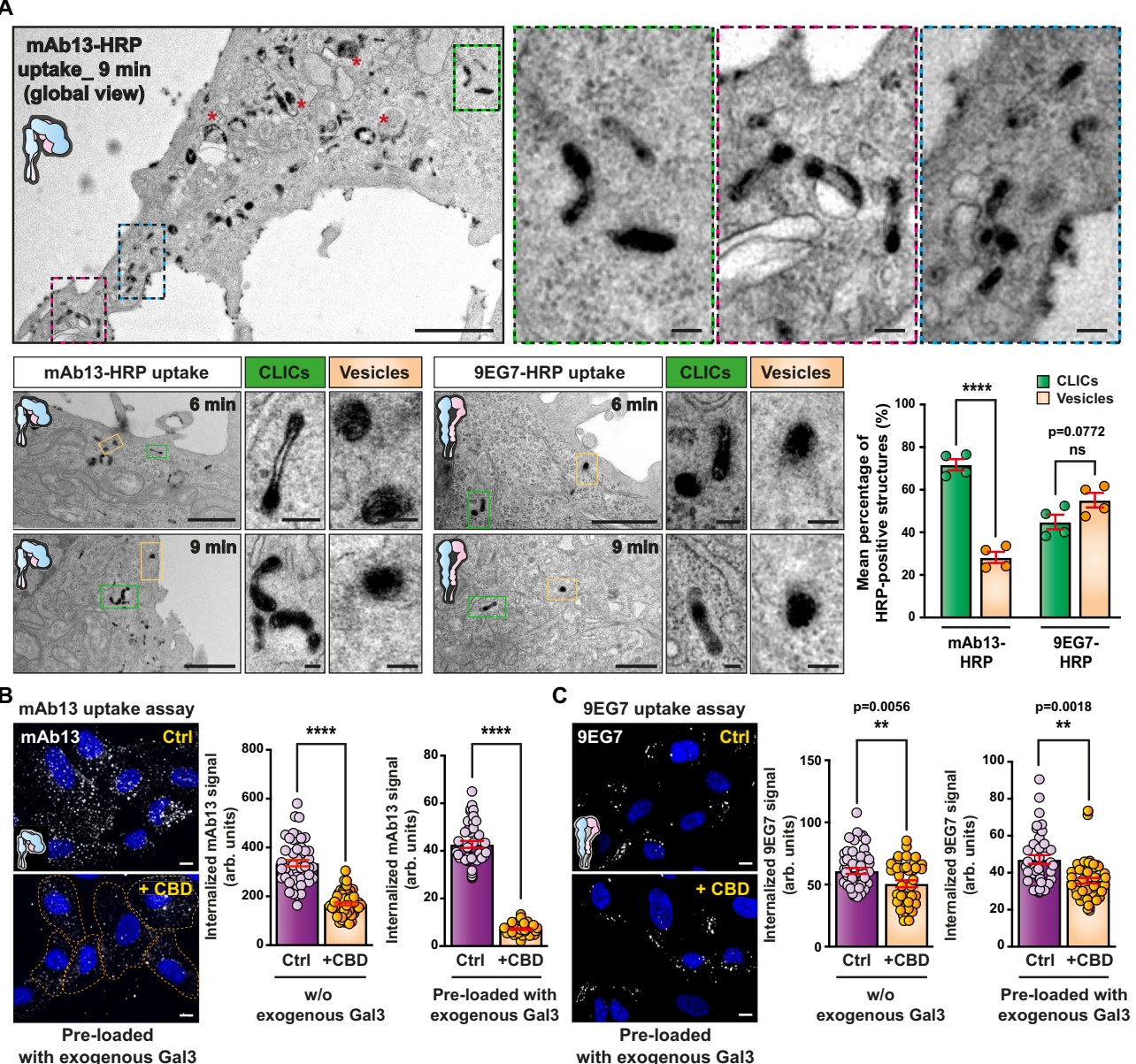

**Fig. 4 | Endocytosis of the inactive bent-closed $\alpha_5\beta_1$ integrin via CLICs.**
**A** Inactive bent-closed $\alpha_5\beta_1$ integrin accumulates in clathrin-independent carriers (CLICs). Top: EM micrographs of HeLa cells that were incubated continuously for 9 min at 37 °C with HRP-coupled mAb13 (mAb13-HRP). Boxes show zooms of areas illustrating the accumulation of mAb13-HRP in typical crescent-shaped CLIC structures. Red stars indicate larger, likely endosomal compartments. Bottom left: Same experiments as above, for both mAb13-HRP or 9EG7-HRP antibodies (6 or 9 min uptake). Green insets: Zooms of CLICs. Orange insets: Zooms of endocytic vesicles. Scale bars = 1 μm in all global views, and 100 nm in the zoomed boxes. Bottom right: Quantification of endocytic structures: $n = 97$ (for mAb13) and $n = 60$ (for 9EG7) HRP-positive structures were counted. Four independent experiments.

Means ± SEM, unpaired two-sided $t$-test; ns = $P > 0.05$, ****$P < 0.0001$. Effect of ciliobrevin D (CBD) on $\alpha_5\beta_1$ integrin endocytosis. Quantification by confocal microscopy of mAb13 (**B**) or 9EG7 (**C**) uptake after incubation for 10 min at 37 °C with RPE-1 cells in the presence (+CBD) or absence (Ctrl) of CBD. Cells were pre-loaded or not for 30 min at 4 °C with exogenous Gal3 (200 nM). Note the strong inhibitory effect of CBD on mAb13 uptake. **B** Left graph, $n = 48$ (for Ctrl) and $n = 60$ (for CBD) cells; right graph, $n = 40$ cells per condition. **C** Left graph, $n = 50$ (for Ctrl) and $n = 40$ (for CBD) cells; right graph, $n = 40$ cells per condition. Representatives of three independent experiments. Means ± SEM, unpaired $t$-test; **$P < 0.002$, ****$P < 0.0001$. Scale bars = 10 μm. Nuclei in blue.

onto specifically the inactive bent-closed conformational state of $\alpha_5\beta_1$ integrin. To understand the molecular basis for this dichotomic behavior, we set out to combine biophysical and structural techniques applied to different biochemical reconstitution systems.

### Specifically the inactive bent-closed $\alpha_5\beta_1$ integrin nucleates the formation of Gal3 oligomers

$\alpha_5\beta_1$ integrin was extracted and purified from rat liver in detergent micelles[82] (Fig. 5A and Supplementary Fig. 4A) and reconstituted into microcavity array-suspended lipid bilayers (MSLBs) in which $\alpha_5\beta_1$

integrin heterodimers could diffuse laterally (Fig. 5B), as described previously[83]. In this reductionist model of the plasma membrane, electrochemical impedance spectroscopy (EIS) allowed the label-free measurement of Gal3 binding to $\alpha_5\beta_1$ integrin through changes in membrane resistance and capacitance. At low nanomolar concentrations of Gal3, the capacitance signal $\Delta Q$ increased for membranes containing inactive bent-closed $\alpha_5\beta_1$ integrin and plateaued at Gal3 concentrations above 5 nM (Fig. 5C, circles), pointing to highly efficient Gal3 binding. In contrast, the capacitance signal remained unchanged on incubation with Gal3 concentrations as high as 37 nM on

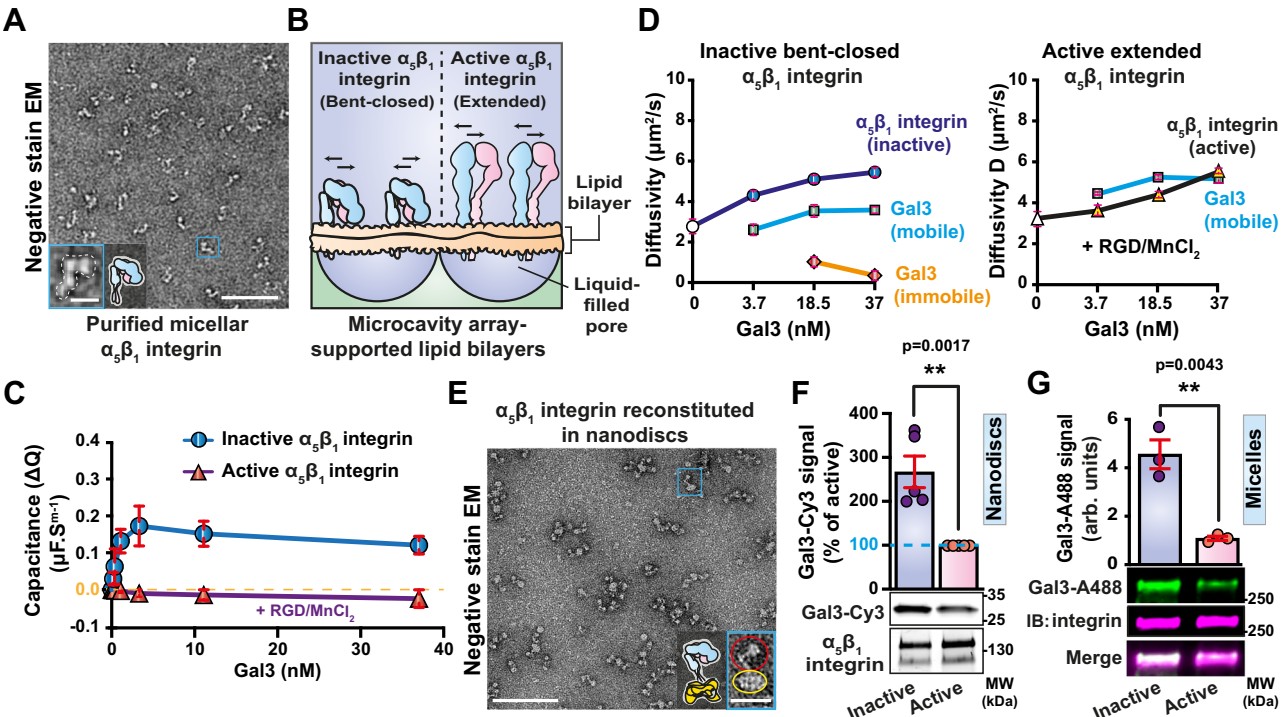

**Fig. 5 | Gal3 preferentially interacts with purified bent-closed $\alpha_5\beta_1$ integrin.**
**A** Negative stain EM micrograph of micelle-embedded $\alpha_5\beta_1$ integrin purified from rat livers. At least three independent experiments with similar observations were monitored. Scale bar = 100 nm. Inset: Zoomed view of one $\alpha_5\beta_1$ integrin. Scale bar = 10 nm. **B** Schematic of inactive bent-closed and active extended ligand-bound $\alpha_5\beta_1$ integrin embedded in microcavity array-suspended lipid bilayers (MSLBs). Arrows indicate the possibility for integrins to diffuse laterally. **C** Capacitance measurements as a function of Gal3 concentration on MSLBs containing $\alpha_5\beta_1$ integrin in the indicated conformational states. Note that capacitance increase, indicative of membrane thickness decrease, only occurs upon incubation of Gal3 with the inactive bent-closed $\alpha_5\beta_1$ integrin. $n = 3$ independent experiments, means ± SD. **D** Diffusivity measurements by FLIM. Micellar $\alpha_5\beta_1$ integrin was reconstituted in MSLBs and incubated with increasing concentrations of Gal3. In the inactive bent-closed conformational state of $\alpha_5\beta_1$ integrin, 2 diffusional behaviors were observed for Gal3, with the appearance of a slowly diffusing population at high Gal3 concentrations. We interpret this finding as an indication of the

transformation of an initial Gal3 organization (possibly Gal3 oligomers) on individual integrins (fast diffusion, "mobile") into some higher order lateral organization (lattices, slow diffusion, "immobile"). In contrast, this 2-phase diffusional behavior of Gal3 was not observed on the active extended ligand-bound conformational state of $\alpha_5\beta_1$ integrin. White circle, inactive integrin alone; white triangle, active integrin alone. **E** EM micrograph of $\alpha_5\beta_1$ integrin (red circle in zoom) reconstituted in nanodiscs (yellow circle in zoom). At least three independent experiments with similar observations were monitored. Scale bars = 100 nm for general, and 20 nm for zoomed views. **F** Pulldown of nanodisc-embedded $\alpha_5\beta_1$ integrin showing preferred interaction of Gal3-Cy3 (200 nM) with the inactive bent-closed conformational state. $n = 5$ independent experiments. Means ± SEM, unpaired two-sided $t$-test; **$P < 0.002$. **G** Pulldown experiments that reveal the preferred interaction of Gal3 (Gal3-A488, 200 nM) with inactive micelle-embedded bent-closed $\alpha_5\beta_1$ integrin. Immunoblotting (IB) documents that the same amount of total integrin was pulled down in both conditions. $n = 3$ independent experiments. Means ± SEM, unpaired two-sided $t$-test; **$P < 0.002$.

membranes containing $\alpha_5\beta_1$ integrin that had been activated with $Mn^{2+}$ and the minimal fibronectin mimicking cRGD peptide[84] (Fig. 5C, triangles).

This preferred binding of Gal3 to inactive bent-closed $\alpha_5\beta_1$ integrin reconstituted in MSLBs was confirmed by fluorescence lifetime imaging microscopy (FLIM) (Supplementary Fig. 4B). Interestingly, co-diffusivity measurements at high Gal3 concentrations revealed a biphasic behavior in the presence of inactive bent-closed $\alpha_5\beta_1$ integrin (Fig. 5D, left). This indicated the existence of both a dynamic Gal3 binding process (Gal3 mobile fraction), likely on individual $\alpha_5\beta_1$ integrin heterodimers, and the formation of an immobile fraction that may have resulted from lateral cross-linking between several $\alpha_5\beta_1$ integrins (i.e., lattices[39], Gal3 immobile fraction). Of note, for the inactive bent-closed conformer, two $\alpha_5\beta_1$ integrin-Gal3 velocity populations had also been measured by lattice light sheet microscopy on cells (Fig. 2D, E, and Supplementary Fig. 2F). In contrast, such biphasic behavior was not observed with active ligand-bound $\alpha_5\beta_1$ integrin (Fig. 5D, right). These findings establish that, also in a minimal model of the plasma membrane, inactive bent-closed $\alpha_5\beta_1$ integrin is the preferred interacting partner for Gal3.

As in the microcavity array-suspended lipid bilayer system, purified $\alpha_5\beta_1$ integrin in micelles (Fig. 5A), nanodiscs[82] (Fig. 5E), or

peptidiscs (Supplementary Fig. 4C, D) was switchable from the inactive bent-closed to the extended ligand-bound conformation upon incubation with $MnCl_2$ and cRGD peptide (Supplementary Fig. 4D). Importantly, Gal3 again interacted to a significantly greater extent with the inactive bent-closed $\alpha_5\beta_1$ integrin, as revealed by pulldown (for nanodiscs, Fig. 5F) or cross-linking (for micelles, Fig. 5G) assays.

These experiments in different model membrane systems establish the preferred interaction of Gal3 with inactive bent-closed $\alpha_5\beta_1$ integrin as an intrinsic property of the system.

To dissect the structural basis for this finding, we used single-molecule photobleaching as a first approach to assess the formation of native complexes between Gal3 and inactive bent-closed $\alpha_5\beta_1$ integrin. For this, biotinylated peptidisc-embedded integrin was complexed with fluorescently labeled Gal3. This preparation was immobilized on neutravidin-coated substrates (Supplementary Fig. 4E). Up to four photobleaching steps could be monitored (Supplementary Fig. 4F, G). In the absence of biotin from the peptidiscs (Supplementary Fig. 4G, H, no biotin) or of integrin altogether (Supplementary Fig. 4I, Gal3 alone), primarily one-step photobleaching was observed, indicative of non-specific Gal3 monomer binding to the neutravidin surface. Together, these data suggested the presence of possible Gal3 oligomers up to tetramers on $\alpha_5\beta_1$ integrin.

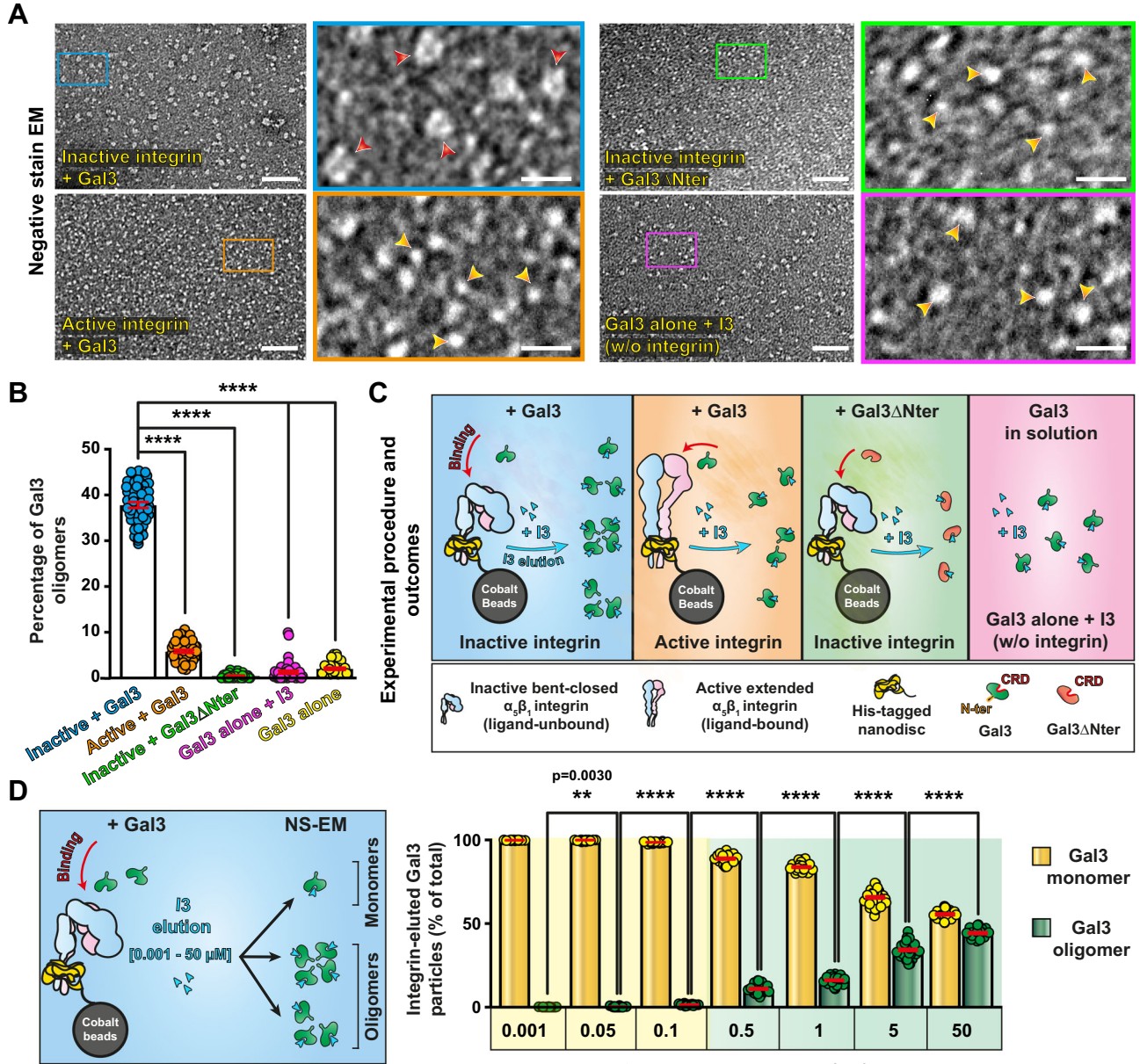

**Fig. 6 | Only inactive bent-closed $\alpha_5\beta_1$ integrin nucleates the formation of Gal3 oligomers. A** Negative stain EM micrographs of Gal3 or Gal3ΔNter that were incubated at 4 μM with nanodisc-embedded inactive or in vitro activated $\alpha_5\beta_1$ integrin, and then eluted with I3. Note the presence of defined particles with a central lumen (Gal3 oligomers, red arrowheads) only in the eluate from inactive $\alpha_5\beta_1$ integrin. These Gal3 oligomers were neither observed with Gal3ΔNter eluted from inactive $\alpha_5\beta_1$ integrin, nor for I3-incubated Gal3 in solution (yellow arrowheads show spherical monomers). At least three independent experiments with similar observations were monitored. Scale bars = 40 nm for general, and 10 nm for zoomed views. **B** Quantification of oligomer particles from (**A**) and Gal3 alone by visual picking. Inactive integrin/Gal3: $n = 50$ EM-fields with 15,707 total particles; active integrin/Gal3: $n = 44$ EM-fields with 21,096 total particles; inactive integrin/Gal3ΔNter: $n = 31$ EM-fields with 9888 total particles; Gal3 alone + I3: $n = 46$ EM-fields with 24,478 total particles; Gal3 alone: $n = 34$ EM-fields with 15,820 total

particles. Three independent experiments. Means ± SEM, one-way ANOVA; ****$P < 0.0001$. **C** Schematic of experimental outcomes from experiments described in (**A**, **B**). **D** Gal3 elution from inactive bent-closed $\alpha_5\beta_1$ integrin using increasing concentrations of I3. Left: Schematic of the experiment. Right: Conditions as in (**A**), except that Gal3 was successively eluted with the indicated I3 concentrations. Gal3 oligomers were visually quantified by negative stain EM. Note that Gal3 oligomers eluted at much higher I3 concentrations than Gal3 monomers. 0.001 μM: $n = 30$ EM-fields with 14,921 total particles; 0.05 μM: $n = 29$ EM-fields with 13,602 total particles; $n = 0.1$ μM: 28 EM-fields with 14,374 total particles; 0.5 μM: $n = 31$ EM-fields with 10,766 total particles; 1 μM: $n = 32$ EM-fields with 12,345 total particles; 5 μM: $n = 44$ EM-fields with 7069 total particles; 50 μM: $n = 28$ EM-fields with 2968 total particles. Three independent experiments. Means ± SEM, unpaired two-sided $t$-test; **$P < 0.002$, ****$P < 0.0001$.

This prediction was tested by breaking the glycan-dependent interaction between nanodisc-embedded $\alpha_5\beta_1$ integrin and Gal3 using the Gal3 inhibitor I3 (Supplementary Fig. 4J), followed by negative stain EM analysis of eluted Gal3 molecules (Fig. 6A). Interestingly, while Gal3 was monomeric in the absence of $\alpha_5\beta_1$ integrin (Fig. 6A, bottom right, purple box), about 40% had transformed into ring-shaped oligomers

when incubated with the inactive bent-closed conformer (Fig. 6A, top left, blue box), as quantified both by visual (Fig. 6B) and automated (Supplementary Fig. 4K) particles picking approaches (schematic of experimental outcomes in Fig. 6C). The nucleation of these oligomers was dependent on Gal3 concentrations (Supplementary Fig. 4L). Of note, Gal3 oligomer elution occurred at much higher I3 concentrations

than that of Gal3 monomers (Fig. 6D and Supplementary Fig. 4M), indicative of multiple-bond (avidity) oligomer interactions with bent-closed $\alpha_5\beta_1$ integrin. Gal3 oligomers disassembled over time into monomers (Supplementary Fig. 4N), thereby documenting that their formation was reversible.

In contrast, the vast majority of Gal3 eluted as small spherical (i.e., <5 nm) particles from active ligand-bound $\alpha_5\beta_1$ integrin (Fig. 6A, bottom left, orange box; Fig. 6B, C, and Supplementary Fig. 4K), likely representing monomers. As for Gal3 that had not been in contact with $\alpha_5\beta_1$ integrin (Fig. 6A, bottom right, purple box; Fig. 6B, C), the Gal3ΔNter mutant was also mostly monomeric, even when the latter was eluted from the inactive bent-closed $\alpha_5\beta_1$ integrin conformer (Fig. 6A, top right, green box; Fig. 6B, C). This correlated with the mutant's reduced efficiency for binding to the integrin (Supplementary Fig. 4O).

These experiments led to the discovery of regular ring-shaped Gal3 oligomers. In line with the dichotomy theme from the cell-based and the in vitro interaction experiments, these oligomers were nucleated only upon Gal3 binding to the inactive bent-closed conformational state of $\alpha_5\beta_1$ integrin, and their nucleation required Gal3's N-terminal oligomerization domain (Fig. 6C).

## Gal3 oligomers are also nucleated on the plasma membrane

We then investigated whether these Gal3 oligomers also formed within the complex environment of the plasma membrane glycocalyx, comprised of a great diversity of glycoproteins, including $\alpha_5\beta_1$ integrin and other integrins, proteoglycans, and GSLs. For this, lactose-washed RPE-1 cells were first incubated on ice with exogenous Gal3, and Gal3 species eluted with the competitive inhibitor I3 were then analyzed by negative stain EM (Fig. 7A). Remarkably, ring-shaped oligomer structures were again abundantly detected (Fig. 7B), similar in both shape and size to those eluted from nanodisc-reconstituted bent-closed $\alpha_5\beta_1$ integrin (Fig. 6A, top left, blue box).

By 2D classification analysis of these cell-derived Gal3 oligomers, we found that out of 14,367 particles, 5279 (37%) resembled dimers in size and shape, 5888 (41%) trimers, and 3200 (22%) tetramers (Fig. 7C). Among the tetramers, the most prevalent groups of particles had annular organization, which were the largest 2D classes among all of the classified subpopulations (722, 682, and 590 particles, respectively, Fig. 7C).

The binding properties of these cell-derived Gal3 oligomers were then assessed. At the same mass equivalents, their rebinding to cells was much more efficient than that of monomeric Gal3 (Fig. 7D, E, and Supplementary Fig. 5A). The rebinding process was inhibited by I3 (Supplementary Fig. 5B), strongly suggesting that Gal3 itself and no other contaminating cellular component was required here. This conclusion was further strengthened by the finding that qualitatively similar differences in binding efficacy over monomers were obtained with Gal3 oligomers that had been eluted from purified bent-closed $\alpha_5\beta_1$ integrin (Supplementary Fig. 5C).

The N-glycan sensitivity of binding was assessed by incubating cells with PNGase F (Fig. 7F), which strongly reduced cell surface N-glycan levels, as validated using the PhaL lectin[85] (Supplementary Fig. 5D). For monomeric Gal3, binding (Fig. 7G, left) and internalization (Fig. 7H, left) were significantly impaired on PNGase F-treated cells, whereas cell-derived Gal3 oligomers performed almost indiscriminately under both conditions (Fig. 7G, H, right). Similar results were obtained with Gal3 that had been pre-oligomerized on purified bent-closed $\alpha_5\beta_1$ integrin (Supplementary Fig. 5E, F). We hypothesized that while monomeric Gal3 needs protein N-glycans for initial recruitment to the cell surface, as indeed described before[23], pre-oligomerized Gal3 can also be recruited to the cell surface by GSLs. In agreement with this hypothesis, binding onto and internalization of preformed Gal3 oligomers into PNGase F-treated cells were significantly decreased when the cells were also GSL-depleted using Genz

(Fig. 7G, H). Of note, surface removal of N-glycans (PNGase F) or depletion of GSLs (Genz) both resulted in strongly reduced numbers of defined Gal3 oligomers scaffolded after binding of exogenous monomeric Gal3 (Fig. 7I), suggesting that at the cell surface, both types of glycoconjugates participate in oligomer nucleation.

These findings document that Gal3 oligomers have key properties as expected from the GL-Lect model, i.e., the capacity to functionally interact not only with N-glycans on glycoproteins, but also with glycans of GSLs.

## Gal3 oligomers link the head and leg pieces of inactive bent-closed $\alpha_5\beta_1$ integrin

To explore the molecular mechanism underlying the conformational state-specific dichotomic Gal3 oligomerization process, we set out to reveal the assembly state of Gal3 directly on inactive bent-closed $\alpha_5\beta_1$ integrin via cryogenic electron microscopy (cryo-EM). We first vitrified peptidisc-embedded $\alpha_5\beta_1$ integrin alone (Supplementary Fig. 6A–C) and refined the structure of the inactive bent-closed conformation to resolutions of 3.7 Å in the headpiece and 4.7 Å in the leg piece, respectively (Supplementary Fig. 6D–J). A homology model of rat $\alpha_5\beta_1$ integrin[86] was fitted individually into the density maps of the headpiece (flexible fit, Supplementary Fig. 6K, L) and leg piece (rigid-body fit, Supplementary Fig. 6M). The structure revealed a bent-closed conformation with an angle between head and leg pieces that is similar to human $\alpha_5\beta_1$ integrin[87], and larger than for other integrins[88].

Next, inactive bent-closed $\alpha_5\beta_1$ integrin was inserted into GSL-containing nanodiscs, immobilized on beads, desialylated to increase affinity for Gal3[73,74], complexed with Gal3, detached from the beads, and immediately vitrified to minimize complex dissociation and aggregation (Supplementary Fig. 7A, B). The cryo-EM density map of all particles (3.9 Å resolution, Supplementary Fig. 7C) revealed the presence of fragmented density between head and leg pieces of $\alpha_5\beta_1$ integrin, indicative of Gal3 binding in this region (Supplementary Fig. 9A, red dashed circles). To precisely position Gal3, we carried out sequential rounds of 3D variability analysis, heterogeneous refinement, and focused 3D classification. Thereby, we identified three subsets with densities at slightly different positions (Supplementary Fig. 7C, D).

Three subsets with the highest levels of density (7.4, 8.5, and 7.6 Å resolution, respectively; Fig. 8A–C and Supplementary Fig. 7D–F) comprised 6% of all particles, indicating a high level of heterogeneity. The size and shape of the density maps allowed us to fit 2, 3, or 4 Gal3 carbohydrate recognition domains (CRDs, in color; originating from PDB 1KJL) as rigid bodies between head and leg pieces of $\alpha_5\beta_1$ integrin (gray) (Fig. 8A–C, top panels). Segmentation of cryo-EM density outside the $\alpha_5\beta_1$ integrin heterodimer from these structures (Fig. 8A–C, bottom panels) revealed that additional densities were present (translucent red), which likely correspond to the N-termini of the Gal3 monomers, even if at the present resolution other interpretations cannot be excluded. These additional densities were oriented towards the center, where they overlap. The Gal3 molecules, therefore, appeared to be interconnected to form dimers (Fig. 8A, bottom panel), trimers (Fig. 8B, bottom panel) or tetramers (Fig. 8C, bottom panel).

In support of this Gal3 oligomer formation hypothesis, we also detected in our vitrified preparation further particles of smaller size compared to the aforementioned complexes, into which again 2, 3, or 4 Gal3 CRDs could be fitted (Fig. 8D–F). Of note, the Gal3 arrangements resembled the oligomeric structures found in the corresponding complexes in overall shape and size (Fig. 8A–C). Notably, the overlapping densities (translucent red) were again oriented towards the center in agreement with overlapping Gal3 N-termini, thereby compatible with the notion that these structures were Gal3 oligomers that likely had detached from the integrin during sample preparation. They were comparable in size and shape with the ones revealed by negative stain EM after elution from cells (Fig. 7B, C, and Supplementary Fig. 9B).

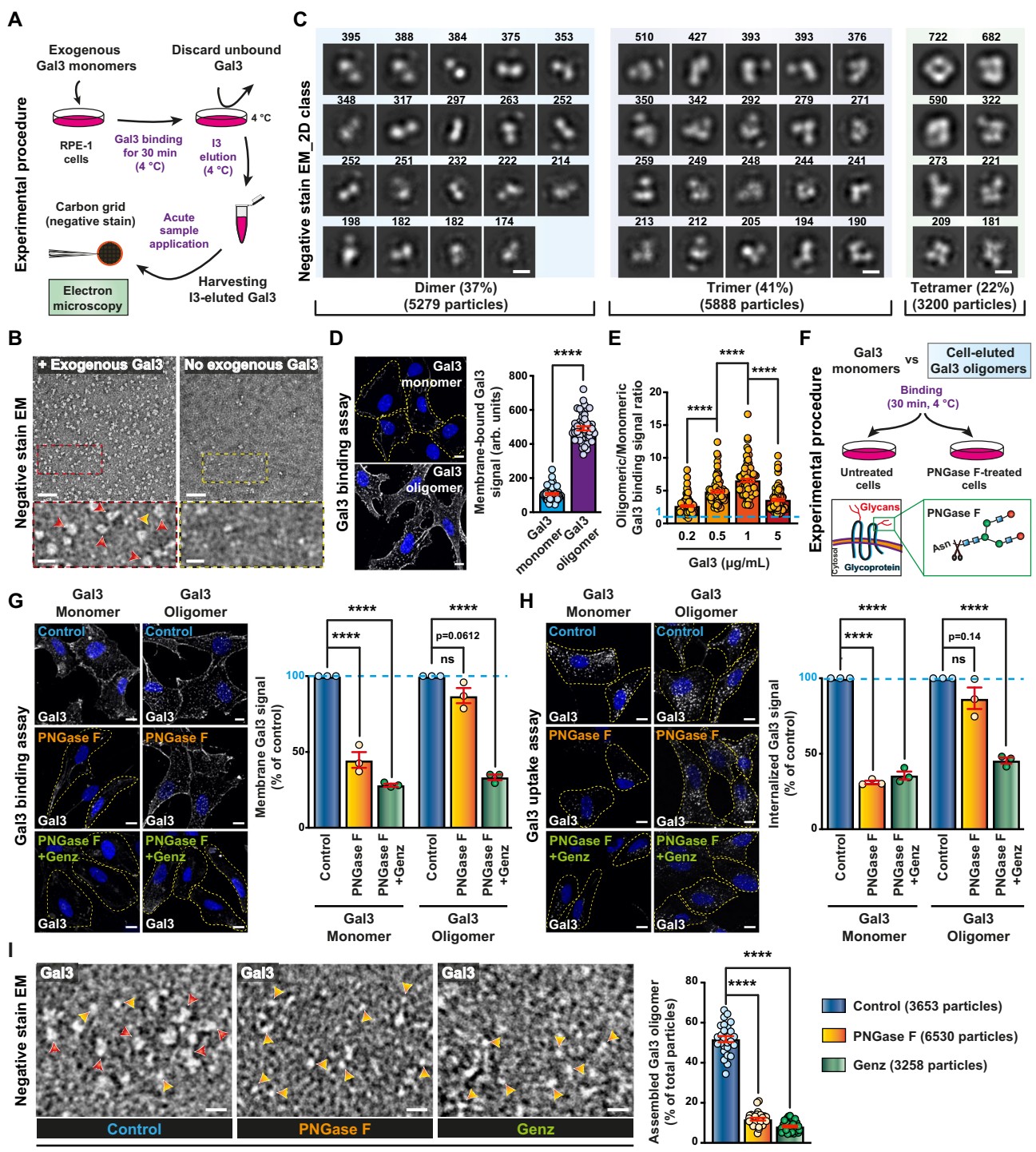

**Fig. 7 | Characterization of Gal3 oligomers from cells. A** Schematic for assembly of Gal3 (2 μM) oligomers on cells and elution with I3 (10 μM). **B** Negative stain EM micrographs of Gal3 eluted from RPE-1 cells as described in (**A**). Red and orange arrowheads indicate oligomeric and monomeric Gal3, respectively. At least three independent experiments with similar observations were conducted. Scale bars = 40 nm for general, and 10 nm for zoomed views. **C** 2D class averages of EM data as in (**B**). Particle numbers are indicated. Scale bar = 5 nm. **D** Gal3 monomer versus oligomer (eluted as in **B**) binding onto RPE-1 cells at 4 °C. Same mass equivalents (5 μg/mL) were used in both conditions. $n = 47$ cells per condition, representative of three independent experiments. Means ± SEM, unpaired two-sided $t$-test; ****$P < 0.0001$. Scale bars = 10 μm. Nuclei in blue. **E** Experiments as in (**D**) using increasing mass equivalents of Gal3 monomers or cell-eluted oligomers. $n = 100$ cells per condition, representative of three independent experiments. Means ±

SEM, one-way ANOVA; ****$P < 0.0001$. **F** Schematic for use of PNGase F to assess the role of N-glycans in Gal3 binding to cells. **G** Binding experiment at 4 °C as in (**D**) on RPE-1 cells (dashed lines) that were pre-incubated with PNGase F, alone as described in (**F**), or in combination with Genz (GSL depletion). $n = 3$ independent experiments. Means ± SEM, one-way ANOVA; ns = $P > 0.05$, ****$P < 0.0001$. Scale bars = 10 μm. Nuclei in blue. **H** Continuous incubation for 2 min at 37 °C in the same conditions as in (**G**). $n = 3$ independent experiments, means ± SEM, one-way ANOVA; ns = $P > 0.05$, ****$P < 0.0001$. Scale bars = 10 μm. Nuclei in blue. **I** Experiment as in (**A**) on RPE-1 cells that were treated either with PNGase F or Genz. Red and yellow arrowheads indicate oligomeric and monomeric Gal3, respectively. Control: $n = 27$ EM-fields, 3653 total particles; PNGase F: $n = 27$ EM-fields, 6530 total particles; Genz: $n = 28$ EM-fields, 3258 total particles. Three independent experiments. Means ± SEM, one-way ANOVA; ****$P < 0.0001$. Scale bars = 20 nm.

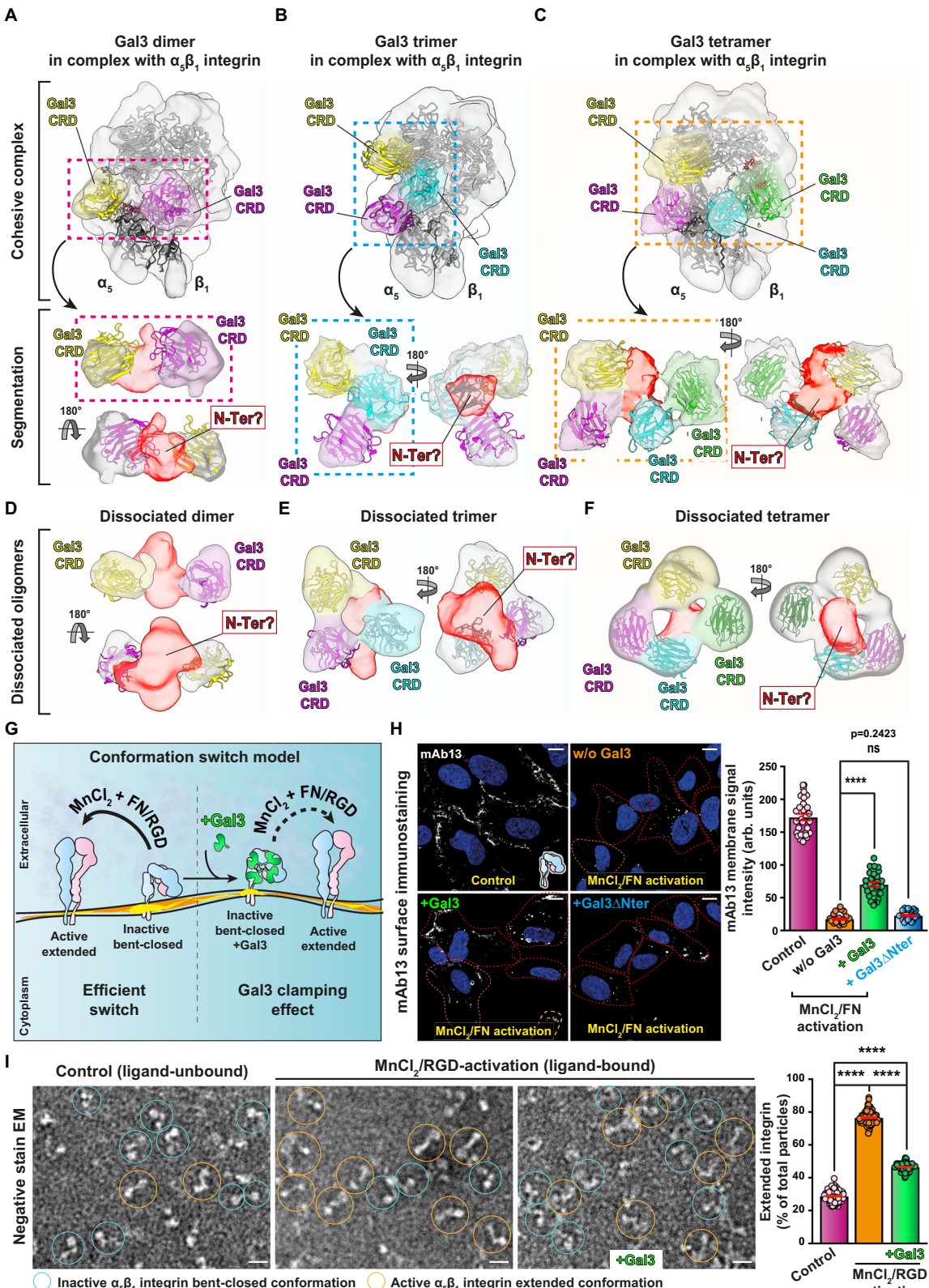

Intriguingly, in peptidisc-embedded $\alpha_5\beta_1$ integrin, a cryo-EM density map of a complex with Gal3 (13% of all particles, 6.9 Å resolution, Supplementary Fig. 8) revealed that only two CRDs could be fitted (Supplementary Fig. 9C), suggesting a requirement for membrane characteristics in complex stability.

The Gal3 oligomer models of the current study are compatible with two key functional properties of the protein. First,

oligomerization is dependent on the N-terminal domain (Fig. 6A, B) and possibly involves biomolecular condensate formation[89]. Second, the capacity to interact with several spatially separated galactoses on both protein N-glycans and GSLs is favored. Such a relative orientation of glycan-binding pockets towards the membrane surface has also been predicted by molecular dynamics simulations[90]. In contrast, a tetramer model obtained by x-ray crystallography with N-terminally

**Fig. 8 | Gal3 oligomers on inactive bent-closed α₅β₁ integrin and in isolation.** A–C Top panel: The α₅β₁ integrin model (Supplementary Fig. 6) was fitted in the cryo-EM density maps of nanodisc-embedded α₅β₁ integrin in complex with Gal3 (gray). Three different subtypes of 3D classes were identified, in which 2 (**A**), 3 (**B**), or 4 (**C**) CRDs (PDB 1KJL; in colors) could be fitted into additional densities found between the head and leg pieces of α₅β₁ integrin. Bottom panel: Segmentation of the additional densities that are connected at the center (translucent red), which suggests that Gal3 dimers (**A**), trimers (**B**), and tetramers (**C**) have formed. Cryo-EM density maps of smaller structures found in vitrified samples, as in (**A–C**), into which 2 (**D**), 3 (**E**), or 4 (**F**) Gal3 CRDs could be fitted. Bridging densities in the center were again visible (translucent red). **G** Schematic of hypothesis on Gal3-mediated clamping, as tested in (**H, I**). **H** RPE-1 cells were incubated at 4 °C with either wild-type Gal3 or oligomerization-deficient Gal3ΔNter (200 nM) prior to activation with 1 mM MnCl₂ and 5 μg/mL soluble fibronectin (FN). Immunolabeling with the mAb13

antibody was quantified. $n = 30$ cells per condition, representative of three independent experiments. Means ± SEM, one-way ANOVA; ns = $P > 0.05$; ****$P < 0.0001$. Scale bars = 10 μm. Nuclei in blue. **I** Peptidisc-embedded α₅β₁ integrin immobilized on beads was incubated with or without 4 μM Gal3 prior to in vitro activation with 5 mM MnCl₂ and 100 μM cRGD. α₅β₁ integrin particles were analyzed by negative stain EM. The propensity of α₅β₁ integrin to be shifted from the bent-closed conformation (blue circles) to the active extended ligand-bound state in MnCl₂/cRGD (orange circles) was significantly reduced by Gal3. Particles with angles between head and leg pieces above 90° were considered as an extended conformation. Control (ligand-unbound): $n = 102$ EM-fields with 6047 total particles; MnCl₂/cRGD activation (ligand-bound): $n = 89$ EM-fields with 8304 total particles; MnCl₂/cRGD activation with Gal3 pre-loading: $n = 85$ EM-fields with 7704 total particles, compiled from three independent experiments. Means ± SEM, one-way ANOVA; ****$P < 0.0001$. Scale bars = 10 nm.

truncated versions of Gal3 assembled under non-natural conditions appears too compact to fit the EM density of the current study, and orients the glycan-binding pockets inwards, which is incompatible with simultaneous interaction with cargo and GSL glycans[91] (Supplementary Fig. 9D).

Our data thereby represent the first 3D models of Gal3 oligomers that were made of full-length protein, and that were nucleated on glycoproteins from physiological sources.

## Gal3 clamps the inactive bent-closed conformational state of α₅β₁ integrin

Our structural data pointed to the intriguing possibility that Gal3 oligomers might physically bridge membrane-proximal (leg piece) and distal (headpiece) parts of α₅β₁ integrin that face each other in the inactive bent-closed conformer (Fig. 8A–C). α₅β₁ integrin in its bent-closed state would thereby be stabilized by a Gal3-mediated clamping effect. To address this possibility, the propensity of α₅β₁ integrin to undergo the MnCl₂ and cRGD/fibronectin-triggered switch from the bent-closed to the extended ligand-bound conformational state was assessed upon pre-binding of Gal3, either on cells or on individual α₅β₁ integrin (Fig. 8G).

Incubation of cells with MnCl₂/fibronectin led to a strong increase of extended conformation-specific labeling (9EG7) (Supplementary Fig. 9E), and a concomitant decrease of inactive bent-closed conformation-specific labeling (mAb13) (Supplementary Fig. 9F). Pre-incubation of cells with Gal3 reduced this decrease of mAb13 labeling under MnCl₂/fibronectin incubation conditions (Fig. 8H), indicating that α₅β₁ integrin activation was inhibited. This was not observed with the Gal3ΔNter mutant (Fig. 8H), which reinforces the notion that oligomerization is required for this integrin clamping effect.

On purified α₅β₁ integrin, the conformational switch was directly monitored by negative stain EM (Fig. 8I) or by electrophoretic mobility (Supplementary Fig. 9G). In both cases, Gal3 again reduced the MnCl₂/cRGD-induced switch to the extended conformational state. Of note, a slight decrease in the angle between head and leg pieces occurred when clearly visible Gal3 densities were present on α₅β₁ peptidisc-embedded integrin (Supplementary Fig. 9H), which is also in support of a clamping effect.

To identify the glycans that are involved in the functional recognition of the bent-closed conformational state of α₅β₁ integrin by oligomerization competent Gal3, we combined cross-linking proteomics, molecular modeling, and site-directed mutagenesis in subsequent experiments.

## Specifically the inactive bent-closed α₅β₁ integrin requires defined glycans for endocytic uptake

In the following, we use "r" or "h" tags in front of each N-glycosylation site to indicate their position in the α₅β₁ integrin of rat or human origins, respectively.

Cross-linking mass spectrometry revealed the proximity between position K227 in the CRD of Gal3 with positions in the membrane distal

and membrane-proximal calf regions of the leg piece of α₅ integrin (Supplementary Fig. 10A, B). Several N-glycans at sites close to cross-linking positions, i.e., rN642, rN761, rN773, rN822, and rN917/rN918 of the α₅ chain, are complex-type multi-antennae structures that are of galectin-binding competent nature, as determined by site-specific glycoproteomics on the same α₅β₁ integrin preparation from rat liver as used in the current study[86]. Using GlycoSHIELD[92], possible conformations of these N-glycans were projected onto the α₅β₁ integrin model (Supplementary Fig. 10C, D).

Notably, the glycan at α₅-rN918 protrudes towards the head/leg interspace (Supplementary Fig. 10E), and its terminal galactose residues allowed for the fitting of a Gal3 CRD into the additional cryo-EM densities of dimeric, trimeric, and tetrameric Gal3 oligomers complexed with nanodisc-embedded α₅β₁ integrin (Fig. 9A–C). Interestingly, in the Gal3-integrin complexes that we have identified in peptidiscs, the glycan at position α₅-rN918 also appeared to be highly compatible for the fitting of one Gal3 CRD (Supplementary Fig. 10F, left panel). In this situation, a second Gal3 CRD could be placed into this additional density for interaction with a galactose residue on the α₅-rN773 glycan on the membrane-distal calf (Supplementary Fig. 10F, left panel, bottom, zoom with purple dashed line).

From these structural and cross-linking proteomics data, we deduced the N-glycans that might be involved in Gal3 binding and oligomerization. To assess their functional relevance, the corresponding glycosylation sites were mutated in human α₅ integrin (Supplementary Fig. 10G, H), and expressed together with wildtype human β₁ integrin in mouse kidney fibroblasts with a double knockout for α₅β₁ integrin (dKO-MKF) (Supplementary Fig. 10I). These cells endogenously express Gal3 (Supplementary Fig. 1A), and we demonstrated their capacity to assemble defined Gal3 oligomers even in the absence of endogenous α₅β₁ integrin (Supplementary Fig. 10J). As for RPE-1 cells (Fig. 7G, and Supplementary Fig. 5D, F), we validated the efficiency of PNGase F treatment in dKO-MKF cells (Supplementary Fig. 10K–M) and demonstrated that both N-glycans and GSLs were required to scaffold such defined Gal3 oligomers (Supplementary Fig. 10N), which reproduced our findings on RPE-1 cells (Fig. 7I).

With this tool in hand, we first investigated glycosylation sites in the membrane-proximal calf region of α₅ integrin by generating the ΔMP mutant (Supplementary Fig. 10H). α₅β₁ integrin cell surface levels were similar (mAb13) or even superior (9EG7) in ΔMP-expressing cells, when compared to wild-type α₅β₁ integrin-expressing cells (Supplementary Fig. 10O). This documents efficient ΔMP-mutant α₅β₁ integrin export from the endoplasmic reticulum and trafficking to the plasma membrane. The increased levels of active ligand-bound α₅β₁ integrin at the cell surface of ΔMP-expressing cells may explain the observed increase in cell area at short (30–180 min) (Supplementary Fig. 11A) or long (24 h) (Supplementary Fig. 11B) times after plating, with the appearance of thick fibrillar-shaped 9EG7 and vinculin-positive focal adhesions (Supplementary Fig. 11C). In contrast, wild-type α₅β₁ integrin-expressing cells mainly exhibited discrete adhesion sites,

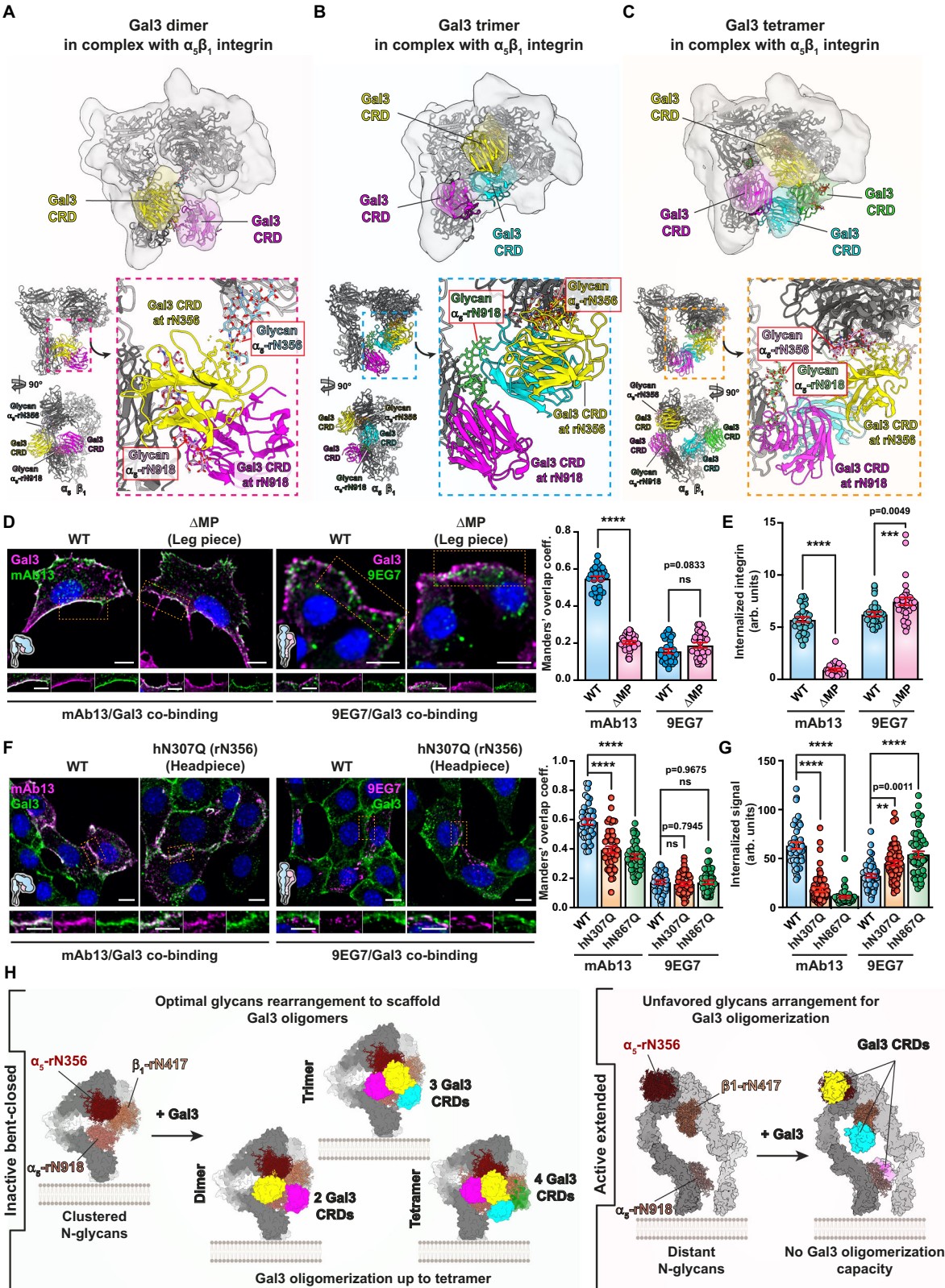

likely corresponding to nascent focal adhesion with dynamic assembly and disassembly rates, prone to sustain efficient cell migration[93,94].

Despite efficient localization of ΔMP-mutant $\alpha_5\beta_1$ integrin to the cell surface, the overlap of mAb13 (inactive bent-closed $\alpha_5\beta_1$ integrin) with Gal3 dropped to similarly low levels on these ΔMP (Fig. 9D) as observed with the oligomerization-deficient Gal3ΔNter mutant on wild-type $\alpha_5\beta_1$ integrin-expressing cells (Supplementary Fig. 12A). This

result suggests that the removal of the membrane-proximal glycosylation sites on the $\alpha_5$ chain diminished the capacity of bent-closed $\alpha_5\beta_1$ integrin to bind Gal3. Consistently, we found that the weak overlap of Gal3 with active ligand-bound $\alpha_5\beta_1$ integrin (9EG7), which does not nucleate Gal3 oligomers (Fig. 6A, B), was not reduced any further on ΔMP-expressing cells (Fig. 9D). Furthermore, the endocytic uptake of the inactive bent-closed conformational state (mAb13) of $\alpha_5\beta_1$ integrin

**Fig. 9 | Glycans for the functional recognition of bent-closed α$_5$β$_1$ integrin by Gal3. A–C** Top panels: Side views of cryo-EM density maps of Gal3 dimers, trimers, and tetramers in complex with α$_5$β$_1$ integrin embedded in a nanodisc. Lower panels: GlycoSHIELD glycan conformations at positions α$_5$-rN356 and α$_5$-rN918 that fit the best with bound Gal3 CRDs (PDB 1KJL) are highlighted (dashed line insets). **D** Colocalization of Gal3 with mAb13 or 9EG7 antibodies after binding at 4 °C to α$_5$β$_1$ integrin-deficient dKO-MKF cells exogenously expressing heterodimers of wild-type or ΔMP (leg piece) mutant human α$_5$ integrin with wild-type human β$_1$ integrin. $n = 33$ (for mAb13) and $n = 30$ (for 9EG7) cells, representative of three independent experiments. Means ± SEM, unpaired two-sided $t$-test; ns = $P > 0.5$, ****$P < 0.0001$. Scale bars = 10 μm and 5 μm for zoomed insets. Nuclei in blue. **E** Continuous incubation for 10 min at 37 °C of mAb13 or 9EG7 with dKO-MKF cells under α$_5$β$_1$ integrin expression conditions as in (**D**). Non-internalized antibodies

were removed. $n = 30$ cells per condition, representative of three independent experiments. Means ± SEM, unpaired two-sided $t$-test; ***$P < 0.0002$, ****$P < 0.0001$. **F** Colocalization experiments as in (**D**), in which positions α$_5$-hN307 (headpiece) and α$_5$-hN867 (leg piece) of human α$_5$ integrin were mutated and expressed with wild-type human β$_1$ integrin. $n = 40$ cells per condition, representative of three independent experiments. Means ± SEM, one-way ANOVA; ns = $P > 0.5$, ****$P < 0.0001$. Scale bars = 10 μm and 5 μm for zoomed insets. Nuclei in blue. **G** Internalization experiments as in (**E**) on dKO-MKF cells expressing the heterodimer of wildtype, α$_5$-hN307Q, or α$_5$-hN867Q α$_5$ integrins with wildtype human β$_1$ integrin. $n = 40$ (for mAb13-WT), 58 (for mAb13-hN307Q), 44 (for mAb13-hN867Q), 54 (for 9EG7-WT), 52 (for 9EG7-hN307Q) and 55 (for 9EG7-hN867Q) cells, representative of three independent experiments. Means ± SEM, unpaired two-sided $t$-test; **$P < 0.002$, ****$P < 0.0001$. **H** Working model. See text for details.

ΔMP mutant was also much reduced, when compared to that of wildtype α$_5$β$_1$ integrin, while no difference was observed for the active ligand-bound conformational state (9EG7) (Fig. 9E). The fact that the cell surface levels of the latter, and not the former, increased in ΔMP mutant expressing cells (Supplementary Fig. 10O) suggested that endocytically constrained inactive bent-closed ΔMP mutant may have been shifted into the active state.

These results again demonstrate a dichotomic relationship in which the membrane proximal glycosylation sites on the α$_5$ chain are critical for efficient colocalization with Gal3 and endocytic uptake of specifically the inactive bent-closed conformational state of α$_5$β$_1$ integrin. We then addressed the role of glycosylation sites in the headpiece.

From the glycans of the α$_5$ chain headpiece that point towards the additional cryo-EM density that we ascribed to Gal3 (Fig. 9A–C), the one at position rN356 (hN307 in human α$_5$, Supplementary Fig. 10G) is a Gal3-binding competent complex-type glycan[86]. Based on GlycoSHIELD modeling, it was indeed possible to position another Gal3 CRD for interaction with a galactose residue on the α$_5$-rN356 (α$_5$-hN307) glycan. With α$_5$-rN918 (α$_5$-hN867), α$_5$-rN356 (α$_5$-hN307) constitutes the core glycan pair, which is common to Gal3 dimers, trimers, and tetramers that all bridge the head and leg pieces of bent-closed α$_5$β$_1$ integrin (Fig. 9A–C, and Supplementary Figs. 9C and 10F, left panel, top, blue dashed line zoom). The corresponding human α$_5$ chain glycosylation site mutant hN307Q, when co-expressed with wildtype human β$_1$ integrin in dKO-MKF-cells, was again efficiently localized at the cell surface (Supplementary Fig. 12B), with increased levels of active ligand-bound α$_5$β$_1$ integrin (9EG7) that may explain the increase in cell area (Supplementary Fig. 11A, B) and thickening of focal adhesions in α$_5$-hN307Q cells (Supplementary Fig. 11C). Yet, for the inactive bent-closed conformational state (mAb13), the overlap with Gal3 was significantly reduced (Fig. 9F), and mAb13 uptake was largely inhibited (Fig. 9G), similar to ΔMP (Fig. 9D, E) and α$_5$-hN867Q mutants (α$_5$-rN918Q; Fig. 9F, G). In contrast, for the active ligand-bound conformational state (9EG7), overlap with Gal3 (Fig. 9F) and endocytosis (Fig. 9G) were similar between α$_5$-hN307Q or α$_5$-hN867Q mutant expressing cells and wild-type α$_5$β$_1$ integrin-expressing cells.

Although thigh-located tri and tetra-antennary glycan carrying glycosylation sites α$_5$-rN642 (α$_5$-hN593) and α$_5$-rN822 (α$_5$-hN773)[86] were robustly identified in our cross-linking proteomics (Supplementary Fig. 10A, B), their mutation (α$_5$-hN593Q or α$_5$-hN773Q) neither affected the surface levels (Supplementary Fig. 12C) nor the endocytic uptake (Supplementary Fig. 12D) of the inactive bent-closed conformational state of α$_5$β$_1$ integrin (mAb13). Very clearly, not all glycans had the same capacity to impact GL-Lect-driven endocytosis.

Interestingly, in the complex between dimeric Gal3 and nanodisc-embedded α$_5$β$_1$ integrin (Fig. 9A), the existence of an alternative to the α$_5$-rN356/α$_5$-rN918 glycan pair can be proposed, in which the glycan at rN417 of β$_1$ integrin also fits with one Gal3-ascribed electron density, in addition to glycan α$_5$-rN356 (Supplementary Fig. 12E). Gal3 binding to glycan β$_1$-rN417 is also compatible with additional densities within the trimer and tetramer complexes (Supplementary Fig. 12F), and cross-

linking mass spectrometry indeed suggested proximity of β$_1$-rN417 with cross-linked lysines in α$_5$β$_1$ integrin and Gal3 (Supplementary Fig. 10B).

These mutational data consolidate the dichotomy notion between inactive bent-closed and active extended ligand-bound conformational states of α$_5$β$_1$ integrin as to their interaction with Gal3. Specifically, it appears that only on bent-closed α$_5$β$_1$ integrin N-glycans from head and leg pieces are arranged such that Gal3 oligomerization can be nucleated (Fig. 9H, left). In contrast, in extended ligand-bound α$_5$β$_1$ integrin, these glycans are too distant from each other to enable such oligomerization (Fig. 9H, right).

Also, α$_V$ and α$_3$ integrins were found to be strong Gal3 interactors and cargoes of the retrograde route (Supplementary Fig. 12G). In both cases, N-glycosylation sites are present at equivalent positions as the key α$_5$ integrin sites α$_5$-rN356 and α$_5$-rN918 (Supplementary Fig. 12H). In contrast, for α integrins that were weakly enriched in our retrograde and Gal3 interaction proteomics lists, such as α$_2$ and α$_6$ integrins (Supplementary Fig. 12G), no strict structural conservation could be detected for the α$_5$-rN356 and α$_5$-rN918 sites (Supplementary Fig. 12H). The N-glycan signature that we have identified here as a trigger for Gal3 oligomerization may therefore apply beyond α$_5$β$_1$ integrin.

## Discussion

Based on extensive structural and functional data, we demonstrate that the active-inactive conformational state switch of α$_5$β$_1$ integrin takes advantage of rearranging the spatial positioning of N-glycans to nucleate Gal3 oligomers and direct the integrin's endocytic fate. Thus, only the non-ligand-bound bent-closed conformational state positions α$_5$β$_1$ integrin N-glycans for Gal3 oligomerization and ultimately GL-Lect driven endocytosis (Fig. 9H, left). Here, we focused on α$_5$β$_1$ integrin, but other known Gal3 binding partners, such as other integrins, CD44[95,96], and epidermal growth factor receptor[97], also undergo critical conformational changes that potentially result in similar spatial rearrangement of their N-glycans. We propose spatial rearrangement of N-glycans on proteins undergoing major conformational changes during ligand-binding (or switching between activity states) as a more general mechanism for directing interactions and oligomeric assemblies of galectins and determine their specific endocytic fates. We think that our structural determination of the first example of Gal3 nucleation on spatially arranged N-glycans in the inactive bent-closed state of α$_5$β$_1$ integrin provides a guide for the N-glycan arrangement pattern needed to nucleate Gal3 oligomers and ultimately for predicting glycoproteins that are capable of doing so.

The concept of obtaining biological specificity in interactions with otherwise commonly expressed glycan structures through recognition of assemblies of multiple glycans (also referred to as clustered saccharide patches) by glycan-binding receptors was originally proposed by Ajit Varki[98,99]. The Gal3 oligomer nucleation competent glycan arrangement on bent-closed α$_5$β$_1$ integrin (Fig. 9H, left) befits this concept and extends it by adding the dynamic switching of the glycan landscape through functional states of a glycoprotein. Molecular

insights into recognition of clustered saccharide patches are only emerging now, with the example of a bacterial mucin-binding module that interacts with a distinct cluster of O-glycans[100], and a bacterial toxin requiring multiple N-glycans for binding[101]. Our study provides an unprecedented structural insight into the recognition of multiple N-glycans and constitutes a pioneering example of dynamic changes in the spatial arrangements of glycan-binding motifs. Most importantly, our study clearly illustrates how one of the most common structural elements in human glycans, the LacNAc disaccharide unit, in fact can serve as the ligand for directing specific biological functions through the context of presentation, as previously proposed[98,99]. Our findings are likely to have a great impact not only on other members of the galectin family, but also on the many other types of endogenous glycan-binding proteins, such as siglecs and selectins, that seemingly bind common glycans and for which the natural endogenous ligands have remained elusive[102–104].

Our work sheds new light on the nature and shape of Gal3 oligomers, which may have wider relevance to the large family of galectins and other glycan-binding proteins. Based on biochemical evidence, Gal3 oligomers have been proposed to exist as ill-defined pentamers[55], as higher order assemblies[54], or as tetramers with inconsistent features made from N-terminally truncated protein[91]. Here, using tissue-derived reconstituted $\alpha_5\beta_1$ integrin or the surface of living cells for assembly, we demonstrate that full-length Gal3 can form well-defined dimers, trimers, and tetramers. Pentamers were not found, which could either mean that they are absent altogether, have very low abundance, and/or are very unstable.

According to our current model of the Gal3 tetramer, glycan-binding sites point in the same outward direction. Such molecular architecture is also found for GSL-binding subunits of bacterial Shiga and cholera toxins and the VP1 protein of simian virus 40, which, like Gal3, drive GSL-dependent narrow membrane bending and the biogenesis of tubular endocytic pits from which CLICs are formed[105–107]. These elements may represent the structural signature of GL-Lect driven endocytic processes. How the molecular organization of Gal3 oligomers on inactive bent-closed $\alpha_5\beta_1$ integrin, as revealed here, reaches onto the membrane to drive GSL-dependent membrane deformation remains to be established in future studies.

While our study reveals the mechanism of Gal3 oligomer assembly at the cell surface, it remains to be established how oligomer disassembly is operated. Our observation that both Gal3 and bent-closed $\alpha_5\beta_1$ integrin are found together in the same endocytic structures and are transported to the Golgi apparatus points to the possibility that Golgi-specific functions may contribute to the disassembly process, e.g., acidification[108] and resialylation on N-glycans[38,109]. This would then free $\alpha_5\beta_1$ integrin and reset the protein for de novo activation after its polarized secretion to the leading edge[9,11,12,56,110].

Our study identifies the bent-closed conformational state of $\alpha_5\beta_1$ integrin as a nucleator of Gal3 oligomerization to selectively drive endocytosis. This finding provides a hitherto undescribed scenario for the recognition of patterns of glycans on glycoproteins and the importance of considering the rearrangement of glycans for the acute regulation of functions and dynamics at the cell surface. Elsewhere, we propose a complementary mechanism for galectin-driven endocytosis of membrane glycoproteins, including integrins, in which acute desialylation at the cell surface by growth factor signaling exposes high-affinity ligands for galectin binding[38]. These two mechanisms, the conformational glycoswitch (this study) and the desialylation glycoswitch[38], are indeed complementary and not mutually exclusive as they may function in a staged manner, i.e., growth factor-induced N-glycan desialylation (desialylation glycoswitch) followed by conformational state-specific rearrangements in the N-glycan landscape to drive nucleation of galectin oligomers (conformational glycoswitch). Sialylation would thereby set the fraction of inactive bent-closed $\alpha_5\beta_1$ integrin that is available for nucleating Gal3 oligomers.

In conclusion, our work highlights a dynamic nature of the glycan landscape at the cell surface, which calls for profound rethinking of interactions with glycans and the roles and functions of the glycans on membrane glycoproteins.

## Methods

### Cells and tissues

HeLa cells, HeLa cells stably expressing the TGN-localized GalT-GFP-SNAP fusion protein[111], RPE-1 cells, genome-edited RPE-1 cells stably expressing AP2-mTag-GFP (see below), $\alpha_5\beta_1$ integrin double KO mouse kidney fibroblast (dKO-MKF, kindly provided by Reinhard Fässler). Rat livers (Charles Rivers), *E. coli* strain Rossetta2-pLysS (Novagen).

### Reagents

WGA-agarose column (Sigma-Aldrich, Ref. 61768-5 mL), FNIII$_{9-10}$-agarose column (GE Healthcare), cRGD (CliniSciences, Ref. A8164), soluble fibronectin (Sigma, Ref. F0895-2mg), recombinant FNIII 9-10 fibronectin fragment[82], NHS-HiTrap (GE Healthcare, Ref. 17071701), His-Pur™ Cobalt Resin (Thermofisher, Ref. 89965), protein G-sepharose beads (Sigma, Ref. P3296), GFP-Trap MA beads (Chromotek, Ref. gtma-20), Bio-Beads™ SM2 adsorbent media (Biorad, Ref. 152-3920), 300-400 mesh carbon-coated copper grids for electron microscopy (Delta Microscopy, Ref. DG400-Cu), Quantifoil Cu 300 mesh QF2/1 grids for cryo-EM, protease inhibitors (Sigma-Aldrich, Ref. P8849), chicken egg phosphatidylcholine (ePC, Avanti Polar, Ref. 840051C), Brain PS (Avanti Polar, Ref. 840032P), brain porcine gangliosides total extract (GSLs) (Avanti Polar 860053P), Transferrin-Alexa546 (Tf-A546) (Invitrogen, Ref. T23364), recombinant purified Gal3, Gal3-His, Gal3ΔNter[23], Ingenio® electroporation buffer (Mirus, Ref. MIR 50111), HiPerFect transfection reagent (Qiagen), siCHC (Qiagen, Refs. SI00299880S1; SI00299873S1), hamster anti-rat $\alpha_5$ and $\beta_1$ integrin primary antibody (BioLegend, Ref. 103902/102202), HRP-coupled secondary anti-hamster antibody, mAb13 antibody (BD Bioscience, Ref. 552828), 9EG7 antibody (BD Bioscience, Ref. 553715), mAb16 and SNAKA51 antibodies (kindly provided by Patrick Caswell), anti-SNAP-tag antibody (New England Biolabs, Ref. P9310S), anti-CHC antibody (BD Bioscience, Ref. 610500), mouse anti-vinculin antibody (Sigma, Ref. V9131), anti-α tubulin antibody (BD Bioscience, Ref. T5168), anti-vps35 antibody (kind gift from Juan Bonifacino), anti-vps26 antibody (Abcam, ab23892), NHS-ATTO-488 (ATTO-TEC, Ref. AD488-31), NHS-Alexa647 (Invitrogen, Ref. A20006), secondary anti-mouse-HRP (Beckman Coulter, Ref. 715-035-151), secondary anti-rabbit-HRP (Beckman Coulter, Ref. 711-035-152), secondary anti-rat-HRP (Beckman Coulter, Ref. 712-035-153), secondary anti-rat Cy3 (Beckman Coulter, Ref. 712-166-153), secondary anti-mouse Alexa488 (Molecular probes Invitrogen, Ref. A-21202), HRP-NHS (AAT Bioquest, Ref. 11025), sialidase (neuraminidase from *Arthrobacter ureafaciens*, Roche Ref. 10269611001), glycerol-free PNGase F (New England Biolabs, Ref. P0705L), Alexa488-conjugated PHA-L lectin (Thermofisher, ref. L11270), Alexa488-conjugated SiaFind Pan-specific Lectenz® lectin (Lectenz, Ref. SK0501F), BG-GLA-NHS (New England Biolabs, Ref. S0151S), ciliobrevin D (CBD) (Merck, Ref. 250401), Genz-123346 (Sigma, Ref. 5382850001), Gal3 inhibitor compound GB0149-03 (Galecto Biotech; 1,1′-sulfanediyl-bis-{3-deoxy-3-[4-(butylaminocarbonyl)−1*H*-1,2,3-triazol-1-yl]-β-D galactopyranoside}, referred to as I3), β-D-lactose (Sigma, Ref. L3750), NHS-Cy3 (GE, PA23001), Cy3 monoreactive succinimidyl ester (Cytivia, Ref. GEPA23001), BS3-d0/BS3-d4 (bis(sulfosuccinimidyl)suberate-d0/d4, Thermoscientific, Ref. 21590 and 21595), DSBU (Disuccinimidyl Dibutyric Urea, Thermoscientific, Ref. A35459), SNAP-Cell® Block (New England Biolabs Ref. S9106S), glutaraldehyde, protein ladder (Thermoscientific, Ref. 26619), Triton X-100™ (Anatrace, Ref. T1001-500 mL), DAB (Sigma-Aldrich, Ref. D8001), ascorbic acid, Nonidet P-40 (NP40, Sigma, Ref. 21-3277), MSP1D1 protein scaffold (MSP1D1-His, Cube Biotech, Ref. 26112), peptidiscs (MSP1D1-derived scaffold peptidisc-His/Biotin, Peptidisc

Lab), dialysis cassettes 10 kDa molecular weight cut-off (Thermo Scientific), 100 kDa molecular weight cut-off concentrator (GE Healthcare), Vivaspin 500 (Merck, Ref. Z614092-25EA), PD10 desalting column (GE, Ref. 17-0851-01), Zeba spin column 7 kDa cut-off (Thermo Fisher, Ref. 89882), 4-15% Stain-Free™ pre-casted polyacrylamide gels (BioRad, Ref. 4561084), ECL reagent, 5 mm cover glass (Electron Microscopy Sciences, Ref. 72195-05); clathrin (CHC) siRNAs Hs CLTC 9 and 10 (Qiagen, Ref. 1027417/SI00299880S1 and 1027417/SI00299873S1); Gal3-CFP mammalian expression vector was obtained by sub-cloning human Gal3 sequence in peCFP-C1/2/3 vector (Clontech); cDNA sequences of the different N-glycosylation mutants of α5 integrin were synthetized by GenScript and cloned in pCI-neo mammalian expression vector.

## Media and buffers
DMEM high glucose (Thermofisher, Ref. 41965039), DMEM-F12 Gibco (Thermofisher, Ref. 11320033), 1 M HEPES pH 7-7.4 (Sigma-Aldrich, Ref. H0887), EDTA pH 8, PBS++ (PBS supplemented with 1 mM MgCl$_2$ and 0.5 mM CaCl$_2$, pH 7.4), HEPES buffer (20 mM HEPES, 150 mM NaCl), Hepes-Tx-100 buffer (HEPES buffer supplemented with 0.2% (v/v) Triton X-100), β-D-lactose solution (150 mM, iso-osmolarized), I3 solution (10 µM or 50 µM), TNE buffer (10 mM Tris, 150 mM NaCl and 5 mM EDTA), low-salt TNE (TNE with 50 mM NaCl), lysis buffer (TNE with 1% NP40, supplemented with proteases inhibitor cocktail), low-salt lysis buffer (lysis buffer with low-salt TNE), acid wash buffer (glycine 0.5 M, pH 2.2). 4% PFA solution (Electron Microscopy Sciences, Ref. 1570), 3x non-reducing SDS sample buffer (2 M Tris/HCl, pH 6.8, 20% SDS, 30% glycerol, 0.03% phenol red), 3x non-reducing low SDS sample buffer (2 M Tris/HCl, pH 6.8, 2% SDS, 30% glycerol, 0.03% phenol red), BSA-saponin buffer (0.5% saponin, 2% BSA, in PBS), HEPES buffer (20 mM HEPES, 150 mM NaCl), HEPES/Tx-100 buffer (HEPES buffer supplemented with 0.2% (v/v) Triton X-100).

## Equipment
Transmission electron microscope (TEM) 80 kV (Tecnai Spirit, Thermo Fisher, USA), equipped with QUEMESA camera (Olympus), TEM 80 kV (TEM 900, Zeiss) equipped with a Morada G2 camera (Olympus), TEM 120 kV (Talos L120C, Thermo Fisher), equipped with Ceta16M camera (Thermo Fisher), TEM 300 kV (Titan Krios, Thermo Fisher), equipped with K3 direct electron detector with energy filter (Gatan), TEM 300 kV (Titan Krios, Thermo Fisher), equipped with Falcon3 direct electron detector (Thermo Fisher), cryoplunger (Vitrobot Mark IV, Thermo Fisher), confocal microscope (A1RHD25 microscope, Nikon Imaging Center, Curie Institute), lattice light sheet microscope LLSM 40 from 3i (Intelligent Imaging Innovations). Custom-built TIRF Microscope (with 100 x objective, 1.45 NA), equipped with an ORCA-Flash 4.0 V3 Digital CMOS camera (Hamamatsu). BioRad ChemiDoc for protein detection (chemiluminescence and fluorescence detection). Microtime 200 system (PicoQuant GmbH, Germany) for fluorescence lifetime imaging and correlation spectroscopy measurements. CHI760e Electrochemical Workstation (CH Instruments, USA) for EIS. nanoAcquity UPLC device (WatersCorporation, Milford, MA, USA), Q-Exactive HF-X mass spectrometer (Thermo Fisher Scientific, Waltham, MA, USA).

## Recombinant protein expression
Human recombinant Gal3-6xHis and Gal3ΔNter-6xHis with aa115-250 of BC053667 were cloned into pHis-Parallel2 and purified as described[23,112]. Briefly, for cloning, Gal3 and 6xHis were separated by a Leu-Glu linker at the C-terminus. Proteins were expressed overnight at 20 °C in Rossetta2-pLysS using LB media with 60 µM IPTG, and purified with cobalt resin affinity chromatography and gel filtration (Superdex75 16 × 60) in PBS at pH 7.4. Coupling with Alexa488 sulfodichlorophenol ester (Alexa488-Gal3) or Cy3 monoreactive succinimidyl ester (Gal3-Cy3) was performed overnight at 16 °C using a 4-fold molar dye excess in PBS and purified with PD10

columns. The labeling efficiency was 1–1.5. Human recombinant Gal3-TEV-6xHis was cloned into pHis-Parallel2 without a linker between Gal3 and TEV cleavage site, and 26 amino acids between the TEV site and 6xHis (FW:5' GGAATTCCATATGGCAGACAATTTTTCGCTCCATGATGCG, RV:5' CTGGATCCGCCCTGAAAATACAGGTTTTCTATCATGGTATATGAAGCACTGG). Protein expression was performed as for Gal3-6xHis. The bacterial pellet was resuspended in buffer-A (20 mM HEPES, pH 7.8, 200 mM NaCl, 80 mM lactose, 10 mM imidazole, 1 mM TCEP), sonicated, centrifuged for 60 min at 75,000 × g, and the supernatant was incubated for 1 h at 4 °C with Ni-Sepharose-Fast Flow beads (Cytiva). The beads were then washed with buffer-A containing 300 mM NaCl, and eluted in buffer-B (20 mM HEPES, pH 7.8, 20 mM NaCl, 40 mM lactose, 0.5 mM TCEP, 500 mM imidazole, pH 7.8). The eluted protein was diluted to 2 mg/mL and dialyzed in buffer-C (20 mM HEPES, 25 mM NaCl, 40 mM lactose, 0.5 mM TCEP, 0.5 mM EDTA) to remove imidazole. 20 µg shTEV-His protease per mg of Gal3 was injected into the dialysis chamber. After 24 h at 4 °C, the dialysis chamber was transferred for 8 h to a lactose and EDTA-free buffer-D (20 mM HEPES 7.3, 150 mM NaCl, 50 µM TCEP). ShTEV-His and non-cleaved Gal3-TEV-6xHis were removed by incubation with a cobalt resin. We noticed that highly purified Gal3 was very sticky to Ni-NTA. The non-bound protein fraction was concentrated and purified by FPLC-gel filtration (Superdex75 16 × 60) using buffer-D without TCEP, with an elution peak at 68 mL. When required, Gal3 was fluorophore-labeled as for Gal3-6xHis using 20 mM HEPES, 150 mM NaCl, pH 7.3, snap-frozen, and stored at −80 °C.

## Preparation of antibody-HRP conjugates
One hundred micrograms of mAb13 or 9EG7 antibodies were added to lyophilized HRP-NHS (300 µg) at a final molar ratio of 1/9 and incubated for 2 h at 21 °C. Unreacted HRP was removed using a 100 kDa cut-off concentrator.

## mAb13-HRP/9EG7-HRP internalization and sample preparation for electron microscopy
Ten micrograms per milliliter of HRP-conjugated mAb13 or 9EG7 antibody were continuously incubated for 6 to 9 min at 37 °C with HeLa cells. Cells were immediately placed on ice, washed once with DMEM supplemented with 15 mM HEPES and 1% BSA, and twice with DMEM supplemented with 15 mM HEPES without BSA. Surface-bound antibodies were removed by incubation for 10 min at 4 °C with ascorbic acid solution of DAB (in DMEM, 15 mM HEPES, 1% BSA). Enzymatic reaction of internalized HRP was developed by incubation for 20 min at 4 °C with the same solution supplemented with H$_2$O$_2$. The cells were washed three times at 4 °C with DMEM supplemented with 15 mM HEPES, and fixed overnight at 4 °C with 2.5% glutaraldehyde in PBS. Cells were washed three times at room temperature during 70 min with 0.1 M Na-cacodylate in water. Membrane fixation was performed with 1% OsO$_4$ in 0.1 M Na-cacodylate in water. After one wash with 0.1 M Na-cacodylate in water and two washes with water, contrast was obtained by incubation for 45 min with aqueous 4% uranyl acetate solution. Cells were again washed with water. Samples were then dehydrated for 10 min at room temperature by incubation with increasing concentration of aqueous ethanol solutions (1 × 50% 5 min, 1 × 70% 5 min, 2 × 90% 10 min) and 3 × 100% anhydrous ethanol solution. Cells were finally embedded in LX112 resin, ultrathin 65 nm sections were obtained using a Reichert Leica UCT ultramicrotome, and mounted on Ni/formvar/carbon-coated grids for observations. Micrographs were acquired by electron microscopy.

## Clathrin heavy chain (CHC) cell surface co-immunoprecipitation
mAb13 or 9EG7 antibodies (10 µg/mL) were incubated for 30 min at 4 °C with RPE-1 cells. Excess antibody was removed by washing with ice-cold PBS++, and cells were lysed with lysis buffer. Antibodies were pulled down from cleared lysates (post-nuclear supernatants) by

overnight incubation with 40 µl bed volume protein G-sepharose at 4 °C. Beads were washed three times with TNE 0.1% NP40, and immunoprecipitated proteins were denatured by boiling (95 °C) in SDS sample buffer. Eluted proteins were separated on denaturing non-reducing SDS-PAGE gels and immunoblotted against mAb13 or 9EG7 (anti-rat-HRP) and CHC (anti-mouse-HRP).

## Antibody binding or uptake

For plasma membrane labeling (binding), 10 µg/mL of mAb13 or 9EG7 antibodies were incubated for 30 min at 4 °C with RPE-1 cells. These were then washed three times with ice-cold PBS++, fixed with 4% PFA for further immunofluorescence detection. For uptake, 3 or 10 µg/mL of mAb13 or 9EG7 antibodies, respectively, were continuously incubated for 5 or 10 min at 37 °C with RPE-1 cells, according to experimental conditions. Cells were shifted to 4 °C and an excess of antibodies removed by washing with ice-cold PBS++. Cell surface-bound antibodies were removed with three acid washes of 45 s each. Acidic pH was then neutralized with three times ice-cold PBS++ washes, and cells were fixed in 4% PFA, followed by secondary antibody labeling (see below) and confocal microscopy.

## Concentrations of exogenous Gal3

200 nM (5 µg/mL) of Gal3 were typically used for cellular binding and uptake assays, as analyzed by confocal microscopy imaging. For experiments in which cargo internalization was stimulated by Gal3 addition (Fig. 3C, D and Supplementary Fig. 3A), we have used a range of concentrations from 20 nM to 400 nM. Note that a substantial increase in mAb13 uptake was already observed with 20 nM of Gal3. All these experiments were performed in the presence of endogenous Gal3. In microcavity array-suspended lipid bilayer (MSLB) experiments, Gal3 concentration from 0.2 to 37 nM was used (Fig. 5C, D, and Supplementary Fig. 4B). In a previous study, the endocytosis of CD44 was indeed stimulated with as little as 0.3 nM of exogenous Gal3 on mouse embryonic fibroblasts that had been depleted for endogenous Gal3[23]. For the Gal3 oligomer experiments on reconstituted $\alpha_5\beta_1$ integrin (Fig. 6A, B) and on cells (Fig. 7), Gal3 concentrations were tuned to obtain maximal amounts of oligomer material.

## Transferrin and Gal3 uptake

5 µg/mL of Tf-A546 or 200 nM of Alexa488-Gal3 were continuously incubated for 5 or 10 min at 37 °C with RPE-1 cells, according to experimental conditions.

## Immunofluorescence

Cells were fixed with 4% PFA at 4 °C for 5 min and an additional 5 min at room temperature. Excess of PFA was then quenched with 50 mM NH4Cl, followed by incubation for 30 min at room temperature with BSA-saponin saturation/permeabilization solution (intracellular immunostaining), or only with 0.2% BSA (saturation for plasma membrane labeling). Cells were incubated for 30 min at room temperature either with primary and secondary antibodies for the labeling of cellular antigens, or only with secondary antibody to detect primary antibodies that had been put in contact with living cells (uptaken antibody). Coverslips were mounted on slides with Mowiol supplemented with DAPI. All fixed cell immunofluorescence images were acquired with a Nikon A1RHD25 confocal microscope. Signals were quantified using the ImageJ program and displayed as mean intensity per cell after maximal projection of Z-stacked images. For colocalization analysis, the JACoP plugin run in ImageJ was used to measure Manders' coefficient signal co-occurrences. This quantification method calculates the percentage of the signal from one channel that overlaps with the signal from the other channel. To do so, it is of importance to image samples with high signal-to-noise ratios, and to select pixels from both channels that are relevant from the biological perspective[113].

## siCHC transfection

HeLa cells stably expressing Golgi-localized GalT-GFP-SNAP were transfected with 8 nM of siCHC (or of siCtrl) to inhibit CHC expression, using HiPerFect reagent. Antibody uptake and retrograde trafficking experiments were performed 48 h after transfection. Inhibition level of CHC expression was assessed by immunoblotting.

## EPS15DN-GFP transfection

RPE-1 cells were electroporated with 10 µg EPS15DN-GFP encoding plasmid 24 h prior to performing antibody uptake experiments. GFP-positive signal (GFP+) served to identify cells that expressed the dominant negative EPS15 mutant, and GFP-negative cells (GFP-) served as internal controls.

## CRISPR of RPE-1 AP2-eGFP

CRISPR-Cas9 genome editing of RPE-1 cells to generate isogenic mTagGFP-labeled AP2 was performed by electroporation of the following molecules. Homology directed repair template: Purified AP2-mTagGFP fragment after Snf-I digestion of pMK-RQ AP2M1-mTagGFP (vector: gift of Guillaume Montagnac, mTagGFP inserted between Ser236 and Gly237 of AP2M1 (NM_004068.4), mTagGFP-spacer at N-term: GSTGGS, at C-term: AGSGT), and four gRNAs in pX330-Cas9, designed with CHOPCHOP[114]. gRNA-2-FW: 5′CACCgagaagaggtctcattggtac, gRNA-2-RV: 5′AAACgtaccaatgagacctcttctC, gRNA-3-FW CACCgattgcttcccgctgcaagc, gRNA-3-RV AAACgcttgcagcgggaagcaatcC. Transfected cells were expanded for 2 weeks. The genome-edited population was identified by three cycles of FACS sorting and expansion, and finally tested for colocalization between AP2-mTagGFP and Tf-Alexa647 (Thermo Fisher Scientific) during endocytic pit formation.

## mAb13-ATTO488 and 9EG7-ATTO488 conjugates

A 10-fold molar excess of NHS-ATTO488 compound was incubated for 2 h at 21 °C in preservative-free PBS buffer with the corresponding antibodies. Unreacted NHS-ATTO488 was first neutralized for 10 min at 21 °C with 10 mM Tris, and further eliminated using 7 kDa cut-off Zeba spin desalting columns.

## Dynamic mAb13 and 9EG7 tracking by lattice light sheet microscopy (LLSM)

Endocytosis experiments were performed in RPE-1 cells stably expressing AP2-GFP (RPE-1 AP2-mTagGFP), as previously described with minor modifications[81]. In brief, RPE-1 AP2-mTagGFP cells were seeded 24 h before imaging on a 5 mm #1.5 thickness cover glass. The medium was changed to CO2-independent lattice light sheet (LLS) imaging medium (phenol-red-free DMEM, high glucose, glutamax, supplemented with sterile 1% BSA, 0.01% penicillin and streptomycin, 1 mM pyruvate, and 20 mM HEPES, pH 7.3). Cells were dipped for 2 min at room temperature in a tube either containing 5 µg/mL of mAb13-Cy3, 9EG7-Cy3, or Tf-Alexa546, diluted in LLS-imaging medium. After one wash in LLS-medium, coverslips were transferred into the imaging chamber of the LLS-microscope, kept at 27 °C. mAb13-Cy3/AP2-GFP and 9EG7-Cy3/AP2-GFP co-tracks were dynamically monitored by LLSM.

## Gal3/mAb13 and Gal3/9EG7 dynamic co-tracking by lattice light sheet microscopy (LLSM)

Galectin-3/integrin co-binding was performed in RPE-1 cells. These experiments were imaged as previously established with minor modifications[81]. 5 mm #1.5 thickness coverslips seeded with RPE-1 cells were sequentially dipped for 2 min at 4 °C in a solution of Gal3-Cy3 (200 nM, in LLS-imaging medium), once washed in LLS-imaging medium, and then plunged at 4 °C in a tube either containing 5 µg/mL of mAb13-ATTO488 or 9EG7-ATTO488, diluted in LLS-imaging medium. After one LLS-medium wash, coverslips were transferred into the

imaging chamber. Co-tracks of Gal3-Cy3/mAb13-ATTO488 and Gal3-Cy3/9EG7-ATTO488 were dynamically monitored by LLSM and further processed.

## LLSM acquisition

Acquisitions in LLS-imaging medium were performed at 27 °C for increased stability of the optical system. 4D acquisition started within 2–4 min using a commercial LLSM of 3i (Denver, USA), as previously described[115]. Cells were scanned incrementally with a 20 µm light sheet in 600 nm steps using a fast piezoelectric flexure stage equivalent to ~325 nm with respect to the detection objective, and were imaged using two sCMOS cameras (Orca-Flash 4.0; Hamamatsu, Bridgewater, NJ). Excitation was achieved with 488 nm (Sapphire Coherent) or 560 nm (MPB Communications) diode lasers at 10–20% acousto-optic tuneable filter transmittance with 300 mW (initial box power) through an excitation objective (Special Optics 28.6 × 0.7 NA 3.74-mm water-dipping lens) and detected via a Nikon CFI Apo LWD 25 × 1.1 NA water-dipping objective with a 2.5× tube lens. LLSM imaging was performed using an excitation pattern of outer NA equal to 0.55 and inner NA equal to 0.493. A composite volumetric dataset of 60 slices per cell was acquired within 1.5–1.8 s using 10 ms exposure per slice and channel for mAb13-Cy3/AP2-GFP or 9EG7-Cy3/AP2-GFP co-tracking experiments, and within 2–3 s using 10–20 ms exposure time per slice and channel for Gal3-Cy3/mAb13-ATTO488 and Gal3-Cy3/9EG7-ATTO488 co-tracking experiments. Eighty to one hundred twenty time points were acquired per cell. Raw images of the obtained datasets were quantitatively analyzed using an adapted version of the previously published cmeAnalysis3D software[81], as described in the following section.

## Quantitative analysis and visualization of mAb13/AP2 or 9EG7/AP2 co-tracking experiments

Post-processing of raw data volumes was carried out as described previously[81]. Automated detection of AP2-coated structures or punctate structures of fluorescently labeled cargoes in 3D (mAb13-Cy3 antibody, 9EG7-Cy3 antibody, and Tf-Alexa546) was performed by numerical fitting with a model of the microscope point spread function (PSF), as described previously[116]. Automated tracking of cargo and clathrin was calculated using the u-track software package[117], as part of the cmeAnalysis3D software[116], which was implemented in Matlab 2021b. AP2 and cargo positions were exploited to map the membrane shape, using the Matlab function of alphaShape[81]. We then calculated the displacement of each cargo from the plasma membrane. The distinction between point-like structures moving inside the plasma membrane and internalized molecules, whether AP2-positive or AP2-negative, was made using membrane position detection. An event was counted as endocytic uptake if an object underwent a net displacement of at least 150 nm from the initial position inside the membrane proximal zone[81]. The membrane proximal zone was limited by the contour of the alpha-Shape and a line 400 nm inwards of the cell[81]. To calculate the number of AP2-positive and AP2-negative events, the presence of AP2 was assessed for each qualified internalization event of the cargo channel using the cmeAnalysis3D software. Only tracks with durations of more than 8 s were used for this analysis. The calculation of the lifetime distributions and intensity cohorts was performed as described previously[116]. The raw LLSM images, used in Supplementary Fig. 3I and L, and Supplementary Movies 3–6 were deconvolved using LLSpy v0.4.8 (https://doi.org/10.5281/zenodo.3554482). Supplementary Movies 3 and 4 were rendered and visualized using Imaris software 9.82. Supplementary Movies 5 and 6 were rendered and visualized using ImageJ[118]/Fiji 1.53c[119]. The analysis code can be found as part of the GitHub repository of llsmtools in https://github.com/francois-a/llsmtools/. Statistical analyses were performed using Prism v9.4.1 software (Graphpad Inc.).

## Quantitative analysis and visualization of mAb13/Gal3 or 9EG7/Gal3 co-tracking experiments

Post-processing of raw data volumes was carried out as described[81]. Automated detection of punctate structures of Gal3-Cy3 and cargoes (mAb13-ATTO488 and 9EG7-ATTO488 antibodies) in 3D was performed by numerical fitting with a model of the microscope PSF as described previously[116]. Automated tracking of cargo and Gal3 was calculated using the u-track software package[116,117], which was implemented in Matlab 2021b. To calculate the number of Gal3-positive and Gal3-negative events, the presence of Gal3 was assessed for each qualified internalization event of the cargo channel using the cmeAnalysis3D software. Cargo tracks were considered Gal3-positive if they colocalized more than 10 s with Gal3. Only tracks with a duration of more than 10 time points were used for the analysis. Speed and distance of co-tracks were estimated using positions from the cmeAnalysis3D software. The raw LLSM images, used in Fig. 2D and Supplementary Movies 1 and 2, were deconvolved using LLSpy (v0.4.8) (https://doi.org/10.5281/zenodo.3554482) before video rendering and visualization using napari (v0.4.12) (doi: 10.5281/zenodo.3555620). Statistical analyses were performed using Prism v9.4.1 software (Graphpad Inc.).

## Ciliobrevin D (CBD) treatment

RPE-1 cells were pre-treated for 30 min at 37 °C with 50 µM CBD. CBD was then kept during all subsequent incubations at 37 °C. For uptake experiments with pre-loaded exogenous Gal3, CBD pre-treatment was followed by sequential incubation of 200 nM Gal3 and 10 µg/mL of mAb13 or 9EG7 antibodies on RPE-1 cells for 30 min at 4 °C, before further incubation for 10 min at 37 °C.

## I3 treatment

Cells were pre-treated with 10 µM I3 for 5 min at 37 °C. I3 was washed out with PBS++ for acute endocytosis assays, or kept during subsequent incubations for retrograde trafficking experiments.

## Genz treatment

RPE-1 cells were continuously incubated for 2 days with 5 µM Genz-123346 in DMEM-F12 medium containing 5% FCS, prior to performing mAb13 and 9EG7 antibody uptake assays.

## PNGase F treatment

RPE-1 or dKO-MKF cells were incubated for 24 h at 37 °C with 5000 units/mL of glycerol-free PNGase F in DMEM-F12 or DMEM supplemented with 1% FCS, to remove N-linked glycans on membrane glycoproteins. Efficient removal of N-glycans from the cell surface was assessed by incubating PNGase F-treated or untreated RPE-1 cells for 15 min at 4 °C with 0.3 µg/mL of Alexa488-conjugated PHA-L lectin and subsequent confocal microscopy analysis.

## Sialidase treatment

RPE-1 cells were incubated for 40 min at 37 °C with 75 mU/ml of sialidase (neuraminidase from *Arthrobacter ureafaciens*) in serum-free DMEM-F12 medium. Efficient removal of sialic acids was assessed by incubating sialidase-treated or untreated RPE-1 cells for 15 min at 4 °C with 10 µg/mL of Alexa488-conjugated SiaFind lectin and subsequent confocal microscopy analysis.

## Incubations with exogenous Gal3/Gal3ΔNter

For antibody binding and uptake assays, 10 µg/mL of mAb13 or 9EG7 antibodies were incubated for 30 min at 4 °C with RPE-1 cells. Excess antibodies were removed by washing with PBS++, and cells were then shifted to 37 °C in the presence of the indicated concentrations of exogenous Gal3 or Gal3ΔNter in serum-free DMEM-F12 medium. For retrograde transport assays, HeLa cells stably expressing GalT-GFP-SNAP were continuously co-incubated for 3 h at 37 °C with 10 µg/mL of

BG-coupled mAb13 antibody and 200 nM of Gal3 in serum-free DMEM medium.

### Gal3/Gal3ΔNter and mAb13/9EG7 or mAb16/SNAKA51 co-binding and co-uptake assays

Exogenous Gal3 or Gal3ΔNter (200 nM) were incubated for 30 min at 4 °C with RPE-1 cells in serum-free DMEM-F12 medium. Excess of Gal3 or Gal3ΔNter was removed by washing with the same ice-cold medium. 10 µg/mL of integrin antibodies (mAb13 or 9EG7 for $\beta_1$ integrin, mAb16 or SNAKA51 for $\alpha_5$ integrin) were then incubated for 30 min at 4 °C with the same cells that had already been pre-incubated with Gal3. Excess antibodies were removed by washing, and cells were either fixed in 4% PFA (co-binding assay) or shifted for 10 min to 37 °C (co-uptake assay). For the latter, residual cell surface accessible Gal3 was removed by incubations at 4 °C with 150 mM β-D-lactose (three times, 5 min), and residual cell surface accessible integrin antibodies by acid washes. Immunofluorescence was then performed as described above.

### Gal3 and mAb13/9EG7 co-binding and co-uptake assays on micropatterned RPE-1 cells

Line patterns were produced, and cells were seeded as previously described[56]. Coverslips (25 mm) were micropatterned with 9 µm-wide lines and covered with 5 µg/mL fibronectin. RPE-1 cells (30,000 per well) were seeded onto these coverslips and left to adhere in the incubator for at least 5 h before further manipulations. Gal3 and mAb13 or 9EG7 co-binding and co-uptake were performed as described above.

### Gal3-CFP/mAb13 or 9EG7 cell surface co-immunoprecipitation experiments

10 µg/mL of mAb13 or 9EG7 antibodies were incubated for 30 min at 4 °C with transiently Gal3-CFP expressing HeLa cells. Excess of antibodies was removed with three times ice-cold PBS++ washes, and cells were lysed in lysis buffer. Cleared lysates were incubated with 30 µl slurry of GFP-Trap beads for an overnight pulldown at 4 °C. After 3 washes in TNE 0.1% NP40, bound proteins were eluted from beads, and denatured at 95 °C heating in SDS sample buffer. Samples were loaded onto SDS-PAGE gels, immunoblotted with anti-rat HRP antibodies against the bound antibodies (co-IP mAb13 and 9EG7). Antibody levels were quantified. Gal3-CFP fluorescence signals (pulldown) served as loading controls.

### Gal3/mAb13 or 9EG7 cell surface co-immunoprecipitation upon sialidase treatment

RPE-1 cells were seeded in 5 cm dishes 24 h before the experiment. After sialidase treatment (see "sialidase treatment" section), SETA-555 labelled His-tagged Gal3 (5 µg/mL) and mAb13 or 9EG7 (5 µg/mL) were sequentially bound at 4 °C to RPE-1 cells, as described above. Cells were then lysed for 30 min at 4 °C, and the post-nuclear supernatants were applied onto 30 µl slurry of pre-washed cobalt-coated beads for an overnight incubation at 4 °C. Proteins were eluted from beads and denatured by boiling in SDS sample buffer, run onto SDS-PAGE, and directly analyzed by fluorescence for the detection of Gal3, or immunoblotted with anti-rat IgG-HRP antibody for the detection of mAb13 or 9EG7.

### mAb13-BG and Gal3-BG conjugates

BG-GLA-NHS compound was incubated for 6 h at 4 °C in preservative-free PBS buffer with a 10-fold molar excess over antibodies, or for 2 h at 21 °C with a 3-fold molar excess over Gal3. For antibodies, unreacted BG was eliminated by overnight dialysis against PBS using 10 kDa cut-off dialysis cassettes. For Gal3, unreacted BG was removed using 7 kDa cut-off Zeba spin desalting columns. For fluorescence microscopy analysis of retrograde trafficking, mAb13-BG and Gal3-BG were further incubated for 1 h at 21 °C with a 3-fold molar excess of Cy3-NHS. Unreacted Cy3 was removed using 7 kDa cut-off Zeba spin desalting columns.

### Biochemical retrograde trafficking analysis

HeLa cells stably expressing GalT-GFP-SNAP were continuously incubated for 2–3 h at 37 °C with 10 µg/mL mAb13-BG or 200 nM Gal3-BG conjugates in the indicated conditions. Excess of antibody or Gal3 was removed by washing with DMEM, and the unreacted GalT-GFP-SNAP was quenched for 20 min at 37 °C with 10 µM SNAP-cell®-Block. Cells were lysed for 30 min at 4 °C with lysis buffer, and cleared lysates were loaded on a 30 µl slurry bed of either G-sepharose (for mAb13-BG or 9EG7-BG pulldown) or GFP-Trap (for Gal3-BG pulldown) beads for overnight pulldown at 4 °C on a rotating wheel. Samples were washed three times with TNE 0.1% NP40 buffer. Proteins were eluted by boiling in SDS sample buffer, loaded on denaturing non-reducing SDS-PAGE gels, and immunoblotted with anti-SNAP antibody. When mAb13-BG or Gal3-BG reached the Golgi compartment, a covalent reaction with the SNAP-tag occurred, resulting in the formation of covalent mAb13 or Gal3/GalT-GFP-SNAP protein species that were detected by western blotting (anti-SNAP immunoblots) and termed SNAP-mAb13 and SNAP-Gal3, respectively.

### Retrograde trafficking analysis by fluorescence microscopy

10µg/mL Cy3-mAb13-BG was continuously incubated for 1 h at 37 °C with GalT-GFP-SNAP expressing HeLa cells. Cells were then prepared for fluorescence imaging by confocal microscopy. For investigation of clathrin-dependent endocytosis contribution to retrograde trafficking, 200 nM Cy3-Gal3-BG or 10 µg/mL Cy3-mAb13-BG were incubated as described above, either under siCtl or siCHC depletion conditions.

### mAb13/9EG7 colocalization with Gal3 and retromer complex

Gal3 and mAb13 or 9EG7 were sequentially bound at 4 °C onto RPE-1 cells, as described above. Cells were then shifted for 15 min to 37 °C, placed at 4 °C, and successively washed with β-D-lactose and acid buffers (see co-binding/co-uptake assays section) to remove cell surface accessible ligands. After PFA fixation, cells were permeabilized and further immunolabeled with anti-VPS26 antibody (see immunofluorescence section).

### mAb13/9EG7 and Gal3 co-immunoprecipitation with the retromer complex

Gal3 (5 µg/mL) and mAb13 (5 µg/mL) or 9EG7 (10 µg/mL) were sequentially bound at 4 °C onto RPE-1 cells seeded in 10 cm dishes 24 h before experiments, as described above. Cells were shifted for 20 min to 37 °C, then cooled to 4 °C, and successively washed with β-D-lactose and acid buffers (see co-binding/co-uptake assays section) to remove cell surface accessible ligands. Cells were lysed for 30 min at 4 °C in low-salt lysis buffer, and the post-nuclear supernatants were applied onto 30 µl slurry of pre-washed G-Sepharose beads for an overnight incubation at 4 °C. Proteins were eluted from beads and denatured by boiling in SDS sample buffer, loaded onto SDS-PAGE gels, and then immunoblotted with anti-VPS35 antibody, or directly with anti-rat IgG-HRP antibody for the detection of mAb13 or 9EG7. Co-precipitated Gal3 was detected by fluorescence.

### Purification of $\alpha_5\beta_1$ integrin from rat livers

Micellar $\alpha_5\beta_1$ integrin was solubilized and purified as described previously[82]. Protein purity was analyzed by Stain-Free™ SDS-PAGE. $\alpha_5\beta_1$ integrin-enriched fractions were then pooled, and the final concentration was determined by the colorimetric Bradford assay. $\alpha_5\beta_1$ integrin was snap-frozen and stored at −80 °C.

### Negative stain EM of purified $\alpha_5\beta_1$ integrin

$\alpha_5\beta_1$ integrin in micelles or reconstituted in lipid nanodiscs or peptidiscs, as described below, was incubated for 30 s on freshly glow-discharged carbon-coated EM grids before staining with 2% of uranyl acetate. Micrographs were recorded at 80 kV on a TEM 900 (Zeiss)

equipped with a Morada G2 camera (Olympus) or a Tecnai Spirit equipped with a QUEMESA camera.

## Negative stain EM of Gal3 oligomers

Gal3 oligomers eluted from $\alpha_5\beta_1$ integrin-Gal3 complexes in nanodiscs, peptidiscs, or from RPE-1 cells (1:3 dilution in PBS) were incubated for 3 min on freshly glow-discharged carbon-coated EM grids before staining with 2% of uranyl acetate. 20 μg/mL of Gal3 alone (monomer) was used as a control. Micrographs were recorded on a Talos L120C TEM equipped with a Ceta16M CCD detector at a pixel size of 1.58 Å per pixel in low-dose mode.

## Sialidase treatment of purified $\alpha_5\beta_1$ integrin

When specified, purified $\alpha_5\beta_1$ integrin was treated with sialidase (neuraminidase from *Arthrobacter ureafaciens*) at a ratio of 0.08 U of enzyme for 100 μg of $\alpha_5\beta_1$ integrin to remove terminal sialic acids from integrin glycans.

## Gal3 interaction with $\alpha_5\beta_1$ integrin in microcavity array-suspended lipid bilayers

$\alpha_5\beta_1$ integrin was reconstituted into microcavity array-suspended lipid bilayers at a lipid to protein ratio of 10:1. The microcavity array-suspended lipid bilayers were prepared at gold or PDMS polymer substrates for EIS or FLIM (FCS), respectively, according to procedures described previously[83]. Membrane capacitance and FLIM measurements were then performed in the presence of increasing concentrations of Gal3. Both FLIM and EIS studies were conducted with microcavity array-suspended lipid bilayers filled with and in contact with 10 mM HEPES buffer. Integrin activation was accomplished through sequential addition of 5 mM $Mn^{2+}$, with 30 min incubation, then 1 mM cRGD with 90 min incubation, to the contacting solution. Addition of Gal3 at the indicated concentrations was followed by an equilibration time of 30 min. These times were confirmed to be sufficient for protein binding/equilibration in all cases. All measurements were carried out at room temperature ($22 \pm 1$ °C) and in triplicate. Capacitance values were extracted from EIS data by fitting to an equivalent circuit model reported previously and analyzed using Z-View software (Scribner Associates, v3.4e). Fits were assessed from both, visual inspection of the fit residuals, and from $\chi^2$ (typically ~0.001). As absolute membrane resistance and capacitance can vary with substrate, the average relative changes to membrane resistance ($\Delta R$) and capacitance ($\Delta Q$) are reported as ($R_M^X - R_M^0$) and ($Q_M^X - Q_M^0$); where $R_M^0$ and $Q_M^0$ represents respectively the absolute membrane resistance and capacitance in the absence of lectin, and $R_M^X$ and $Q_M^X$ are the respective values of membrane resistance and capacitance values in presence of lectin. For FLIM and FCS measurements, $\alpha_5\beta_1$ integrin and Gal3 were fluorescently labeled with ATTO488 and Alexa647, respectively. Out of a 20% weight of the total integrin, 5% weight was fluorescently labeled with ATTO488. FLIM/FCS measurements were carried out in triplicate at 20 °C. Data collection, analysis, and extraction of D values were conducted as described previously[83].

## Direct Gal3 interaction with $\alpha_5\beta_1$ integrin conformers in Triton X-100 micelles

The shift from the inactive bent-closed $\alpha_5\beta_1$ integrin to the active extended ligand-bound conformational state was achieved as follows: 0.5 μg $\alpha_5\beta_1$ integrin was sialidase-treated as described above, and further incubated for 1 h at 21 °C under gentle agitation at a final concentration of 10 μg/mL in 45 μl HEPES/Triton X-100 buffer, supplemented with 5 mM $MnCl_2$. cRGD peptide was then added at a final concentration of 100 μM, and incubated overnight at 4 °C under gentle agitation. This freshly activated $\alpha_5\beta_1$ integrin, or the inactive bent-closed one, was then incubated for 1 h at 21 °C under gentle agitation with 200 nM of Alexa488-labeled Gal3 (Alexa488-Gal3). Samples were chilled for 10 min on ice, and $\alpha_5\beta_1$ integrin-Alexa488-Gal3 complexes

were cross-linked for 20 min at 4 °C with 2.5 mM glutaraldehyde. Reactions were stopped by incubation for 20 min at 4 °C with 20 mM Tris, pH 7.4. Samples were supplemented with 3x non-reducing SDS sample buffer (2% SDS), run for PAGE at 100 V without SDS (semi-native conditions). Alexa488-Gal3 signal was directly detected on gels (Alexa488 channel). Proteins were then transferred onto nitrocellulose membranes and immunoblotted for rat $\beta_1$ integrin.

## $\alpha_5\beta_1$ integrin reconstitution in nanodiscs

Reconstitutions were performed at a R1 = lipid/MSP1D1 ratio of 50 mol/mol, and a R2 = $\alpha_5\beta_1$ integrin/nanodisc ratio of 1 mol/mol, as described previously[82]. For one reconstitution reaction (50 μl final volume), His-tagged MSP1D1 scaffold protein was diluted to 4 μM (100 μg/mL) in HEPES buffer supplemented with 0.06% Triton X-100 and incubated for 10 min at 21 °C under gentle stirring. ePC/bPS (90/10, mol/mol) lipid mix was added at a final concentration of 90 μg/mL, still under stirring, and further incubated for 10 min at 21 °C under gentle stirring. $\alpha_5\beta_1$ integrin in HEPES/Triton X-100 buffer was added at a final concentration of 0.25 mg/mL, and incubated for 20 min at 4 °C under gentle stirring. Triton X-100 detergent was finally removed by incubation under gentle stirring for 30 min at 4 °C with 6 mM of Heptakis (2,6-di-O-methyl)-β-cyclodextrin. For Cryo-EM of nanodisc-embedded $\alpha_5\beta_1$ integrin-Gal3 complexes, ePC/bPS/GSLs (85/10/5, mol/mol/mol) lipid mix was used instead.

## Direct Gal3 interaction with $\alpha_5\beta_1$ integrin conformers in nanodiscs

For in vitro activation, $\alpha_5\beta_1$ integrin in nanodiscs was first incubated for 30 min at 21 °C with 5 mM $MnCl_2$, followed by an additional 2 h incubation at 21 °C with 10 μM $FNIII_{9-10}$ fibronectin fragment. This freshly activated $\alpha_5\beta_1$ integrin, or the untreated inactive bent-closed one, was adjusted to 500 μl HEPES buffer (supplemented with 5 mM $MnCl_2$ and 10 μM $FNIII_{9-10}$ fibronectin fragment for the active condition) and incubated with 200 nM Cy3-Gal3-His for 15 min at 18 °C on a rotating wheel. The mixture was then incubated for 1 h at 4 °C with 30 μl bed volume of cobalt beads. Beads were washed three times with 500 μl HEPES buffer, and $\alpha_5\beta_1$ integrin-Gal3 complexes were eluted by incubation for 1 h at 21 °C with 40 μl of HEPES buffer supplemented with 10 mM EDTA. Samples were denatured with 3x concentrated non-reducing SDS sample buffer, boiled for 5 min at 95 °C, and run on SDS-PAGE. $\alpha_5\beta_1$ integrin and Gal3 were detected using the Stain-Free™ and fluorescence (Cy3 channel) modes, respectively.

## Glycan-dependent binding of Gal3 to purified $\alpha_5\beta_1$ integrin

Gal3 was pre-incubated or not for 10 min at room temperature with either 50 mM β-D-lactose or 100 μM I3 inhibitor. Binding to $\alpha_5\beta_1$ integrin in Triton X-100 micelles or reconstituted in nanodiscs was then performed as described above. Samples were denatured with 3x concentrated non-reducing SDS sample buffer, boiled for 5 min at 95 °C, and run on SDS-PAGE. $\alpha_5\beta_1$ integrin and Gal3 were detected using the Stain-Free™ SDS-PAGE.

## Elution of Gal3 or Gal3ΔNter from $\alpha_5\beta_1$ integrin in nanodiscs

Four reconstitution reactions as described above were incubated overnight at 4 °C on a rotating wheel with 30 μl cobalt bead bed volume in 500 μl HEPES buffer. Beads were washed three times with 500 μl HEPES buffer. The cobalt bead-immobilized $\alpha_5\beta_1$ integrin in nanodiscs was incubated in 500 μl HEPES buffer for 2 h at 21 °C on a rotating wheel, either with 4 μM of Gal3 or Gal3ΔNter. Beads were washed three times with 500 μl HEPES buffer. Gal3/Gal3ΔNter were specifically eluted with 40 μl of 50 μM I3 inhibitor diluted in HEPES buffer. Elution was performed by gentle manual shaking for 10 min at 21 °C, and samples were immediately negatively stained for EM as previously described. As control, monomeric Gal3 in solution (0.8 μM) was incubated with 50 μM I3 inhibitor for 10 min at 21 °C, and

negatively stained for EM. For the oligomer stability study, eluted samples were kept for the indicated times at 4 °C until loading onto EM grids.

### Gal3 titration for oligomer assembly

Cobalt bead-immobilized $\alpha_5\beta_1$ integrin in nanodiscs was incubated in 500 µl HEPES buffer for 2 h at 21 °C on a rotating wheel, with 0.1, 0.4, 1.2, and 4 µM of Gal3 to reach 1, 4, 12, and 40 molar ratios between Gal3 and $\alpha_5\beta_1$ integrin, respectively. Beads were washed three times with 500 µl HEPES buffer. Gal3 was specifically eluted with 40 µl of 50 µM I3 inhibitor diluted in HEPES buffer. Elution was performed by gentle manual shaking for 10 min at 21 °C. Samples were immediately negatively stained for EM as described above, and analyzed by Stain-Free™ SDS-PAGE.

### I3 titration for Gal3 elution from $\alpha_5\beta_1$ integrin

Cobalt bead-immobilized $\alpha_5\beta_1$ integrin in nanodiscs was incubated in 500 µl HEPES buffer with 4 µM of Gal3 for 2 h at 21 °C on a rotating wheel. Beads were washed three times with 500 µl HEPES buffer, and Gal3 was successively eluted with 40 µl of 0.001, 0.05, 0.1, 0.5, 1, 5, and 50 µM I3 inhibitor solutions, under gentle manual shaking for 10 min at 21 °C. At each elution step, the supernatant was harvested, and the beads were immediately resuspended in solution with the next higher I3 concentration. Samples were instantly analyzed by negative stain EM and SDS-PAGE.

### $\alpha_5\beta_1$ integrin reconstitution in peptidiscs

For each reconstitution reaction, 10 µg of purified $\alpha_5\beta_1$ integrin was incubated for 20 min at 21 °C at a final concentration of 0.5 mg/mL with 4.5 µg of peptidisc/peptidisc-His mixture (1:1, w/w). Triton X-100 detergent was removed by the addition of 15 mM of Heptakis (2,6-di-O-methyl)-β-cyclodextrin. Four reconstitution reactions were pooled in 500 µl HEPES buffer, mixed with 30 µl bed volume cobalt beads, and incubated overnight at 4 °C on a rotating wheel. Beads were then washed three times with 500 µl of HEPES buffer. Peptidisc-reconstituted $\alpha_5\beta_1$ integrin was finally eluted by incubation for 20 min at 21 °C under shaking in 40 µl of the same buffer supplemented with 10 mM EDTA. Samples were further analyzed by Blue Native PAGE and negative stain EM to qualitatively assess mono-insertion of $\alpha_5\beta_1$ integrin heterodimers in peptidiscs.

### $\alpha_5\beta_1$ integrin activation in peptidiscs

$\alpha_5\beta_1$ integrin in peptidisc (two reconstitution reactions as above) was incubated for 1 h at 21 °C with 5 mM MnCl$_2$, and 100 µM cRGD peptide. Samples were then incubated for 2 h at 4 °C on a rotating wheel with 30 µl bed volume of cobalt beads. Beads were washed three times with HEPES buffer, supplemented with 5 mM MnCl$_2$, and integrin eluted for 20 min at 21 °C with 250 mM imidazole HEPES solution, 5 mM MnCl$_2$. Imidazole was removed using 7 kDa cut-off Zeba spin desalting columns equilibrated with HEPES buffer, 5 mM MnCl$_2$. Efficient switch from the inactive bent-closed to the active extended ligand-bound conformation was monitored by negative stain EM and Blue Native PAGE.

### Preparation of $\alpha_5\beta_1$ integrin-Gal3 complexes in peptidiscs for photobleaching experiments

$\alpha_5\beta_1$ integrin was reconstituted into peptidiscs as described above, except that a peptidisc/peptidisc-His/peptidisc-biotin (1/1/1, w/w/w) mixture was used. After overnight incubation with cobalt beads, immobilized peptidiscs were resuspended in 500 µl of HEPES buffer and incubated with sialidase on a rotating wheel according to conditions described above. Beads were washed three times with 500 µl of HEPES buffer and further incubated for 2 h at 21 °C on a rotating wheel with 4 µM of Gal3-Cy3. Beads were washed three times with 500 µl of HEPES buffer, $\alpha_5\beta_1$ integrin-Gal3-Cy3 complexes were eluted for 10 min

with 40 µl of HEPES buffer supplemented with 10 mM EDTA under manual and gentle resuspension of beads, and immediately analyzed in photobleaching experiments.

### Photobleaching analysis of $\alpha_5\beta_1$ integrin-Gal3 complexes in peptidiscs

Glass coverslips (Menzel-Gläser, thickness #1) were first washed for 20 min under sonication with chloroform, followed by 5 min washes with water. The coverslips were then sonicated for 20 min with 1 M KOH buffer, followed by three times rinsing with water, and a final sonication for 20 min with water. The clean coverslips were then dried under a gentle nitrogen stream and plasma cleaned for 1 min. A double-sided tape mask was used to create a micro chamber for sample incubation to generate surface-immobilized peptidisc samples for fluorescence imaging[120]. To immobilize the freshly prepared $\alpha_5\beta_1$ integrin-Gal3 complexes, chambers were first incubated for 1 h with silane-PEG2000 mixed with 1.5% of silane-PEG3400-biotin (LaysanBio) at 5 mM concentration, for 10 min with neutravidin solution at 20 µg/mL, and for 15 min with β-casine at 0.5 mg/mL. The chambers were thoroughly rinsed at each incubation step by injecting 100 µL of buffer. Finally, the $\alpha_5\beta_1$ integrin-Gal3 complexes in peptidiscs were incubated for 10 min in the chamber and were imaged under glucose oxidase oxygen scavenger conditions. For photobleaching experiments, a custom-built TIRF Microscope (with 100× objective, 1.45 NA) was used to excite Gal3-Cy3 with a 532 nm laser. Fluorescence images were recorded with 50 ms exposure time using an ORCA-Flash 4.0 V3 Digital CMOS camera (Hamamatsu). All experiments were performed at 21 °C. The recorded fluorescence stream was processed using Matlab-based open-source iSMS and AutoStepfinder software[121,122]. $\alpha_5\beta_1$ integrin reconstituted in biotin-free peptidiscs, as well as Gal3-Cy3 alone, were used as controls.

### Gal3 oligomers elution from RPE-1 and dKO-MKF cells

RPE-1 or dKO-MKF cells (knockout for $\alpha_5$ and $\beta_1$ chains) were cooled for 10 min on ice and then washed three times for 5 min with ice-cold 150 mM β-D-lactose solution to remove endogenous surface-bound Gal3. Cells were then extensively washed with ice-cold PBS$^{++}$ and further incubated for 30 min at 4 °C with 50 µg/mL exogenous Gal3 diluted in serum-free DMEM-F12. Unbound Gal3 was removed by 3 washes with PBS$^{++}$. Surface-bound Gal3 was eluted for 30 min at 4 °C with 20 µM of I3. Samples were analyzed by negative stain EM for 2D classification as described below. For binding and uptake assays (see below), parts of the RPE-1 cell samples were desalted two times with 7 kDa cut-off spin-columns to remove the I3 compound. Non-desalted Gal3 sample (I3 present) was used as a negative control to assess glycan-dependency in the rebinding experiment, as described below. For binding and uptake assays with eluted oligomers, Cy3-labeled Gal3 was used.

The importance of N-glycans and GSLs in the Gal3 oligomerization process was investigated using PNGase F or Genz treatments, respectively, as described in previous sections. Experimental procedures for Gal3 binding and oligomerization analysis by negative stain EM are equally described in previous sections, except that Gal3 was used at 10 µg/mL instead of 50 µg/mL.

### Characterization of cell-eluted Gal3 oligomer binding strength

RPE-1 cells were cooled for 10 min on ice and then incubated for 10 min at 4 °C with increasing concentrations (0.2–5 µg/mL) of either Gal3-Cy3 oligomers eluted from cells, or monomeric Gal3-Cy3. Cells were then washed with cold PBS$^{++}$, immediately fixed with 4% PFA, and analyzed by confocal fluorescence microscopy.

### Glycan and GSL-dependent binding and uptake of Gal3 oligomers that were eluted from cells or from $\alpha_5\beta_1$ integrin

RPE-1 cells were untreated (control) or PNGase F-treated (see above) to remove N-linked glycans from membrane glycoproteins. For binding

assays, cells were cooled for 10 min on ice. Five micrograms per milliliter 5 μg/mL of cell-eluted Gal3-Cy3 oligomers (see above) or monomeric Gal3-Cy3 were incubated for 10 min at 4 °C with PNGase F-treated or untreated RPE-1 cells. These cells were then washed with cold PBS++, immediately fixed with 4% PFA, and analyzed by immunofluorescence. For uptake experiments, cells were continuously incubated for 2 min at 37 °C with Gal3 oligomers or monomers at 5 μg/mL, placed on ice, incubated with cold β-D-lactose to remove non-internalized Gal3-Cy3, washed with cold PBS++, and immediately fixed with 4% PFA. In some conditions, these experiments were performed on RPE-1 cells from which GSLs had also been depleted using Genz-123346. In some experiments, I3 was not removed from cell-eluted Gal3-Cy3 oligomers to ascertain that binding specificity was maintained and that differences in binding efficiency between cell-eluted Gal3-Cy3 oligomers and Gal3-Cy3 monomers do not come from contaminating factors. In some experiments, Gal3 oligomers were eluted from $\alpha_5\beta_1$ integrin. Their interaction with PNGase F-treated or untreated RPE-1 cells was then studied as above.

### Data processing of Gal3 oligomers

**Autopicking method for Gal3 oligomers eluted from nanodiscs-embedded $\alpha_5\beta_1$ integrin-Gal3 complexes.** Monomers and oligomers on negative stain EM micrographs of Gal3 eluted from inactive and active $\alpha_5\beta_1$ integrin were automatically picked and quantified using crYOLO[123]. First, Gal3 monomers and oligomers, distinguishable by their different size, were manually picked on 11 micrographs of inactive $\alpha_5\beta_1$ integrin and used to train two autopicking models. 100 and 145 negative stain EM micrographs of Gal3 eluted from inactive and active $\alpha_5\beta_1$ integrin, respectively, were then automatically picked with the two models.

**Manual picking method for Gal3 oligomers eluted from nanodiscs-embedded $\alpha_5\beta_1$ integrin-Gal3 complexes.** Auto-quantification may include irrelevant objects that resemble Gal3 oligomers, but which don't have the defined ring-shaped structure. For example, free nanodiscs can erroneously be selected as Gal3 oligomers. We therefore also provided additional quantification by manually counting structures that rigorously meet our defined criteria. From EM micrographs of 1 × 0.665 μm size, four smaller fields of 0.4 × 0.25 μm were selected to facilitate the counting of individual objects. Percentages of defined oligomers and monomers per rectangle were quantified.

**Autopicking method for Gal3 oligomers eluted from RPE-1 cells.** Eighty-three negative stain micrographs were imported into cryoSPARC[124]. An initial set of 16,640 particles was picked using the blob picker, followed by two rounds of 2D classification to generate templates for autopicking. Template-based autopicking then identified 29,789 particles, which were extracted at a size of 3.16 Å per pixel and subjected to 2D classification. The obtained 2D classes were inspected and sorted into Gal3 dimers, trimers, and tetramers. Monomers and oligomers, including dimers, trimers, and tetramers, were automatically re-picked using Topaz[125]. Initially, 13,565 particles were selected, and 12,728 particles were retained after the first round of 2D classification. Subsequently, three 3D models were generated in cryoSPARC from these 12,728 particles using 3D classification, with four Gal3 CRDs manually fitted to represent the Gal3 tetramer.

### Cryo-EM of peptidisc-embedded $\alpha_5\beta_1$ integrin

Peptidisc-embedded and sialidase-treated $\alpha_5\beta_1$ integrin was vitrified on Quantifoil 2/1 Cu 300 mesh grids using a Vitrobot Mark IV set to a blot force of −1, blotting time of 3.0 s, 100% humidity, and temperature of 12 °C. 2905 micrographs were acquired using a FEI Titan Krios G3i microscope (Thermo Fisher Scientific) operated at 300 kV equipped with a FEI Falcon 3EC detector (Thermo Fisher Scientific) running in counting mode at a nominal magnification of 96,000×, giving a calibrated pixel size of 0.832 Å/px. Movies were recorded for 40.78 s, accumulating a total electron dose of $42 \, e^-/Å^2$ fractionated into 33 frames. EPU 2.8 was utilized for automated data acquisition with AFIS enabled using a nominal defocus between −0.8 and −2 μm.

### Data processing of peptidisc-embedded $\alpha_5\beta_1$ integrin

Data processing is outlined in Supplementary Fig. 6D. 2905 movies were aligned in MotionCor2[126] and imported into cryoSPARC for subsequent patch CTF estimation. After sorting out bad images, 2884 micrographs were chosen for further processing. 45,681 particles were selected and extracted using the blob picker for initial 2D classification and generation of autopicking templates. Subsequent template-based autopicking identified 2,005,236 particles. After particle curation, 1,444,502 particles were extracted (2x binning) for 2D classification. 470,092 particles in integrin-shaped 2D classes were re-extracted without binning, an initial model was generated from 66,630 particles, and all particles were processed in a further round of 2D classification, after which 432,656 particles were selected. 3D refinements of these particles resulted in anisotropic density maps because of preferred orientation; therefore, "Rebalance 2D classes" was executed with two different rebalance factors, resulting in 320,816 and 277,162 remaining particles, respectively. These were subjected to heterogeneous 3D refinement with three classes each. In both cases, the 3D class with the least pronounced preferred particle orientation also showed the most well-defined head and upper leg parts. Particles from these two classes were combined, duplicates removed (101,741 remained), and subjected to homogeneous refinement, followed by non-uniform refinement. The headpiece of the density (3.8 Å overall resolution) was well-resolved, whereas the leg piece was fragmented. Therefore, we subtracted the density of the leg piece and carried out local refinement of the remaining headpiece, followed by local CTF refinement. The resolution of the headpiece density map was 3.7 Å according to the gold-standard Fourier shell correlation (FSC) criterion (Supplementary Fig. 6D, E, G). DeepEMhancer[127] was applied for map sharpening. For the leg piece, homogeneous refinement of 277,162 particles after "Rebalance 2D classes" yielded a density map (4.3 Å resolution) in which headpiece and leg piece were equally pronounced. Here, we subtracted the signal of the headpiece and carried out local refinement of the remaining leg piece, yielding a resolution of 4.7 Å according to the gold-standard FSC criterion (Supplementary Fig. 6D, F, H). This map was filtered according to local resolution. Finally, the individually processed maps of headpiece and leg piece were fitted into the 4.3 Å global map of $\alpha_5\beta_1$ integrin.

### Atomic modeling of $\alpha_5\beta_1$ integrin

A previously generated homology model of the extracellular domain of rat $\alpha_5\beta_1$ integrin (residues 94–1041 and 23–719, respectively)[86] based on human $\alpha_5\beta_1$ integrin (PDB 7NXD)[87] was used as the starting model. The homology model was split into headpiece and leg piece ($\alpha_5$, residues 94 – 691 and 692 – 1041; $\beta_1$, residues 23–504 and 505–720, respectively) and rigid-body fitted into the sharpened density maps of headpiece and leg piece using UCSF Chimera[128], followed by flexible fitting with imodfit[129]. The model of the headpiece was then manually adjusted in Coot[130] and ISOLDE[131], and refined by real-space refinement in Phenix[132] in an iterative manner (Supplementary Fig. 6K, L). Cryo-EM data processing and model refinement statistics are summarized in Supplementary Table 1.

### Glycan attachment to the atomic model with GlycoShield

Glycans were added to the glycosylation sites in $\alpha_5\beta_1$ integrin using the reductionist molecular dynamics simulation method described in GlycoSHIELD[92]. Glycans were added as identified in ref. 86 ($\alpha_5$ integrin, biantennary at N136, N231, N356, N642, N761, N773, N918, triantennary at N822, hybrid at N346, high mannose at N365, N657, N724; $\beta_1$ integrin, complex biantennary at N50, N97, N212, N269, N363, N406,

N417, N482, N521), and models were obtained in coarse grain mode. 100 conformations for every glycan were then combined on the $\alpha_5\beta_1$ integrin model.

### Preparation of peptidisc-embedded $\alpha_5\beta_1$ integrin-Gal3 complexes for cryo-EM

$\alpha_5\beta_1$ integrin was reconstituted in peptidiscs as described above and immobilized onto cobalt beads by overnight incubation. Cobalt beads were then washed and further resuspended in 500 µl of HEPES buffer supplemented with sialidase according to conditions described earlier. Beads were washed three times with 500 µl of HEPES buffer and further incubated for 2 h at 21 °C on a rotating wheel with 4 µM of Gal3. Beads were washed three times with 500 µl of HEPES buffer, and $\alpha_5\beta_1$ integrin-Gal3 complexes were rapidly eluted for 10 min with 40 µl of HEPES buffer supplemented with 10 mM EDTA, with manual and gentle resuspension of beads. Eluted complexes were immediately used for subsequent negative stain and cryo-EM analysis.

### Cryo-EM of peptidisc-embedded $\alpha_5\beta_1$ integrin-Gal3 complexes

Immediately after elution from cobalt beads, samples were vitrified on Quantifoil 2/1 Cu 300 mesh grids using a Vitrobot Mark IV set to a blot force of −1, blotting time of 3.5 s, 100% humidity, and temperature of 12 °C. Micrographs were acquired using a FEI Titan Krios G3i microscope (Thermo Fisher Scientific) operated at 300 kV equipped with a Bioquantum K3 direct electron detector and energy filter (Gatan) running in CDS super-resolution mode at a slit width of 20 eV and at a nominal magnification of 81,000×, giving a calibrated physical pixel size of 1.06 Å/px. Movies were recorded for 3.0 s, accumulating a total electron dose of 61 e⁻/Å² fractionated into 60 frames. EPU 2.12 was utilized for automated data acquisition with AFIS enabled using a nominal defocus between −1.0 and −2.8 µm. Two datasets (6028 and 10,325 movie images, respectively) were recorded under identical conditions and merged after the initial processing steps.

### Data processing of peptidisc-embedded $\alpha_5\beta_1$ integrin-Gal3 complexes

Drift correction and estimation of CTF parameters of movies of both datasets were carried out in cryoSPARC (version 3.3.2) using the built-in patch motion correction and patch CTF estimation algorithms. After sorting out bad images, 5614 and 10,225 micrographs remained from the datasets for subsequent processing steps, respectively. Using the blob picker on 1000 images of the first dataset and subsequent 2D classification in cryoSPARC, autopicking templates were generated and used to pick 1,327,764 and 3,894,571 particles from the datasets, respectively. These were subjected to two subsequent rounds of 2D classification with 8× and 4× binning (pixel sizes of 4.24 and 2.12 Å/pixel, respectively), leaving 289,613 and 556,786 integrin-shaped particles, respectively. These were combined for the subsequent processing steps. Non-uniform refinement[133] in cryoSPARC with 2x binned data (1.06 Å/pixel) using an initial model generated from the second dataset resulted in a density map with 4.1 Å overall resolution, according to the gold-standard FSC criterion. Different 3D sorting approaches in cryoSPARC (version 3.3 and 4.0.3) and Relion (version 4.0)[134] were applied, with the best final results achieved by 3D classification in cryoSPARC (version 4.03) in combination with starting models generated by previous 3D classifications in Relion and a focused mask on the space between headpiece and leg piece. Transfer of particles between cryoSPARC and Relion was performed by pyem (DOI: 10.5281/zenodo.3576630). Two rounds of 3D classification were carried out with four classes each, the first round in input mode, the second round in simple mode. For the first round, the 3D map originating from all particles (see above) and three different 3D maps originating from Relion 3D classifications served as input models, low-pass filtered to 20 Å resolution. 387,525 particles from the two classes with the most pronounced densities outside $\alpha_5\beta_1$ integrin were

transferred to a second round of 3D classification in simple mode. This resulted in two classes with 111,868 and 112,569 particles, respectively, that showed more density outside $\alpha_5\beta_1$ integrin than the others. The class with 111,869 particles (termed subset 1) that has the most pronounced extra density was then refined (non-uniform refinement), resulting in a final resolution of 6.9 Å, according to the gold-standard FSC criterion (Supplementary Fig. 8A, D, E).

### Preparation of nanodisc-embedded $\alpha_5\beta_1$ integrin-Gal3 complexes for cryo-EM

$\alpha_5\beta_1$ integrin was reconstituted in GSLs-containing nanodiscs as described above (4 reconstitution reactions) and immobilized onto cobalt beads by overnight incubation. Bead-immobilized nanodiscs were resuspended in 500 µl of HEPES buffer and incubated with sialidase on a rotating wheel according to conditions described earlier. Beads were washed three times with 500 µl of HEPES buffer and further incubated for 2 h at 21 °C on a rotating wheel with 4 µM of Gal3. Beads were then washed three times with 500 µl of HEPES buffer, and $\alpha_5\beta_1$ integrin-Gal3 complexes were rapidly eluted for 10 min with 40 µl of HEPES buffer supplemented with 10 mM EDTA, with manual and gentle resuspension of beads. Eluted complexes were immediately used for subsequent negative stain and cryo-EM analysis.

### Cryo-EM of nanodisc-embedded $\alpha_5\beta_1$ integrin-Gal3 complexes

Immediately after elution from cobalt beads, samples were vitrified on Quantifoil 1.2/1.3 Cu 300 mesh grids using a Vitrobot Mark IV set to a blot force of 0, blotting time of 3.0 s, 100% humidity, and temperature of 8 °C. Micrographs were acquired using a FEI Titan Krios G3i microscope (Thermo Fisher Scientific) operated at 300 kV equipped with a Bioquantum K3 direct electron detector and energy filter (Gatan) running in CDS super-resolution mode at a slit width of 20 eV and at a nominal magnification of 81,000×, giving a calibrated physical pixel size of 1.06 Å/px. Movies were recorded for 1.7 s, accumulating a total electron dose of 80.3 e⁻/Å² fractionated into 42 frames. EPU 2.12 was utilized for automated data acquisition with AFIS enabled using a nominal defocus between −1.3 and −2.6 µm. Three datasets (7188, 5139, and 9695 movie images, respectively) were recorded under identical conditions. Particles of them were merged after the initial processing steps.

### Data processing of nanodisc-embedded $\alpha_5\beta_1$ integrin-Gal3 complexes

Drift correction and estimation of CTF parameters of movies of both datasets were carried out in cryoSPARC (version 4.2.1) using the built-in patch motion correction (bin factor 2 of super-resolution movies) and patch CTF estimation algorithms. After sorting out bad images, 6831, 5082, and 9526 micrographs remained from the datasets for subsequent processing steps, respectively. With the blob picker on the first dataset and subsequent 2D classification in cryoSPARC, autopicking templates were generated and used to pick 1,393,918, 1,198,669, and 2,149,500 particles from the three datasets, respectively. These were extracted with 2.35× binning (pixel size of 2.48 Å/pixel, respectively) and subjected to two subsequent rounds of 2D classification and one hetero refinement, leaving 330,842, 589,145, and 1,081,636 integrin-shaped particles, respectively. Moreover, the cryoSPARC blob picker was used on all micrographs from the three datasets, resulting in 1,626,764, 1,482,629, and 2,983,868 particles, respectively. These were extracted and processed in 2D classification and 3D hetero refinement as described for the template-picked particles, resulting in 279,906, 636,367, and 1,153,552 integrin-shaped particles, respectively. All particles from the two picking methods were combined, duplicates removed, and extracted unbinned (320 px box size) for the subsequent processing steps. A non-uniform refinement[133] in cryoSPARC resulted in a density map with 3.9 Å overall resolution, according to the gold-standard FSC criterion (Supplementary Fig. 7C). 3D variability

analysis[135], hetero refinement and focused 3D classification (simple mode, hard classification activated) approaches in cryoSPARC (version 4.2.1) were carried out and combined as outlined in Supplementary Fig. 7C. Particles from several 3D classes with most pronounced density between head and leg of $\alpha_5\beta_1$ integrin were combined and duplicates removed to yield 104,728, 21,425, and 41,848 particles for density maps corresponding to a bound Gal3 dimer, trimer, and tetramer, respectively, with final resolutions of 7.4, 8.5, and 7.5 Å, respectively, according to the gold-standard FSC criterion (Supplementary Fig. 7D, E).

## Gal3 modeling on $\alpha_5\beta_1$ integrin

For fitting Gal3 into the map of the nanodisc-embedded $\alpha_5\beta_1$ integrin-Gal3 complexes, the head and leg piece maps of $\alpha_5\beta_1$ integrin, including the fitted atomic models, were placed into the respective domains of the nanodisc-embedded complex maps first. Subsequently, Gal3 CRDs (PDB 1KJL)[136] were placed into well-defined densities between head and leg pieces. For the map of $\alpha_5\beta_1$ integrin-Gal3 dimeric complex, the density is compatible to fit 2 Gal3 CRDs that interacted with glycans at $\alpha_5$-N356 and $\alpha_5$-N918, or at $\beta_1$-N417 and $\alpha_5$-N356. For the $\alpha_5\beta_1$ integrin-Gal3 trimeric complex, the density is compatible to fit 3 Gal3 CRDs that interact with glycans at $\alpha_5$-N356, $\alpha_5$-N918, and at $\beta_1$-N417. Gal3 CRDs were manually placed in densities outside the $\alpha_5\beta_1$ integrin model such that the following requirements were fulfilled: (i) match of galactose in the Gal3 PDB with terminal galactose of glycan chains ($\alpha_5$-N356 and $\alpha_5$-N918, or $\alpha_5$-N918 and $\beta_1$-N417), (ii) close spatial proximity of Gal3 K227 with $\alpha_5$-K965/K1028 (Gal3 contacting glycan at $\alpha_5$-N918), or Gal3 K227 with $\beta_1$-K100/K122/K130 (Gal3 contacting glycan at $\beta_1$-N417), (iii) N-terminal ends of Gal3 CRDs protruding to the same directions to allow for oligomerization via the N-terminal domain, (iv) no overlap/clashes of Gal3 CRDs with each other and the $\alpha_5\beta_1$ integrin model. In the cryo-EM density map corresponding to $\alpha_5\beta_1$ integrin-Gal3 tetrameric complex, 1 further Gal3 CRD was fitted into a further density between headpiece and leg piece without being modeled on a terminal galactose of glycan chains, in agreement with points (iii)−(iv) above.

For fitting Gal3 into the map of the peptidisc-embedded $\alpha_5\beta_1$ integrin-Gal3 dimeric complexes, the same methodology was applied to fit two Gal3 CRDs that interacted with glycans $\alpha_5$-N356 and $\alpha_5$-N918, or $\alpha_5$-N356 and $\alpha_5$-N773.

For representation of the glycan distribution on active extended ligand-bound $\alpha_5\beta_1$ integrin (Fig. 9H), $\alpha_5$ and $\beta_1$ integrin headpiece and leg piece of the glycosylated homology model were rigid-body fitted into the cryo-EM density of the active extended ligand-bound conformation of human $\alpha_5\beta_1$ integrin (EMDB 12634, working map including the low-resolution parts)[87]. Gal3 CRDs were placed on glycans at $\alpha_5$-N356, $\alpha_5$-N918, and $\beta_1$-N417 (see above).

## Micellar $\alpha_5\beta_1$ integrin-Gal3 complex cross-linking and mass spectrometry analysis

$\alpha_5\beta_1$ integrin at 40 µg/mL and His-tagged Gal3 (Gal3-His) at 400 nM in 500 µl HEPES/Triton X-100 buffer were co-incubated for 20 min on a rotating wheel. Samples were chilled for 10 min on ice, and $\alpha_5\beta_1$ integrin-Gal3-His complexes cross-linked for 2 h at 4 °C using a 40 µM BS3-d0/-d4 mixture (20 µM each), or 100 µM DSBU. Excess cross-linkers were quenched for 20 min on ice with 20 mM Tris, pH 7.4. Gal3 and excess cross-linkers were removed by loading samples on 40 kDa cut-off 2 ml desalting spin-columns equilibrated with HEPES/Triton X-100 buffer. Two reactions as described above were pooled and incubated with 25 µl bed volume of cobalt beads prepared according to the manufacturer's instructions. Beads were washed three times with 500 µl HEPES/Triton X-100 buffer. After the last wash, beads were resuspended in HEPES/Triton X-100 buffer, supplemented with deglycosylation denaturing buffer and boiled for 5 min at 95 °C. Samples were chilled for 10 min on ice and supplemented with 1% (v/v)

NP40. Deglycosylation was achieved upon addition of 2 µl of PNGase F and incubation for 2 h at 37 °C. Reactions were stopped by the addition of 3x non-reducing SDS sample buffer and boiling for 5 min at 95 °C. For biochemical characterization of cross-linked complexes, samples were loaded on SDS-PAGE and proteins were detected in the Stain-Free™ mode. For mass spectrometry, samples were run on SDS-PAGE at 100 V until the migration front reached half of the 4% acrylamide stacking gel. Proteins were fixed by incubating gels for 30 min at room temperature in ethanol/acetic acid 50/3 (v/v) solution. After 3 washes with ultra-pure water, gels were stained for 1 h at room temperature with Coomassie blue solution. Stained protein bands that remained after washing in water were cut, washed with 25 mM $NH_4HCO_3$ before reduction and alkylation with 10 mM DTT and 55 mM iodoacetamide. In-gel digestion was performed overnight with trypsin. Peptides were recovered and injected on a nanoLC-MS/MS system. XL-BSA was used as a quality control. NanoLC-MS/MS analyses were performed with a nanoAcquity UPLC device coupled to a Q-Exactive HF-X mass spectrometer. Peptide separation was performed on an Acquity UPLC BEH130 C18 column (250 mm × 75 µm with 1.7-µm-diameter particles) and a Symmetry C18 precolumn (20 mm × 180 µm with 5-µm-diameter particles, Waters). The solvent system consisted of 0.1% formic acid (FA) in water and 0.1% FA in acetonitrile (ACN). The system was operated in data-dependent acquisition mode with automatic switching between MS and MS/MS modes. The ten most abundant ions were selected on each MS spectrum for further isolation. The HCD fragmentation method was used with different collision energies (NCE 27, 30, and 33). The dynamic exclusion time was set to 60 s. Raw data were processed and converted into *.mgf format. The MS/MS data were analyzed using MeroX software v2.0.1.4[137]. Mass tolerances of 5 ppm for precursor ions and of 10 ppm for product ions were applied. A 5% FDR cut-off and a signal-to-noise >2 were applied. Lys and Arg residues were considered as protease sites with a maximum of three missed cleavages. Carbamidomethylation of cysteine was set as a static modification and oxidation of methionine as a variable modification (max. mod. 2). Primary amino groups (Lys side chains and N-termini) as well as primary hydroxyl groups (Ser, Thr, and Tyr side chains) were considered as cross-linking sites. An in-house database comprising rat $\alpha_5$ and $\beta_1$ integrin, human galectin-3, and bovine serum albumin was used. Cross-links composed of consecutive amino acid sequences were ignored. Each cross-linked product automatically annotated with MeroX was manually validated. Two biological replicates with technical triplicate were made for the DSBU experiment (a total of six samples). For BS3, three different experiments were performed. Cross-linked peptides were validated when seen in two out of six experiments and two out of three experiments for DSBU and BS3, respectively. The XL-MS dataset has been deposited to the ProteomeXchange Consortium via the PRIDE partner repository with the dataset identifier PXD041522[138].

## Monitoring of $\alpha_5\beta_1$ integrin's conformational switch on cells

RPE-1 cells were cooled for 10 min on ice, and integrins were activated or not for 1 h at 4 °C with 1 mM $MnCl_2$ and 5 µg/mL of soluble fibronectin mixture. The efficiency of activation was monitored by surface binding and immunostaining using mAb13 (inactive bent-closed) and 9EG7 (active extended) antibodies.

## Gal3-induced clamping of $\alpha_5\beta_1$ integrin in the inactive bent-closed conformational state

*Cellular approach*: RPE-1 cells were cooled for 10 min on ice and incubated for 30 min at 4 °C in the presence or absence of 200 nM Gal3 or Gal3ΔNter. Cells were then washed, and integrins were activated for 1 h at 4 °C with 1 mM $MnCl_2$ and 5 µg/mL of soluble fibronectin mixture. The efficiency of activation was monitored by surface immunostaining using mAb13 or 9EG7 antibody, whose labeling intensity decreased and increased respectively, in the absence of exogenous Gal3 addition. In

vitro *approach using $\alpha_5\beta_1$ integrin in micelles*: 100 μg/mL of sialidase-treated $\alpha_5\beta_1$ integrin in micelles were pre-incubated or not for 3 h at 18 °C with 2 μM of Alexa488-Gal3, and then activated by incubation with 5 mM $MnCl_2$ and 100 μM cRGD. Samples were supplemented with 3x non-reducing SDS (2% SDS) sample buffer, and run for PAGE at 100 V without SDS (semi-native conditions). Alexa488-Gal3 signal was directly detected on gels (Alexa488 channel). Proteins were then transferred onto nitrocellulose membranes and immunoblotted against rat $\beta_1$ integrin. In vitro *approach using $\alpha_5\beta_1$ integrin in peptidiscs*: $\alpha_5\beta_1$ integrin was reconstituted into peptidiscs using peptidisc/peptidisc-His (1/1, w/w) mix as described above. After overnight incubation with cobalt beads, immobilized peptidiscs were resuspended in 500 μl of HEPES buffer and incubated with sialiase on a rotating wheel according to conditions described above. Beads were washed three times with 500 μl of HEPES buffer and further incubated with 4 μM Gal3 for 2 h at 21 °C on a rotating wheel. Unbound Gal3 was then removed before activation of $\alpha_5\beta_1$ integrin according to conditions described above. $\alpha_5\beta_1$ integrin-Gal3 without activation and $\alpha_5\beta_1$ integrin activation without Gal3 incubation were used as control conditions. Beads were washed three times either with 500 μl HEPES buffer (no integrin activation) or HEPES buffer supplemented with 5 mM $MnCl_2$ and 100 μM cRGD (integrin activation). $\alpha_5\beta_1$ integrin was finally eluted from beads with HEPES buffer supplemented with 0.5 M imidazole (±5 mM $MnCl_2$ and 10 μM cRGD) for 30 min under manual shaking. Imidazole was eliminated using 7 kDa cut-off Zeba spin desalting columns equilibrated with HEPES buffer ±5 mM $MnCl_2$ and 10 μM cRGD. Samples were immediately loaded on grids for negative stain EM analysis.

### Expression of human $\alpha_5\beta_1$ integrin N-glycosylation mutants in dKO-MKF cells

N-to-Q substitutions of key N-glycosylation sites of human $\alpha_5$ integrin were designed, and both $\alpha_5$ integrin (wildtype or N/Q mutants) and $\beta_1$ integrin (wildtype) cDNAs (GeneScript, 3.5 μg of each) were co-electroporated into dKO-MKF cells. All experimental procedures on these cells were performed 24 h post-transfection.

### Gal3 and mAb13 or 9EG7 co-binding and uptake assays on dKO-MKF cells expressing wild-type or glycosylation site-mutated human $\alpha_5\beta_1$ integrin

mAb13 or 9EG7 antibody uptake and Gal3/mAb13 or 9EG7 co-binding experiments and confocal imaging were performed as described above, except that for binding conditions, glass coverslips were coated with 10 μg/mL fibronectin for 30 min at room temperature prior to cell seeding.

### Cell adhesion/spreading assay on dKO-MKF cells expressing wild-type or glycosylation site-mutated human $\alpha_5\beta_1$ integrins

dKO-MKF cells expressing wild-type or N-glycosylation site mutants of human $\alpha_5\beta_1$ integrin were detached with PBS complemented with 0.5 mM EDTA. Cells were then allowed to adhere for the indicated times at 37 °C onto fibronectin-coated glass coverslips (see above). mAb13 and 9EG7 antibody binding assays were then performed as described above. Focal adhesions and actin were labeled using anti-vinculin antibody and phalloidin, respectively.

### Statistics and reproducibility

Student's *t*-test was used to compare the differences between two independent groups, one-way analysis of variance (ANOVA) was used for comparisons among three or more groups, and results are presented as the mean values ± SEM. All statistical tests were two-sided. Unless specified in the figure legend, experiments were performed at least three times independently (biological replicates), and representative images and quantifications are shown. Statistical analyses and plots were generated using Prism version 10.1.1 software

(Graphpad Inc.). All graphs and schematic drawings in the article were done using Adobe Illustrator software version 29.4.

### Reporting summary

Further information on research design is available in the Nature Portfolio Reporting Summary linked to this article.

## Data availability

The relevant raw data from each figure or table (in the main manuscript and in the Supplementary Information) are represented by a single Excel file labeled "Source Data File," provided with this paper. The proteomic XL-MS dataset (Supplementary Data 1) has been deposited to the ProteomeXchange Consortium via the PRIDE partner repository[138] with the dataset identifier PXD041522. The Cryo-EM density maps of peptidisc-embedded $\alpha_5\beta_1$ integrin have been deposited in the Electron Microscopy Data Bank (EMDB) under accession codes EMD-17269 (headpiece), EMD-17270 (leg piece), and EMD-54200 (complex with Gal3), respectively. The atomic coordinates of peptidisc-embedded $\alpha_5\beta_1$ integrin have been deposited in the PDB under accession code 8OXZ. The Cryo-EM density maps of nanodisc-embedded $\alpha_5\beta_1$ integrin in complex with Gal3 have been deposited in the EMDB under accession codes [EMD-51027] (Dimer, Supplementary Data 2), EMD-51028 (Trimer, Supplementary Data 3), and EMD-51029 (Tetramer, Supplementary Data 4). The cryo-EM datasets generated in this work are available from the corresponding authors on request. Source data are provided with this paper.

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

## Acknowledgements

The authors acknowledge the Cell and Tissue Imaging core facility (PICT IBiSA), Institut Curie, member of the French National Research Infrastructure France-BioImaging (ANR-10-INBS-04) for help with microscopy, the Recombinant Protein core facility CNRS-Institut Curie for protein purification, the Curie Cytometry core facility, Reinhard Fässler and Jakob Reber (Max Planck Institute of Biochemistry, Munich, Germany) for the dKO-MKF cell line and helpful comments on the manuscript, Guillaume Montagnac and Nathalie Ly (Institut Gustave Roussy) for help with the generation of AP2-mTagGFP CRISPR cell lines, Jean Salamero for help with LLSM experiments and data analysis, Fredrik Zetterberg (Galecto Biotech Inc.) for I3 compound, Ewan MacDonald for helpful discussion, and Henrik Clausen for extremely thoughtful comments on the manuscript. The authors also acknowledge access to electron microscopic equipment at the core facility BioSupraMol of Freie Universität Berlin, supported through grants from the Deutsche Forschungsgemeinschaft (DFG) and the state of Berlin for large equipment according to Art. 91b GG (INST 335/588-1 FUGG, INST 335/589-1 FUGG, INST 335/590-1 FUGG), Tarek Hilal for cryo-EM data collection of α5β1

integrin, and Thiemo Sprink of the Core Facility for cryo-Electron Microscopy (CFcryoEM) of the Charité—Universitätsmedizin Berlin, supported by DFG (INST 335/588-1 FUGG) for cryo-EM data collection of $\alpha_5\beta_1$ integrin-Gal3. Finally, the authors acknowledge the use of resources of the French Proteomic Infrastructure ProFI ANR-10-INBS-08–03. This work was supported by grants from Fondation ARC ARCPGA2024110009062_9628 (L.J.), Mizutani Foundation reference n° 200014 (L.J.), Agence National de la Recherche ANR-19-CE13-0001-01, ANR-20-CE15-0009-01, ANR-22-CE11-0030-03, ANR-25-CE11-0988-01 (L.J.), DALLISH-ANR-16-CE23-0005 (C.A.V.-C., L.L., L.J.), Fondation pour la Recherche Médicale EQU202103012926 (L.J.), LabEx Cell(n)Scale (ANR-11-LABX-0038) as part of the Idex PSL (ANR-10-IDEX-0001-02) (C.A.V.-C., L.L., L.J.), ITMO Cancer (18CQ091) (C.A.V.-C., L.L.), Science Foundation Ireland (19/FFP/6428) and (14/IA/2488) (T.E.K.). RR acknowledges funding from the European Union's Horizon 2020 research and innovation program under the Marie Skłodowska-Curie grant agreement no: 101025342.

## Author contributions

L.J., M.S.-Z., E.D., D.R. are listed for conceptualization; L.J., M.S.-Z., E.D., D.R., I.H., S.R., C.W., V.C., C.A.V.-C., R.R., L.L., D.L., A.H., N.K.S., J.R., U.J.N. for methodology; M.S.-Z., E.D., I.H., C.A.V.-C., L.L., R.R., D.L., A.D.C., A.H., N.K.S., J.R., R.B. for investigation; M.S.-Z., E.D., D.R., I.H., V.C., D.L., A.D.C., R.R., A.H., N.K.S., J.R. for visualization; L.J. for funding acquisition; L.J. for project administration; L.J., M.S.-Z., E.D., T.E.K., S.C.S., S.R., D.R. for supervision; L.J., M.S.-Z., E.D., D.R. for writing of the original draft; and D.L., C.W., H.L., V.C., R.R., C.A.V.-C., L.L., T.E.K. for review and editing.

## Competing interests

H.L. and U.J.N. are shareholders in Galecto Biotech Inc., a company that develops galectin inhibitors. The other authors declare that they have no conflict of interest with the contents of this article.
