## [Transparent Peer Review file · Nature Communications]

Spatial N-glycan rearrangement on $\alpha 5\beta 1$ integrin nucleates galectin-3 oligomers to determine endocytic fate

Corresponding Author: Dr Ludger Johannes

Version 0:

Reviewer comments:

Reviewer #1

(Remarks to the Author)

In the present study, the authors described how galectin-3 preferentially binds to the $\alpha 5\beta 1$ integrin in a bent-closed state, an inactive form, rather than in an extended-open state, the active form. They also found that these bindings promote endocytic uptake and subsequent retrograde trafficking to the Golgi for cell-polarized distribution, highlighting the importance of glycans and their associated galectins. Although the study was well conducted and the results were precise, some points need further clarification.

Major concerns

1. Many galectins interact with glycans. Why did the authors select galectin-3 for this study? Does the cell line express galectin-3 at high levels?
2. As the manuscript describes, sialylation is crucial for integrin activation and galectin binding. Therefore, the authors must investigate the impact of reduced sialylation of $\alpha 5\beta 1$ on the interaction between galectin-3 and mAb13 or 9EG7. The reduction in sialylation can be achieved by treating cells with sialidases to remove sialylation from the cell surface (such as the treatment with PNGase F in Figure 5) or ST6GAL1- or ST3GAL4-knockout cells. Then, the impacts can be investigated, as shown in Figure 1, though not for all studies.

Minor concerns

1. Some words are underlined, such as on page 37. What do they mean?
2. All Western blots should show the positions of molecular weight markers above and below the bands of interest.

Reviewer #2

(Remarks to the Author)

Overall this is a novel, interesting, and substantive study. The experiments are thorough and rely on a broad and diverse range of state-of-the-art approaches. The authors present compelling evidence showing that distinct conformations of A5B1 integrin modulate galectin binding through the repositioning of N-glycans, thereby differentially influencing internalization and retrograde trafficking. The manuscript distinguishes between the active (ligand-bound) and inactive (ligand-unbound) conformations of the glycoprotein. The authors propose that the active form adopts a structure that allows for segregated galectin binding, preventing oligomer formation. In contrast, the inactive conformation facilitates galectin binding at membrane-proximal regions, promoting oligomer formation. These oligomers promote retrograde trafficking of the inactive form to the Golgi apparatus for the redistribution of the inactive receptor to the leading edge of the cell, where it can engage with ligands. This study provides valuable insights into the molecular mechanisms underlying galectin-3-glycoprotein interactions. While previous research has demonstrated that galectin-3 interactions can drive endocytosis, this manuscript is the first to provide molecular and structural insights at this resolution. The discovery that ligand binding induces spatial rearrangements of glycans opens new avenues for the study of glycobiology and cellular trafficking. While the authors have presented a strong body of evidence to support their hypothesis, addressing the following points would strengthen the manuscript.

Major Comments:

- 1) The study relies heavily on the ability of the two mAbs to distinguish A5B1 conformations. However, alternative hypotheses are not explored or discussed. For example, is it possible that the two antibodies have a preference for different glycoforms of A5B1, and that this is the basis for differences in gal3 binding and trafficking? Or that the two conformations

exist in different PM domains, and that this rather than conformation per se drives gal3 binding trafficking? This should be addressed or discussed in the manuscript.

2) The majority of internalization and trafficking data for mAb13 and 9EG7 are presented in isolation. However, since the cell surface levels of these conformations may vary, normalization to cell surface levels would aid interpretation. If for example, the inactive form is more abundant at the plasma membrane, the observed increased internalization and co-localization with galectin-3 could be attributed to receptor abundance rather than receptor state. The authors should either demonstrate that the cell surface signal is identical, or normalize to the PM signal. Related to this point, it is unclear how dramatic changes in internalization rate exist without concordant changes in cell surface levels. For example, the glycosite mutants in Fig7E,G show large differences in internalization, but unchanged or discordant changes in membrane levels (FigS9I,J,N). This should either be explored further, reconciled, or at the very least discussed in the manuscript.

3) The mechanistic connection linking integrin conformation and retrograde trafficking remains unclear. The paper begins with a demonstration that the retrograde trafficking of the inactive and active conformations to the Golgi/TGN differ. It is this rather than internalization/uptake that appear robustly distinct between the conformations. However, when the glycosite mutants are explored at the end of the paper, only internalization data is presented. It is unclear how or whether this ties into retrograde trafficking.

4) The role of N-glycan LacNAc/branching is not sufficiently examined. While some experiments examining the role of glycolipids are performed, N-glycan branching is not convincingly targeted pharmacologically/genetically, particularly for the gal3 oligomerization experiments. Is A5B1 integrin needed for PM-driven gal3 oligomerization? If as expected, A5B1 KO/mut cells can drive gal3 oligomerization, are glycolipids and/or N-glycans required? Related to this broad concern, the PNGase F experiments in Fig5 and FigS4 need additional evidence. PNGase F works most efficiently for denatured proteins. Live cell treatment is likely to only partially remove N-glycan and may be biased toward certain structures. Without demonstrating efficient (>90%) removal of branched N-glycans, the interpretation of these data is problematic, since galectin-ligand binding is known to be density/multivalency/avidity dependent. Thus it is expected that the monomer and oligomer forms of gal3 may have different thresholds of N-glycan density for efficient binding. The authors should either demonstrate that both high mannose (ConA) and branched (PHA-L) glycans are efficiently removed by their PNGase treatment (>90%), and/or repeat the experiments with effective inhibitors of N-glycosylation (e.g. kifunensine), while also demonstrating effectiveness with PHA-L.

5) No functional data are presented demonstrating the physiological relevance of the proposed mechanism to cell/organism function. In particular, the glyco-site mutants that appear to specifically impact the inactive conformation would be expected to result in a measurable change on adhesion/shape/motility. This would be particularly helpful given the discrepancy between internalization rates and cell surface levels noted in point#2 above.

Minor Comments:

- For the SNAP-BG experiments, the entire WB should be supplied. The blots should demonstrate a molecular weight shift when (for example) GalT-GFP-SNAP captures mAb13-BG.
- Is there evidence that the integrins themselves are present in the TGN rather than just the conformation-specific antibodies? Is it possible that the antibodies trafficked to the TGN but not integrin? In the SNAP-GFP pulldown experiments, can you also blot for integrin to demonstrate this?
- Antibodies are known to have N-glycans. Are the antibodies (mAb13, mAb16, 9EG7, SNAP51) free from nonspecific binding to Gal3? Treating them with PNGase could help eliminate potential interactions.
- CHC knockdown may affect trafficking of newly synthesized ITGA5/ITGB1, potentially reducing integrin levels at the PM. The authors should clarify whether this effect was considered in their data quantification.
- CBD treatment reduces retrograde trafficking, but internalization was assessed over just 10 minutes. Extending this time frame could confirm Golgi localization and validate CIE in retrograde trafficking.
- Variation in Galectin Staining: In Fig. 7D, there's a noticeable difference in Galectin staining between conditions in the WT image. The authors should clarify the cause of this variation.
- Site-Specific Mutants for Calf Region Glycosites: The authors refer to N918 as the critical leg region glycan extensively in the text. However, they only present data where all four calf region NXST sites are mutated (Δ MT). It is unclear why individual calf sites have not been examined. At the very least this should be clarified in the text and references to N918 tempered.
- Figure 5E needs statistical analysis.
- In Fig6I, some of the structures in yellow circles look bent at first glance. Please confirm that they are not mislabeled.

Reviewer #3

(Remarks to the Author)

In this work the authors investigate the endocytosis pathway of integrin $\alpha 5 \beta 1$ in mammalian cells. They use conformation specific antibodies to show that only the inactive bent closed conformation undergoes retrograde trafficking from the plasma membrane to the golgi, and that this is mediated by Gal3 (rather than clathrin). The authors show that this endocytosis of the inactive bent close $\alpha 5 \beta 1$ integrin is preferentially occurring via elongated CLIC compartments and dependent on dynein, both previously described hallmarks of this clathrin-independent endocytic pathway. Biochemical experiments using purified integrins are used to show that only this inactive bent close conformation complex nucleates the formation of Gal3 oligomers. Similar Gal3 oligomers are then found to be formed on the surface of cells. Using cryoEM, the authors present a structural characterization of Gal3 oligomers in interaction with the inactive bent close conformation of the integrin complex. Finally mapping glycan sites on these structures, the authors propose a model in which 3 glycans located near Gal3 binding sites and close to each other in the bent closed conformation only, could act as the trigger for Gal3 oligomerization. Removing these glycans results in decreased Gal3 integrin colocalization on cell plasma membranes and decreased internalization of the inactive bent closed integrin complex.

The manuscript is very interesting and well written, and the data is rich. In my view the part that is less convincing or more

ambiguous as such is the final structural interpretations, on which the “glycoswitch” part of the model is based. My impression is that the attractive model put forward by the authors is one possibility, that does not disagree with the data presented here. However, as such, it seems difficult to exclude alternative interpretations. My recommendation for publication would be to repeat a couple of endocytosis assays used in the first figures of the paper and/or Gal3 NS oligomerization analysis including: i) glycan mutants for the sites proposed to be involved in the glycoswitch, as well as ii) mutants for a couple of other glycosylation sites, proposed not to be involved in the glycoswitch, as important controls. Alternatively, the glycoswitch claim could be downgraded in the proposed model.

Specific points

Maybe clarify the fate of integrin complex internalised by clathrin for a broader readership.

Fig. 1: colIP on Vps35: is Gal-3 also found in the IP? How is this IP affected when the cells are treated with I3?

Fig. 2: It seems that the more exogenous recombinant Gal3 is added to the system, the more integrin uptake and retrograde transport is observed. Does it mean that Gal3 is limiting and that the amount of Gal3 is enough to regulate the process? Is there any corresponding evidence maybe that Gal3 levels are regulated in human cells (or diseases in the case of imbalance etc.).

Fig. 3A: it would be useful to mention the approximate number of HRP positive structures that were counted in each of the 4 experiments (is it ~10 or 100 or 1000?)

Figure 4I:

What about Gal3 wild type alone, without any integrin nor I3 (not even to reproduce “elution conditions”), at 4uM? Does it oligomerise as seen by NS?

Figure 6H: The authors test the stabilization, upon Mn²⁺ activation in presence or absence of Gal3, of the inactive bent close conformation by checking binding of mAb13, but what happens to the binding of 9EG7?

Also does NS provide evidence of where, on the integrin complex, each antibody binds and how the binding may interfere with Gal3?

In the text related to Figure 7, p 10 | 13, it reads to me as the structural analysis points to the importance of the membrane proximal glycans only, rather than the distal ones. However, 2 distal sites are then also tested and included in the glycoswitch model. So it may be useful to rephrase and clarify this sentence.

Despite efficient cell surface expression of the deltaMP mutant, the overlap with Gal3 is lost and the authors conclude: “This striking result suggests that the removal of the membrane-proximal glycosylation sites on the $\alpha 5$ chain diminishes the capacity of bent closed $\alpha 5 \beta 1$ integrin to trigger Gal3 oligomerization”. I think this needs some clarification. The direct conclusion from this experiment seems to be that deltaMP and Gal3 do not colocalise, hence do not interact. This means that the problem could occur before any possible Gal3 oligomerization, by loosing binding in the first place.

Figure 7D-F: Here it seems that removing any of the glycans tested (leg and head pieces) has an effect. In order to convince the reader that the effect observed is indeed specific to removing these 3 glycans, it would be important to use mutants for other glycan sites and show that these have no effect. This would be needed to support the glycoswitch model.

Based on Figure S9C it seems that the integrin complex is quite covered in glycans and hence even in an open conformation it seems that a couple of glycans might still be close enough to bind several Gal3. To visually convince the reader that this is maybe unlikely to occur, a figure similar to S9C plotting all the glycans in the extended conformation of the integrin complex would also be useful.

In addition is there any evidence that removing these glycans does not affect the conformational equilibrium of the integrin complex in the first place?

Importantly, it looks like at this resolution the Gal3 CTDs could be equally fitted facing in one direction or the other. The translucent red part is proposed to be the part where these multimerization NTDs join but these Nter are long unstructured regions and the density could also be a partial / substoechiometric / more flexible density for an extra copy of Gal3 CTD for instance (as pentamers were previously described in the literature). At this resolution, it seems difficult to unambiguously conclude. It is probably wiser to say that the structural data can be interpreted in agreement with this hypothesis, rather than claiming that the structural data supports/demonstrates this model. (Including in the discussion “These defined oligomeric structures that orient four Gal3 with glycan binding sites pointing in the same outward direction are striking hallmarks of the Gal3 tetramer that we have discovered.”)

Reviewer #4

(Remarks to the Author)

Version 1:

Reviewer comments:

Reviewer #1

(Remarks to the Author)

This reviewer is satisfied with the responses.

Reviewer #2

(Remarks to the Author)

We appreciate the authors for their detailed and thoughtful responses. The authors have clearly made a sincere effort to respond comprehensively to all the points raised. Most of the concerns have been addressed satisfactorily, either through additional experimental data, corrections, further discussion, or by citing relevant previously published literature. Although broad questions remain, they are reasonably beyond the scope of the current paper and the authors' findings are appropriately contextualized. The new data shown on page 11 of the response to reviewers document (demonstrating PNGase F efficacy) should be incorporated into the manuscript. We otherwise support acceptance and publication of the manuscript.

Reviewer #3

(Remarks to the Author)

The authors have added a number of experiments to rigorously address my points and adjusted the manuscript accordingly. I congratulate the authors on this very rich piece of work and recommend the manuscript for publication.

Reviewer #4

(Remarks to the Author)

We would like to thank the Reviewers for their insightful comments that have helped to further improve our manuscript. The original comments are shown below in black, and our responses in green.

In most cases, newly obtained data have been integrated into the revised manuscript. In a few cases, we decided to show them “for Reviewers’ eyes only” in this rebuttal letter. The reasons for that are explained in each of these cases.

REVIEWER COMMENTS

Reviewer #1 (Remarks to the Author):

In the present study, the authors described how galectin-3 preferentially binds to the $\alpha_5\beta_1$ integrin in a bent-closed state, an inactive form, rather than in an extended-open state, the active form. They also found that these bindings promote endocytic uptake and subsequent retrograde trafficking to the Golgi for cell-polarized distribution, highlighting the importance of glycans and their associated galectins. Although the study was well conducted and the results were precise, some points need further clarification.

Major concerns

1. Many galectins interact with glycans. Why did the authors select galectin-3 for this study? Does the cell line express galectin-3 at high levels?

Our study is focused on β_1 integrin, for which functional links with galectin-3 have been documented earlier: (i) Furtak et al. identified galectin-3 as a key extracellular molecule to drive β_1 integrin internalization through a by then unknown molecular mechanism (doi: 10.1006/bbrc.2001.6064); (ii) Priglinger et al. have shown that galectin-3 binds and clusters β_1 integrin in RPE-1 cells in a glycan interaction-dependent manner to modulate cell behavior (doi: 10.1371/journal.pone.0070011); (iii) galectin-3 is crucial for the homeostasis of RPE-1 cells and plays a key role in pathophysiological processes (doi: 10.1371/journal.pone.0070011; doi: 10.1074/mcp.M116.066381; doi: 10.3390/ijms242115516); (iv) we have identified $\alpha_5\beta_1$ integrin as a functional galectin-3 interactor using proteomics analysis (doi: 10.1038/ncb2970) , which was further validated by later pull down assays (doi: 10.1074/mcp.M116.066381). Following reviewer’s question, we have now included a sentence on this important point (page 4, lines 25-27).

In addition, we have newly obtained Western blot data to document the expression of endogenous galectin-3 in the different cell lines that were used in our study (RPE-1, HeLa and dKO-MKF).

These findings have been integrated into the revised manuscript (new panel A of Figure S1).

2. As the manuscript describes, sialylation is crucial for integrin activation and galectin binding.

Integrin function is sensitive to sialylation, as we have indeed mentioned in our manuscript. Recently, we published an extensive study in which we have uncovered a mechanisms by which growth factors (notably EGF) induce desialylation of cell surface glycoproteins, including β_1 integrins, leading to enhanced galectin-3 binding and GL-Lect driven endocytosis in relation to cell migration and bone remodeling (doi: 10.1038/s41556-025-01616-x; cited in the current manuscript, e.g., page 5, line 8).

Therefore, the authors must investigate the impact of reduced sialylation of $\alpha_5\beta_1$ on the interaction between galectin-3 and mAb13 or 9EG7. The reduction in sialylation can be achieved by treating cells with sialidases to remove sialylation from the cell surface (such as the treatment with PNGase F in Figure 5) or ST6GAL1- or ST3GAL4-knockout cells. Then, the impacts can be investigated, as shown in Figure 1, though not for all studies.

This was a great suggestion. We have used neuraminidase from *Arthrobacter ureafaciens* (Roche, Ref.10269611001) to remove all types of sialic acids from the surface of RPE-1 cells. With the pan sialic acid-specific lectin SiaFind (Lectenz Bio SK0501F) we show that desialylation indeed was achieved. In this condition, galectin-3 binding was more than 60% increased, in agreement with our published findings (doi: 10.1038/s41556-025-01616-x). These data are shown below and in new panel G of Figure S2.

As suggested by the Reviewer, we have then used the neuraminidase protocol (new panel F of Figure 2; see below) to investigate the impact of sialic acids on galectin-3's interaction with bent-closed inactive (mAb13) and with extended-open active (9EG7) $\alpha_5\beta_1$ integrin. Both antibodies do not as such affect the increased binding of galectin-3 to neuraminidase treated cells (new panel G of Figure 2; see below). In contrast, it was only for $\alpha_5\beta_1$ integrin's bent-closed conformer (mAb13) that increased levels were pulled down together with galectin-3 (new panel H of Figure 2; see below). This finding is consistent with the dichotomy theme of the whole study according to which only the bent-closed conformer of $\alpha_5\beta_1$ integrin interacts efficiently and functionally with galectin-3. These new data are now presented on page 5, lines 5-15.

Minor concerns

1. Some words are underlined, such as on page 37. What do they mean?

The underlining of sub-headings has been transformed into *italic style*.

2. All Western blots should show the positions of molecular weight markers above and below the bands of interest.

Molecular weight markers are now shown throughout. Please note that for all Western blots uncropped versions with molecular weight markers are provided as source data as part of the revised manuscript files.

Reviewer #2 (Remarks to the Author):

Overall this is a novel, interesting, and substantive study. The experiments are thorough and rely on a broad and diverse range of state-of-the-art approaches. The authors present compelling evidence showing that distinct conformations of A5B1 integrin modulate galectin binding through the repositioning of N-glycans, thereby differentially influencing internalization and retrograde trafficking. The manuscript distinguishes between the active (ligand-bound) and inactive (ligand-unbound) conformations of the glycoprotein. The authors propose that the active form adopts a structure that allows for segregated galectin binding, preventing oligomer formation. In contrast, the inactive conformation facilitates galectin binding at membrane-proximal regions, promoting oligomer formation. These oligomers promote retrograde trafficking of the inactive form to the Golgi apparatus for the redistribution of the inactive receptor to the leading edge of the cell, where it can engage with ligands. This study provides valuable insights into the molecular mechanisms underlying galectin-3-glycoprotein interactions. While previous research has demonstrated that galectin-3 interactions can drive endocytosis, this manuscript is the first to provide molecular and structural insights at this resolution. The discovery that ligand binding induces spatial rearrangements of glycans opens new avenues for the study of glycobiology and cellular trafficking.

We would like to thank the Reviewer for their supportive comments.

While the authors have presented a strong body of evidence to support their hypothesis, addressing the following points would strengthen the manuscript.

Major Comments:

1) The study relies heavily on the ability of the two mAbs to distinguish A5B1 conformations.

We have indeed used well-characterized conformational state-specific antibodies against β_1 integrin (doi: 10.1242/jcs.056770; doi: 10.1101/cshperspect.a004994). These allow the detection of ligand-bound (9EG7 mAb) and non-ligand-bound (mAb13 mAb) conformers of β_1 integrin, that correspond to the extended-open versus bent-closed conformations, respectively.

However, the originality of our study is to combine cell-based work using these conformational state-specific antibodies, with in vitro approaches based on purified and conformationally tunable $\alpha_5\beta_1$ integrin. It is the convergent lines of arguments that originate from both types of approaches that constitute the real strength of our findings.

However, alternative hypotheses are not explored or discussed. For example, is it possible that the two antibodies have a preference for different glycoforms of A5B1, and that this is the basis for differences in gal3 binding and trafficking?

As for any piece of science, alternative interpretations are always possible, and we fully acknowledge this. For reasons that are developed below (including new experimental

data), we strongly believe that in this specific case, the glycoform hypothesis is rather unlikely.

Timothy Springer used negative stain EM on $\alpha_5\beta_1$ integrin to show that (i) mAb13 binding to an epitope located at the β_1 domain of β_1 integrin stabilizes the closed conformation, whereas (ii) 9EG7 binding to an epitope located at the EGF-repeat leg domain of β_1 integrin stabilizes the extended conformation (doi: 10.1073/pnas.1605074113 ;doi: 10.1074/jbc.271.34.20365). These studies established a tight link between both antibodies and their conformational target specificity. This as such does not exclude the glycoform hypothesis, but constitutes a solid basis for the use of these antibodies as conformer sensors.

Importantly, we also used conformational state-specific antibodies that target the α_5 integrin subunit: SNAKA51 recognizes an epitope at exposed calf domains, conformationally converting $\alpha_5\beta_1$ integrin into a ligand competent primed/extended form (doi: 10.1242/jcs.01623); mAb16 binds to an epitope located at the β -propeller domain, keeping the integrin resistant to RGD-activation, most likely in the closed conformation (PMC1220672). As reported in Figure S2A, mAb16 and SNAKA51 phenocopy mAb13 and 9EG7, respectively. The fact that 2 sets of antibodies that share conformer specificity but recognize different integrin chains produce the same results makes it highly likely that the common denominator is the capacity to molecularly recognize different conformational states of the integrin, and not a hypothetical preference for different glycoforms of the protein.

In a new set of experiments, we now report that upon incubation of RPE-1 cells with $MnCl_2$ and fibronectin (FN), an increase of 9EG7 labeling (new panel E of Figure S8; see below) was mirrored by the concomitant reduction of mAb13 labeling (new panel F of Figure S8; see below). Importantly, the $MnCl_2$ /FN incubations were done at 4 °C, which excludes the possibility of glycoform remodeling by $\alpha_5\beta_1$ integrin trafficking to intracellular sites where glycosyltransferases are localized. The most reasonable interpretation of these data therefore appears to be that the same $\alpha_5\beta_1$ integrin molecules (i.e., the same glycoform(s) of the integrin) are conformationally switched. An updated section is now provided on page 10, lines 9-11.

Of note, in the set of negative stain EM data of Figure 6I (see below), we indeed directly demonstrate that the $MnCl_2$ and RGD (minimal functional peptidic sequence from FN)

reagents that we use are able to induce $\alpha_5\beta_1$ integrin's switching from the bent-closed to the extended-open conformation.

We also newly performed bulk neuraminidase treatment on RPE-1 cells to remove sialic acids (desialylation) from cell surface proteins (new panel F of Figure 2; see below). We then investigated the impact of desialylation on galectin-3's interaction with bent-closed inactive (mAb13) or with extended-open active (9EG7) $\alpha_5\beta_1$ integrin. Interestingly, we found that binding capacity of these antibodies almost remained identical upon neuraminidase treatment (new panel H of Figure S2), indicating that these antibodies are at least not sialic acid sensitive, likely supporting a conformational specificity rather than a glycoform one (documented on page 5, lines 13-15).

Both antibodies as such did not affect the increased binding of galectin-3 to neuraminidase treated cells (new panel G of Figure 2, see below; doi: 10.1038/s41556-025-01616-x). In contrast, it was only for $\alpha_5\beta_1$ integrin's bent-closed conformer (mAb13) and not for the extended-open conformer (9EG7) that increased levels were pulled down together with galectin-3 (new panel H of Figure 2, see below; documented on page 5, lines 5-15). This confirms that the recognition by mAb13 and 9EG7 antibodies is highly conformational state specific.

Finally, in our study the antibody-based experiments on cells were complemented by in vitro studies using purified proteins. We indeed demonstrate that the differences in interaction with galectin-3 that we have measured on cells with mAb13 versus 9EG7 antibodies were reproduced in vitro with purified $\alpha_5\beta_1$ integrin that was conformationally switched using $MnCl_2$ and RGD (Figures 4F and 4G). Importantly, in these biochemical experiments, the $\alpha_5\beta_1$ integrin molecules in the bent-closed and extended-open conditions had strictly the same glycan makeup, as they were obtained from the same highly pure protein batch (Figure S4A and doi: 10.1111/boc.202200017). We can therefore exclude the different glycoform hypothesis. It is indeed the power of in vitro approaches to be able to test molecular hypothesis under highly controlled conditions, which is chiefly why we have chosen to base our study on such two-pronged approach.

Or that the two conformations exist in different PM domains, and that this rather than conformation per se drives gal3 binding trafficking? This should be addressed or discussed in the manuscript.

There is no doubt that these two conformers are located in distinct plasma membrane domains: Predominantly ventral for the extended ligand-bound conformer (9EG7) in association with either nascent or fibrillar focal adhesion, and mainly dorsal for the bent-closed non-ligand-bound conformer (mAb13) (doi: 10.1083/jcb.201707075). In our study, the two layered localization of these conformers is particularly visible on polarized cells grown on line micropatterns (Figures S2D and S2E) where the extended-active $\alpha_5\beta_1$ integrin (9EG7) was found at the ventral side with low overlap with galectin-3, and the bent-closed inactive (mAb13) located slightly above with high overlap with galectin-3. However, both subdomains were accessible for exogenously added galectin-3.

In addition, our in vitro binding experiment using purified $\alpha_5\beta_1$ integrin confirmed a preferential binding of galectin-3 to the bent-closed conformer and the capacity of only this conformer to scaffold defined and functional galectin-3 oligomers (Figures 4F-J), unveiling the central role of integrin conformation in this interaction.

By following the Reviewer's recommendation, we have added a sentence on page 4 lines 41-42 to indicate that on cells other components of the membrane domains to which these conformers are localized may also contribute to trafficking.

2) The majority of internalization and trafficking data for mAb13 and 9EG7 are presented in isolation. However, since the cell surface levels of these conformations may vary, normalization to cell surface levels would aid interpretation. If for example, the inactive form is more abundant at the plasma membrane, the observed increased internalization and co-localization with galectin-3 could be attributed to receptor abundance rather than receptor state. The authors should either demonstrate that the cell surface signal is identical, or normalize to the PM signal.

It has been described that in resting conditions integrins are mainly found in the inactive bent-closed conformational state at the plasma membrane, and that they are regulated through acute activation to respond to defined physiological needs (reviewed in doi: 10.1016/s0092-8674(02)00971-6). For example, Johanna Ivaska's group nicely demonstrated that in MDA-MB-231 cells, most of cell surface $\alpha_5\beta_1$ integrin was in the mAb13-positive inactive conformation, and only 20% in the 9EG7-positive active conformation (doi: 10.1111/j.1600-0854.2012.01327.x).

Although less abundant than the inactive form, 9EG7-positive labeling was clearly visible both at the cell surface and upon internalization, but barely overlapped with galectin-3. As a striking example, polarized cells that were grown on line micropatterns showed clearly detectable signal for both conformers at the plasma membrane, and distinct overlap with galectin-3: extensive colocalization for the inactive conformer (mAb13), and almost no colocalization for the active conformer (9EG7) (Figures S2D and S2E).

When considering the possibility of a direct comparison between the conformers, it needs to be kept in mind that the affinity of both antibodies (i.e., mAb13 and 9EG7) for their targets (inactive bent-closed and ligand-bound extended-open $\alpha_5\beta_1$ integrin, respectively) might be different. A direct comparison of fluorescent signal intensities might therefore not be meaningful.

In the light of this limitation, we have based our study on relative quantifications between experimental conditions separately for each conformer. In other words, we have determined for each condition (e.g., I3 inhibitor, GSL depletion...) how a given conformer was affected (e.g., endocytosis, retrograde trafficking...). For this approach, it was important to validate that cell surface levels of each conformer were not affected between conditions, which was indeed not the case (doi: 10.3390/biom14091169).

As mentioned above, we have used a two-pronged approach to nevertheless also be able to make direct comparisons between conformers. With purified $\alpha_5\beta_1$ integrin under controlled in vitro conditions, it was possible to use the same amounts of extended-open or bent-closed conformers. We thereby confirmed that galectin-3 preferentially binds to the bent-closed conformational state, and that it's only this conformer that nucleates functional galectin-3 oligomers (Figure 4F-J). These findings are entirely consistent with the results from the cell-based studies.

Related to this point, it is unclear how dramatic changes in internalization rate exist without concordant changes in cell surface levels. For example, the glycosite mutants in Fig7E,G show large differences in internalization, but unchanged or discordant changes in membrane levels (FigS9I,J,N). This should either be explored further, reconciled, or at the very least discussed in the manuscript.

In new results shown in Figures S8E and S8F, we find that the different conformers of $\alpha_5\beta_1$ integrin can be interconverted at the cell surface. We speculate that glycosite mutants (e.g., Δ MP) whose GL-Lect driven endocytosis is prevented can shift to the active state, thereby explaining the correspondingly increased cell surface levels of this conformer (Figure S9M) and the increase in cell spreading that is observed with cells expressing Δ MP (new panels N and O of Figures S9). As suggested by the Reviewer, a corresponding discussion has been added to the revised manuscript (page 11, lines 30-33).

3) The mechanistic connection linking integrin conformation and retrograde trafficking remains unclear. The paper begins with a demonstration that the retrograde trafficking of the inactive and active conformations to the Golgi/TGN differ. It is this rather than internalization/uptake that appear robustly distinct between the conformations. However, when the glycosite mutants are explored at the end of the paper, only internalization data is presented. It is unclear how or whether this ties into retrograde trafficking.

For the glycosite mutant experiments, we used a mouse kidney fibroblast cell line (doi: 10.1038/ncb2501) in which the α_5 and β_1 integrin genes have been knocked out (dKO-MKF, kind gift from Prof. Dr. Reinhard Fässler). The expression of other integrins such as α_2 , α_3 , α_6 , α_V , β_3 and β_5 (PMID: 26821125) likely explains why these cells are undergoing normal cell growth, adhesion, and spreading. To study glycosite mutants, we transiently transfected corresponding α_5 and β_1 integrin cDNAs into these cells.

For retrograde trafficking, cells need to express the GalT-GFP-SNAP fusion protein, which is not the case for dKO-MKF cells. One way would be to perform triple transient transfections followed by single cell-based immunofluorescence analysis. While α_5 and β_1 integrin cDNA double transfection rates were already very low, triple transfection rates unfortunately turned out to be vanishingly low and therefore quantitative analysis was unachievable in the light of some heterogeneity linked to the expression levels of GalT-GFP-SNAP.

Generating a stable GalT-GFP-SNAP dKO-MKF cell line including the rigorous analysis of GalT-GFP-SNAP localization in function of expression levels is by experience a long-lasting task that is beyond what can be achieved in the time frame of revision. We would therefore keep this approach for follow-up studies.

4) The role of N-glycan LacNAc/branching is not sufficiently examined. While some experiments examining the role of glycolipids are performed, N-glycan branching is not

convincingly targeted pharmacologically/genetically, particularly for the gal3 oligomerization experiments. Is A5B1 integrin needed for PM-driven gal3 oligomerization? If as expected, A5B1 KO/mut cells can drive gal3 oligomerization, are glycolipids and/or N-glycans required?

To address these questions, we first incubated galectin-3 with dKO-MKF cells and used I3 to specifically elute the protein, followed by negative stain EM analysis. Our newly obtained results clearly show that galectin-3 also assembles into defined oligomers in the absence of $\alpha_5\beta_1$ integrin (new panel J of Figure S9, see below).

These data document that $\alpha_5\beta_1$ integrin is not the only membrane protein capable of nucleating Gal3 oligomers. We then analyzed whether N-glycans and glycolipids were necessary to drive Gal3 oligomerization, using PNGase F and Genz treatments, respectively. Our newly obtained results show that both on RPE-1 (new panel I of Figure 5, see below; updated sentence page 8, line 26-28) and dKO-MKF (new panel L of Figure S9, see below; updated sentence page 11, line 5-8) cells, the capacity of galectin-3 to form oligomers is strongly compromised upon N-glycan removal or glycolipid depletion.

Related to this broad concern, the PNGase F experiments in Fig5 and FigS4 need additional evidence. PNGase F works most efficiently for denatured proteins. Live cell treatment is likely to only partially remove N-glycan and may be biased toward certain structures. Without demonstrating efficient (>90%) removal of branched N-glycans, the interpretation of these data is problematic, since galectin-ligand binding is known to be density/multivalency/avidity dependent. Thus it is expected that the monomer and oligomer forms of gal3 may have different thresholds of N-glycan density for efficient binding. The authors should either demonstrate that both high mannose (ConA) and branched (PHA-L) glycans are efficiently removed by their PNGase treatment (>90%), and/or repeat the experiments with effective inhibitors of N-glycosylation (e.g. kifunensine), while also demonstrating effectiveness with PHA-L.

Kifunensine induces high mannose glycoprotein species in the biosynthetic/secretory pathway and thereby has a global effect on glycoprotein maturation and trafficking processes. In contrast, in the context of our experiments, PNGase F has only access to the cell surface pool of already correctly transported proteins. We have therefore chosen to proceed with PNGase F.

As suggested by the Reviewer, we first quantified the effect of PNGase F treatment on RPE-1 and dKO-MKF cells, using Pha-L to detect the efficient removal of branched N-glycans (doi: 10.1016/0304-3835(96)04386-8). A decrease of 64-72% of Pha-L surface signal was obtained under PNGase F treatment condition (see below for Figure). A concomitant reduction of monomeric galectin-3 binding by 70-75% was measured.

These new data have at this stage not been integrated into the revised manuscript, which is already very data heavy. We are of course prepared to do so should the Reviewer think that this was necessary.

We are aware that this 64-72% reduction of Pha-L labeling may not reflect the >90% removal of branched N-glycans that the Reviewer had mentioned. However, for the following 3 reasons, we nevertheless believe that our findings are meaningful.

First, we are again comparing between conditions, rather than absolute values: While plasma membrane binding of monomeric galectin-3 clearly decreased on PNGase F-treated cells, that of preassembled oligomeric galectin-3 was preserved (Figures 5G and 5H). As suggested by the reviewer, we interpret this difference with a passage from monovalent to a multivalent (avidity) mode of binding, which as such is an important

finding. This hypothesis is now discussed in the revised manuscript (page 13, lines 12-17).

Second, the PNGase F-treatment condition has allowed to highlight another striking difference between monomeric and preassembled oligomeric galectin-3: The capacity of the latter to bind to cells in a GSL-dependent manner, as deduced from the fact that its binding capacity to PNGase F-treated cells is strongly reduced upon GSL depletion using Genz (Figures 5G and 5H). Even at the current level of N-glycan removal, the PNGase F condition has thereby allowed to make an important discovery. In as such, oligomeric galectin-3 appears to behave as bacterial Shiga and cholera toxins, whose receptor-binding B-subunits exist as stable oligomers with the capacity to directly bind to GSLs (respectively, Gb3 and GM1).

Third, in newly conducted experiments we found that upon PNGase F treatment, galectin-3 failed to assemble into defined oligomers, both on RPE-1 (new panel I of Figure 5; see page 8, lines 26-28) and dKO-MKF cells (new panel L of Figure S9; see page 11, lines 5-8). Once again, the PNGase F condition has allowed to obtain an important finding, even at the current level of N-glycan removal.

5) No functional data are presented demonstrating the physiological relevance of the proposed mechanism to cell/organism function. In particular, the glyco-site mutants that appear to specifically impact the inactive conformation would be expected to result in a measurable change on adhesion/shape/motility. This would be particularly helpful given the discrepancy between internalization rates and cell surface levels noted in point#2 above.

Following the Reviewer's recommendations, we have newly performed adhesion and spreading assays on dKO-MKF cells expressing glycosite mutants (new panels N-P of Figures S9; see below). Both Δ MP and N307Q mutants showed an enhanced cell spreading phenotype (new panels N and O of Figure S9, see below), which correlated with increased levels of extended-open active $\alpha_5\beta_1$ integrin (9EG7) at their cell surface (new panel M of Figure S9). This increased spreading phenotype was accompanied by thick and likely stable fibrillar focal adhesions positive for both the 9EG7-labeled ligand-bound β_1 integrin and the mature focal adhesion marker vinculin (new panel P of Figure S9, see below). In contrast, focal adhesions appeared more discrete in wildtype conditions, likely reflecting the dynamic integrin turnover at the plasma membrane, suitable for cell migration (doi: 10.1155/2012/310616). The cell spreading assays are now documented in the manuscript (page 11, lines 13-19).

Minor Comments:

- For the SNAP-BG experiments, the entire WB should be supplied. The blots should demonstrate a molecular weight shift when (for example) GalT-GFP-SNAP captures mAb13-BG.

The retrograde trafficking assay using the BG/SNAP strategy has already been published and the molecular weight shift that is mentioned by the Reviewer has been documented therein. We would kindly like to refer the Reviewer to doi: 10.1038/ncb3287. Uncropped Western blots with molecular weight markers are provided as source data as part of the revised manuscript files.

- Is there evidence that the integrins themselves are present in the TGN rather than just the conformation-specific antibodies? Is it possible that the antibodies trafficked to the TGN but not integrin? In the SNAP-GFP pulldown experiments, can you also blot for integrin to demonstrate this?

In our foundational 2016 NCB paper, we have used cell surface benzylguanine-modifications (akin to cell surface biotinylation) to identify the proteome of the retrograde route in HeLa cells (doi: 10.1038/ncb3287). $\alpha_5\beta_1$ integrin was amongst the most robust and highly scored retrograde clients. No antibody was used in these experiments, and it could be concluded that $\alpha_5\beta_1$ integrin itself was indeed transported from the plasma membrane into the Golgi compartment.

In the same study, we used several antibodies (9EG7, 12G10, mAb13, SNAKA51 and mAb16) and ligands (fibronectin fragment FN-50K) against different $\alpha_5\beta_1$ integrin conformations and found that only the ones that recognize the bent-closed conformation (mAb13 for the β_1 chain, and mAb16 for the α_5 chain) reacted with the Golgi-localized GalT-GFP-SNAP fusion protein following incubation with live cells. Very

clearly, retrograde trafficking is a selective process that can be followed in the antibody uptake modality.

From a technical perspective one may mention that in the antibody uptake modality, arrival in the Golgi is scored by antibody immunoprecipitation using protein G and blotting against the SNAPtag protein. Since the covalent reaction has occurred between benzylguanine-tagged antibodies and the GalT-GFP-SNAP fusion protein, it is in this setup not possible to differentiate by Western blotting antibody-associated $\alpha_5\beta_1$ integrin that has reached the Golgi from the one that has not.

- Antibodies are known to have N-glycans. Are the antibodies (mAb13, mAb16, 9EG7, SNAP51) free from nonspecific binding to Gal3? Treating them with PNGase could help eliminate potential interactions.

We are indeed aware of this important point. When performing co-binding/co-uptake assays, we systematically operate sequential binding incubations at 4 °C of both partners, starting with galectin-3, extensive washes to remove unbound galectin-3, followed by the antibodies themselves. In this way, galectin-3 is already associated with its cell surface targets before the antibodies are added.

The specificity of the observed colocalizations validates this protocol: While galectin-3 strongly colocalized with 2 different antibodies, i.e., mAb13 (β_1 integrin) and mAb16 (α_5 integrin) that both have in common to recognize the bent-closed conformation of $\alpha_5\beta_1$ integrin, low colocalization was observed with the 2 other antibodies, i.e., 9EG7 (β_1 integrin) and SNAKA51 (α_5 integrin) that both target the extended conformation.

In addition, all these antibodies (mAb13, mAb16, 9EG7 and SNAKA51) are monoclonal immunoglobulin G (IgG) antibodies that, like most of IgGs, may contain poorly exposed single N-glycan on their heavy chains, unlikely suitable to efficiently bind Gal3 (doi: [10.1093/glycob/cwn015](https://doi.org/10.1093/glycob/cwn015)).

Following the two-pronged approach that was already mentioned above, we performed antibody-independent experiments in controlled in vitro conditions to further harden the antibody-based conclusions from the complex cellular systems. Specifically, we performed protein-protein interaction assays using our purified and conformationally tunable $\alpha_5\beta_1$ integrin. We hereby confirmed that galectin-3 indeed preferentially binds to the bent-closed conformer (Figures 4F and 4G), and that only this conformer nucleates galectin-3 oligomers (Figure 4H-J).

- CHC knockdown may affect trafficking of newly synthesized ITGA5/ITGB1, potentially reducing integrin levels at the PM. The authors should clarify whether this effect was considered in their data quantification.

As suggested by the reviewer, we analyzed integrin levels at the plasma membrane under control and clathrin depletion conditions. No major binding difference was

observed (new panel D of Figure S1, see below; updated in the manuscript page 4, lines 2-3), which confirmed that our data quantification was done under appropriate conditions.

- CBD treatment reduces retrograde trafficking, but internalization was assessed over just 10 minutes. Extending this time frame could confirm Golgi localization and validate CIE in retrograde trafficking.

To address this comment, we have first of all determined the effect of prolonged CBD treatment (as needed for the retrograde transport assay) on the endocytosis of the bent-closed $\alpha_5\beta_1$ integrin. In recently published work, we indeed observed that prolonged inhibition of endocytic machinery may lead to compensatory changes in uptake mechanisms (doi: 10.3390/biom14091169).

While under acute CBD treatment conditions (30 min CBD pretreatment at 37 °C, and 10 min continuous incubation at 37 °C with mAb13 in the presence of CBD) a significant inhibition of endocytosis was observed, this effect had vanished under prolonged CBD treatment conditions (30 min CBD pretreatment and 1h30 incubation with mAb13 and CBD, all at 37 °C) (see below). These findings suggested that compensatory endocytic processes may have kicked in upon prolonged inhibition of dynein function.

Even if the starting conditions were not optimal, we nevertheless performed retrograde trafficking analysis under prolonged treatment conditions. Unexpectedly, we even observed an increase of SNAP-mAb13 crosslinking signal (see below).

The reasons for this increased signal are not clear to us and need further investigation. One might speculate on the following points:

- The compensatory endocytic process that is apparently observed in the prolonged treatment protocol might for some reason couple highly efficiently to retrograde trafficking. To test this, one will need to identify the molecular nature of this compensatory endocytic process.
- We can't exclude at this stage that the GalT-GFP-SNAP capture reagent itself isn't mislocalized in CBD treated cells. To support this hypothesis, we have found that the morphology of the Golgi compartment was clearly affected under prolonged dynein inhibition (see below): While being perinuclear and compact in control condition, Golgi localized GalT-GFP-SNAP appeared more fragmented in the prolonged presence of CBD with a likely redistribution into endosomal compartments. To test this, one will need to perform high-resolution imaging under the corresponding experimental conditions.

All the approaches that are mentioned above are major investments that would go beyond the scope of the current revision process. We have therefore decided to keep these for follow-up work.

- Variation in Galectin Staining: In Fig. 7D, there's a noticeable difference in Galectin staining between conditions in the WT image. The authors should clarify the cause of this variation.

As discussed earlier, working with transiently transfected dKO-MKF cells was quite challenging due to intrinsic heterogeneities (integrin expression levels, variations in galectin-3 binding, ...). This being said, we agree that the panel needed to be redesigned properly, and the signals needed to be adjusted between different conditions. In addition, a labeling error was corrected: Gal3 is in red, and integrins in green. In the revised version of the manuscript, the panel was correspondingly updated, thanks for having pointed this out.

- Site-Specific Mutants for Calf Region Glycosites: The authors refer to N918 as the critical leg region glycan extensively in the text. However, they only present data where all four calf region NXST sites are mutated (deltaMT). It is unclear why individual calf sites have not been examined. At the very least this should be clarified in the text and references to N918 tempered.

Following the Reviewer's suggestion, we have now analyzed the N918Q glycosite mutation (human N867Q) alone outside the context of Δ MP. Gratifyingly, we found that it overall phenocopied Δ MP: (i) reduced colocalization between galectin-3 and the bent-closed conformation of $\alpha_5\beta_1$ integrin (mAb13), (ii) reduced endocytic uptake of the bent-closed conformer, (iii) slightly increased endocytic uptake of the extended-open conformation (new panels F and G of Figures 7, see below; now updated in the manuscript on page 12, lines 6-11).

- Figure 5E needs statistical analysis.

As requested by the Reviewer, statistical analysis is now being provided in the revised version of Figure 5E (see below), and documented in the caption.

- In Fig6I, some of the structures in yellow circles look bent at first glance. Please confirm that they are not mislabeled.

Thanks for this valuable indication. Some particles were indeed mislabeled, which has been corrected.

Reviewer #3 (Remarks to the Author):

In this work the authors investigate the endocytosis pathway of integrin $\alpha 5\beta 1$ in mammalian cells. They use conformation specific antibodies to show that only the inactive bent closed conformation undergoes retrograde trafficking from the plasma membrane to the golgi, and that this is mediated by Gal3 (rather than clathrin). The authors show that this endocytosis of the inactive bent close $\alpha 5\beta 1$ integrin is preferentially occurring via elongated CLIC compartments and dependent on dynein, both previously described hallmarks of this clathrin-independent endocytic pathway. Biochemical experiments using purified integrins are used to show that only this inactive bent close conformation complex nucleates the formation of Gal3 oligomers. Similar Gal3 oligomers are then found to be formed on the surface of cells. Using cryoEM, the authors present a structural characterization of Gal3 oligomers in interaction with the inactive bent close conformation of the integrin complex. Finally mapping glycan sites on these structures, the authors propose a model in which 3 glycans located near Gal3 binding sites and close to each other in the bent closed conformation only, could act as the trigger for Gal3 oligomerization. Removing these glycans results in decreased Gal3 integrin colocalization on cell plasma membranes and decreased internalization of the inactive bent closed integrin complex.

The manuscript is very interesting and well written, and the data is rich. In my view the part that is less convincing or more ambiguous as such is the final structural interpretations, on which the “glycoswitch” part of the model is based. My impression is that the attractive model put forward by the authors is one possibility, that does not

disagree with the data presented here. However, as such, it seems difficult to exclude alternative interpretations. My recommendation for publication would be to repeat a couple of endocytosis assays used in the first figures of the paper and/or Gal3 NS oligomerization analysis including: i) glycan mutants for the sites proposed to be involved in the glycoswitch, as well as ii) mutants for a couple of other glycosylation sites, proposed not to be involved in the glycoswitch, as important controls.

We would like to thank the Reviewer for these valuable suggestions. As shown below, these have been addressed experimentally.

Alternatively, the glycoswitch claim could be downgraded in the proposed model.

Specific points

Maybe clarify the fate of integrin complex internalised by clathrin for a broader readership.

Clathrin-dependent endocytosis of integrins has been largely explored in the literature and linked to the dynamic turn-over of focal adhesions that allows for efficient cell adhesion and cell migration (doi: 10.1038/ncb1262 ; doi: 10.1016/j.febslet.2009.03.037). Many clathrin adaptors, such as Dab2, Numb, Eps8, and AP-2 (doi: 10.1016/j.devcel.2007.05.003; doi: 10.1016/j.cub.2015.09.049), and the scission GTPase dynamin-2 have been implicated in the regulation of integrin endocytosis near focal adhesion sites (doi: 10.1016/j.devcel.2007.05.003; doi: 10.1016/j.cub.2015.09.049; doi: 10.1091/mbc.E10-09-0785; doi: 10.1073/pnas.262791999). Numb, Kif15, and AP-2 are also involved in the clathrin-dependent endocytosis of active β_1 integrin towards the ventral site and the leading-edge to ensure efficient cell migration (doi: 10.1016/j.devcel.2007.05.003; doi: 10.1016/j.febslet.2009.03.037; doi: 10.1083/jcb.200904054).

A corresponding paragraph has been added to the revised version of the manuscript (page 2, lines 31-34).

Fig. 1: colP on Vps35: is Gal-3 also found in the IP?

As suggested by the Reviewer, we have performed new co-IP experiments on VPS35 in which we have included galectin-3 in the process. In agreement with our co-immunostaining results (Figures 1G and 1H), we have found that not only VPS35 robustly co-immunoprecipitated with the mAb13-labeled bent-closed conformation of $\alpha_5\beta_1$ integrin, but also galectin-3 (new panel J of Figure S1, see below; updated in the revised manuscript, page 4, lines 17-19). In contrast, no co-immunoprecipitation of VPS35 or Gal3 was observed with the 9EG7-labeled extended conformation of $\alpha_5\beta_1$ integrin.

How is this IP affected when the cells are treated with I3?

Based on our model, we think that addressing an I3 effect on that interaction might not be rationally meaningful for the following reasons. First, we have demonstrated that Gal3 binding to $\alpha_5\beta_1$ integrin is glycan-dependent as illustrated by I3 clearly preventing this interaction (Figure S4J). It would therefore be expected to not coimmunoprecipitate Gal3 in this condition. Second, we have shown that I3 treatment substantially inhibits mAb13 labeled integrin internalization (Figure 2J, top) and since the Co-IP experiment is performed upon 20 minutes internalization, we may not immunoprecipitate decent level of relevant and meaningful endocytosed integrin.

Fig. 2: It seems that the more exogenous recombinant Gal3 is added to the system, the more integrin uptake and retrograde transport is observed. Does it mean that Gal3 is limiting and that the amount of Gal3 is enough to regulate the process?

The amount of galectin-3 can clearly be used to tune the endocytic uptake of cell surface glycoproteins. On RPE-1 cells that express endogenous levels of galectin-3, we show that incubation with 20 nM of exogenous galectin-3 allows to stimulate the internalization of bent-closed inactive $\alpha_5\beta_1$ integrin in a process that plateaus between 200 nM and 400 nM (Figure 2L, below; new statistical analysis).

In a previous study, knock down of galectin-3 inhibited CD44 uptake, which was then rescued with as little as 0.4 nM of exogenous Gal3 and plateaued at 40 nM (doi: 10.1038/ncb2970).

Although many studies have reported on serum concentrations of galectin-3 in the range of 0.4 (0.016 nM) to 10 ng/ml (0.4 nM) in healthy tissues (doi: 10.1038/srep40994; doi: 10.1038/s41598-024-83499-w; doi: 10.3389/fnins.2018.00430; PMID: 10778968), higher local concentrations of Gal3 through controlled paracrine secretion cannot be excluded (doi: 10.1242/jcs.208884). It therefore appears entirely possible that glycoprotein uptake could be tuned by modulating galectin concentrations in tissues.

In a recently published paper, we have demonstrated an alternative possibility. We have found growth factors (notably EGF) trigger desialylation of glycoproteins (including integrins), leading to enhanced galectin-3 binding and GL-Lect driven endocytosis in physiological processes such as cell migration and bone remodeling (doi: 10.1038/s41556-025-01616-x). In this case, tuning endocytic uptake can be achieved without altering galectin concentrations.

Is there any corresponding evidence maybe that Gal3 levels are regulated in human cells (or diseases in the case of imbalance etc.).

Galectin-3 levels have indeed been associated with pathophysiological situations. Galectin-3 initiates an acute inflammatory response through macrophages recruitment at sites of tissue damage, hence activating proinflammatory pathways. In case of inefficient repair, this can lead to fibrotic disease. In myocardium remodeling, increased levels of galectin-3 induce cardiac defects leading to cardiac dysfunction due to fibro-inflammation and cardiomyocyte loss, defining serum galectin-3 levels as a hallmark of heart failure and atrial fibrillation, and a marker of cardio-vascular dysfunction (doi: 10.1161/CIRCHEARTFAILURE.121.008510; doi: 10.1016/j.cardfail.2022.09.017; doi: 10.3892/mmr.2017.8323). In chronic kidney disease, increased galectin-3 levels in serum and urinal fluids were associated with defects in glomerular filtration, proteinuria, and high risk of chronic renal fibrosis (doi: 10.3390/biomedicines10030585; doi: 10.3389/fmed.2021.748225). Elevated levels of galectin-3 were also associated with other inflammatory and fibrotic diseases, including Alzheimer and Huntington neuroinflammation (doi: 10.1007/s00401-019-02013-z; doi: 10.1038/s41467-019-11441-0), and pulmonary infections such as idiopathic pulmonary fibrosis, Hermansky-Pudlak syndrome, and COVID-19 (doi: 10.2332/allergolint.O-06-449; doi: 10.1165/rcmb.2013-0025OC; doi: 10.1016/j.rmed.2021.106556).

A corresponding paragraph has been provided in the revised version of the manuscript (page 3, lines 6-14).

Fig. 3A: it would be useful to mention the approximate number of HRP positive structures that were counted in each of the 4 experiments (is it ~10 or 100 or 1000?)

Following the Reviewer's suggestion, this information is now fully documented in the revised legend of the figure.

Figure 4I:

What about Gal3 wild type alone, without any integrin nor I3 (not even to reproduce “elution conditions”), at 4uM? Does it oligomerise as seen by NS?

In the absence of integrin or I3, galectin-3 in solution remains monomeric as quantified by negative-stain EM and newly documented in Figure 4I (see below). Only 2% oligomers were counted for a total of 15k galectin-3 particles, a percentage that is close to the one found for the Gal3+I3 condition (new Figure 4I; see below). The monomeric state of Gal3 without any glycan source is in agreement with previously reported DLS analysis, where Gal3 up to 150 μ M remained monomeric in solution (doi: 10.1074/jbc.C112.358002).

Figure 6H: The authors test the stabilization, upon Mn²⁺ activation in presence or absence of Gal3, of the inactive bent close conformation by checking binding of mAb13, but what happens to the binding of 9EG7?

To address this excellent point, we have repeated the Mn²⁺/RGD activation and also measured the effect on 9EG7 binding. As seen before, mAb13 binding dramatically dropped upon integrin activation (new panel F of Figure S8, see below). Importantly, 9EG7 signal concomitantly increased (new panel E of Figure S8, see below), suggesting that the same $\alpha_5\beta_1$ integrin molecules were converted from one conformational state to the other. An updated paragraph is now provided in our revised manuscript (page 10, lines 9-11).

Also does NS provide evidence of where, on the integrin complex, each antibody binds and how the binding may interfere with Gal3?

We have used a two-pronged approach — based on antibodies in the cellular system and on purified proteins in model membrane systems — that has led us to the conclusion of preferential galectin-3 binding to the bent closed conformation of $\alpha_5\beta_1$ integrin.

(1) We have indeed used conformation-specific antibodies and performed co-binding, co-uptake, and interaction assays with galectin-3 in the cellular context. These antibodies, namely mAb13 (doi: 10.1083/jcb.109.2.863) and 9EG7 (doi: 10.1074/jbc.270.43.25570), have been extensively characterized and their epitopes were elegantly localized onto $\alpha_5\beta_1$ integrin by negative-stain EM single particle analysis in previous work (doi: 10.1073/pnas.1605074113): mAb13 binds to an epitope located at the β I domain of β_1 integrin, stabilizing the closed conformation (doi: 10.1074/jbc.271.34.20365; doi: 10.1083/jcb.109.2.863; doi: 10.1073/pnas.1605074113), while 9EG7 binds to an epitope located at the EGF-repeat leg domain of β_1 integrin, stabilizing the extended conformation (doi: 10.1073/pnas.1605074113; doi: 10.1074/jbc.270.43.25570).

Since almost all glycosites that were identified in our study for galectin-3 binding and endocytic functions are on the α_5 subunit, it seems unlikely that they interfere with mAb13 or 9EG7 binding to the β_1 subunit. Of note, we also used conformational state-specific antibodies that target the α_5 subunit (i.e., mAb16 for the bent-closed conformation, and SNAKA51 for the extended-open conformation), with qualitative and quantitatively similar results as for the β_1 subunit-specific antibodies. Having similar results with 2 different sets of antibodies whose common denominator is the conformer specificity makes us confident about our results.

We would also like to point out that we have designed our experimental protocol to make an interference with galectin-3 rather unlikely. Specifically, we have performed sequential binding: first galectin-3 binding to its cellular targets on cells, washing to remove unbound galectin-3, and only then incubation with the antibodies. This staged binding protocol is expected to strongly diminish the risk that galectin-3 binds to the antibodies and/or that antibody binding interferes with galectin-3 binding.

To confirm our antibody-based findings, we have used purified $\alpha_5\beta_1$ integrin and measured direct integrin/galectin-3 interaction in vitro, in an approach that did not rely on the use of any antibody. We thereby confirmed preferential galectin-3 binding to the bent-closed conformation of $\alpha_5\beta_1$ integrin, and the unique ability of this specific conformer to scaffold defined and functional galectin-3 oligomers (Figure 4F-J).

In the text related to Figure 7, p 10 l 13, it reads to me as the structural analysis points to the importance of the membrane proximal glycans only, rather than the distal ones.

However, 2 distal sites are then also tested and included in the glycoswitch model. So it may be useful to rephrase and clarify this sentence.

As suggested by the Reviewer, this sentence has now been rephrased and clarified (page 11, line 9). Of note, the distal sites, N356 and N417, were also considered and documented on page 11, lines 38-40 and page 12, line 18-24, respectively.

Despite efficient cell surface expression of the deltaMP mutant, the overlap with Gal3 is lost and the authors conclude: “This striking result suggests that the removal of the membrane-proximal glycosylation sites on the α_5 chain diminishes the capacity of bent closed $\alpha_5\beta_1$ integrin to trigger Gal3 oligomerization”. I think this needs some clarification. The direct conclusion from this experiment seems to be that deltaMP and Gal3 do not colocalise, hence do not interact. This means that the problem could occur before any possible Gal3 oligomerization, by losing binding in the first place.

We agree that the wording was somewhat ambiguous. We have rephrased the sentence in the revised version of the manuscript (page 11, lines 23-25).

Figure 7D-F: Here it seems that removing any of the glycans tested (leg and head pieces) has an effect. In order to convince the reader that the effect observed is indeed specific to removing these 3 glycans, it would be important to use mutants for other glycan sites and show that these have no effect. This would be needed to support the glycoswitch model.

To address this excellent point, we have now performed mAb13 binding and endocytosis experiments in dKO-MKF cells transiently expressing either N642Q (N593Q in human) or N822Q (N773Q in human) α_5 integrin glycosite mutants. These are localized in the thigh and calf domains, respectively. While the Δ MP mutant consistently produced a significant decrease of mAb13-labeled β_1 integrin endocytosis, none of these newly tested glycosites mutants showed significant endocytic effects (new panel D of Figure S10, see below). Of note, mAb13 surface binding levels remained almost unchanged for these mutants, when compared to control conditions (new panel C of Figure S10, see below). An updated paragraph is now added in the revised manuscript (page 12, lines 12-17).

Based on Figure S9C it seems that the integrin complex is quite covered in glycans and hence even in an open conformation it seems that a couple of glycans might still be close enough to bind several Gal3. To visually convince the reader that this is maybe unlikely to occur, a figure similar to S9C plotting all the glycans in the extended conformation of the integrin complex would also be useful.

As suggested by the Reviewer, the GlycoShield simulation of the extended conformation of $\alpha_5\beta_1$ integrin has been now added as new panel C in Figure S9 (see below).

In addition is there any evidence that removing these glycans does not affect the conformational equilibrium of the integrin complex in the first place?

The membrane binding levels of each conformational state-specific antibody upon removal of these glycans are documented in the manuscript. While surface levels of bent-closed $\alpha_5\beta_1$ integrin (mAb13) remained globally unchanged, we observed increased levels of the extended-open conformation (9EG7) in both Δ MP and N307Q mutants (Figures S9M and S10B). The removal of these glycans specifically inhibited the endocytic uptake of the bent-closed conformation (mAb13), which may suggest that the excess of non-internalized bent-closed $\alpha_5\beta_1$ integrin could become substrate for activation and switch to extended-open conformation. This statement is now provided in the revised manuscript (page 11, lines 30-33).

In agreement, Δ MP and N307Q expressing cells with increased extended-open $\alpha_5\beta_1$ integrin (9EG7) at the cell surface exhibited increased cell spreading with thicker and fibrillar vinculin/9EG7-labelled focal adhesions (new panels N-P of Figure S9, see below; updated in the manuscript on page 11, lines 13-19).

Importantly, it looks like at this resolution the Gal3 CTDs could be equally fitted facing in one direction or the other. The translucent red part is proposed to be the part where these multimerization NTDs join but these Nter are long unstructured regions and the density could also be a partial / substoichiometric / more flexible density for an extra copy of Gal3 CTD for instance (as pentamers were previously described in the literature). At this resolution, it seems difficult to unambiguously conclude. It is probably wiser to say that the structural data can be interpreted in agreement with this hypothesis, rather than claiming that the structural data supports/demonstrates this model. (Including in the discussion “These defined oligomeric structures that orient four Gal3 with glycan binding sites pointing in the same outward direction are striking hallmarks of the Gal3 tetramer that we have discovered.”)

Following the Reviewer’s comment, we adapted the wording of the corresponding paragraphs (page 9, lines 14-17 and 25-28). The following paragraph in the discussion was removed altogether:

The unstructured N-terminal proline-rich domain of Gal3⁸⁵ is visualized for the first time by our study. Strikingly, within oligomers the N-termini of constituting Gal3 molecules overlap. This domain has been described to undergo biomolecular condensate formation^{68, 86, 87}, which would be one possible mechanism by which Gal3 oligomers might be stabilized in a way that allows them to accommodate variable numbers of Gal3 molecules.